# Climate response to Nature Future scenarios in a regional Earth System Model

Petra Sieber [1] ✉, Dirk Nikolaus Karger [2], Niklaus E. Zimmermann [2], Sara Si-Moussi[3], Wilfried Thuiller [3], Gabriele Midolo [4], Milan Chytrý [5], Irena Axmanová [5], Jonas Schwaab[1,6,7], Edouard L. Davin[8,9,10], Matthieu Leclair[11], Stephan Kambach [12], Helge Bruelheide [12,13], Thomas Hickler [14,15], Zvjezdana Stančić [16], Idoia Biurrun [17], Behlül Güler[5,18], Jürgen Dengler [19,20], Jan Divíšek [5], Peter H. Verburg [21] & Sonia I. Seneviratne [1]

Land use management can help address the human-induced climate and biodiversity crises. However, substantial transformations in land systems are needed to meet internationally agreed targets concerning nature conservation, restoration, sustainable agriculture, and tree cover. Such transformations influence land-atmosphere exchanges of energy, water, and carbon, and could have particularly strong effects on local to regional climate through changes in albedo and evapotranspiration. Here, we explore how land use management in Europe, consistent with the Kunming-Montreal Global Biodiversity Framework, the Nature Futures Framework, and a sustainable low-emissions scenario, would affect the European climate mid-century. Using Earth System Modelling and detailed land use, habitat, and species projections, we show that policy implementation guided by relational values (Nature as Culture) could lead to additional warming and drying further threatening biodiversity and human well-being. Conversely, promoting intrinsic values (Nature for Nature) or ecosystem services (Nature for Society) would not add major challenges for climate adaptation and mitigation. These different outcomes highlight the need to develop integrative land use scenarios that enhance biodiversity and stabilise the climate, while considering feedbacks from land to the atmosphere. Such scenarios could help navigate trade-offs and inform policy implementation in Europe.

The climate and biodiversity crises are among the most urgent challenges that society must address to ensure the stability and resilience of the Earth system and avoid harmful impacts on humans[1–4]. Climate change and biodiversity loss are intertwined due to common drivers such as land use and pollution, and combined impacts on human well-being and nature's contributions to people[5,6]. Moreover, climate change is expected to increasingly affect species, habitats, and ecosystem functioning[5,7–9]. In turn, biodiversity loss, in conjunction with ecosystem change and degradation, can exacerbate climate change

through reduced ecosystem productivity, stability, and resilience[10–13], and impair ecosystems' contribution to intended mitigation and adaptation measures[7,14–16].

Land use management is central to addressing both crises[4,6,17–19]. The protection, restoration, and improved management of carbon- and biodiversity-rich ecosystems could contribute to achieving the goals of the Paris Agreement and the Kunming-Montreal Global Biodiversity Framework[20–22]. Furthermore, trees and forests can contribute to climate change adaptation by cooling temperatures locally,

particularly during daytime and the boreal summer[23,24]. However, land systems are multifunctional and complex, so managing nature for climate and biodiversity goals may compete, and it may affect current livelihoods[25–27]. Many interventions have been shown to entail trade-offs among the societal benefits and values associated with land[28,29]. For instance, tree planting and forest expansion to increase above-ground carbon stocks may threaten grassland biodiversity and eco-system services[30]; conversely, protecting important habitats for biodiversity may restrict areas available for renewable energy or agri-cultural production[31]. Therefore, integrative land use scenarios that jointly consider various demands, pressures, and impacts are needed to mitigate such conflicts[32].

Integrative land use scenarios for Europe show that existing policies for climate and biodiversity protection could lead to land system (i.e., use and intensity) changes on almost one third of Eur-opean land by 2050[33]. However, the consequences for climate are not yet understood. Land use influences energy, water, and carbon fluxes at the Earth's surface, which in turn affect climate variables such as temperature and precipitation. Earth System Models (ESMs) are well suited to simulate such processes, and are commonly used to explore the effects of, e.g., large-scale forestation and deforestation[34–36], land use and management[37–39], or historical and projected future land use and land cover changes[40–42]. However, to date, climate projections with ESMs have focused on the Shared Socioeconomic Pathways (SSPs, describing trends in population, economy, technologies, and envir-onmental regulation) and Representative Concentration Pathways (RCPs, describing greenhouse gas emissions)[43–45]. These scenarios have been criticised for being too focused on climate change and neglecting other sustainability challenges such as biodiversity loss and ecosystem degradation[45–50]. For instance, impact modelling showed that a sustainable low-emissions scenario (i.e., combining SSP1 and RCP2.6) would not halt further declines in biodiversity and regulating ecosystem services in many world regions[51]. To explore specific poli-cies for improved nature management (i.e., the use of natural resour-ces balancing human needs and the health of natural systems), drivers of change in biodiversity and ecosystem services must be considered in the scenario development[46,49,51].

To support the development of such nature-focused scenarios, the Intergovernmental Science-Policy Platform on Biodiversity and Ecosystem Services (IPBES) developed the Nature Futures Framework. This framework offers three main ways in which people value and interact with nature: the Nature for Nature (NfN) perspective illustrates intrinsic values and promotes biodiversity and ecosystems for their own sake; the Nature for Society (NfS) perspective illustrates instru-mental values and prioritises ecosystem services such as climate reg-ulation and food production; and the Nature as Culture (NaC) perspective illustrates relational values and focuses on cultural land-scapes and traditions[45,49]. Based on these perspectives, scenarios can be developed that describe alternative, desirable pathways for nature and its contributions to people. Such scenarios can help explore the consequences of future policies (e.g., different implementations of the 30×30 target) and inspire the societal transformation necessary to achieve sustainability goals[49].

The Nature Futures Framework was recently used as a lens for developing plural land system scenarios for Europe in 2050[33]: a reference that complies with the projected population, commodity demands, and climate under a sustainable low-emissions scenario (SSP1-2.6), and three scenarios that additionally implement targets of the Kunming-Montreal Global Biodiversity Framework and the Eur-opean Union's Green Deal. Dou et al.[33] implemented these climate and biodiversity policies while focusing on either intrinsic, instru-mental, or relational values according to the Nature Futures Frame-work (Fig. 1). Specifically, Dou et al.[33] prioritised biodiversity conservation and land sparing in the NfN scenario, climate change mitigation and land sharing in the NfS scenario, and cultural

landscapes and human-nature interactions in the NaC scenario. In an NfS perspective, various regulating and provisioning ecosystem services could be considered[45], and climate change mitigation was chosen as one key function in the climate crisis. Several provisioning services (e.g., production of crops, livestock, and wood) are guar-anteed in all scenarios to meet human demands under the SSP1 storyline. Biodiversity is represented by habitat suitability for terrestrial vertebrate species, climate change mitigation by carbon sequestration potential in forests, and cultural value by agricultural heritage landscapes (i.e., rural areas characterised by traditions, low management intensity, landscape heterogeneity, and high value and meaning). Detailed scenario assumptions, their origins, and para-meterisation are described in Dou et al.[33]. The resulting land system maps differ in the area covered by various land systems (e.g., greatest proportions of low-intensity systems in NfN, high-intensity forests in NfS, and multi-purpose mosaic systems in NaC) and their spatial distribution across Europe. The scenarios diverge from the SSP1 reference on ~30% of European land, and among each other on ~20% of European land[33]. Land use changes on such a scale could have profound impacts on land surface processes, with particularly strong effects on local to regional climate through changes in albedo and evapotranspiration (i.e., biogeophysical effects)[41,52–54].

In this study, we aim to incorporate the land system scenarios of Dou et al.[33] in a regional ESM[55] to simulate the biogeophysical climate effects of implementing European policies for improved nature man-agement with a focus on biodiversity conservation, climate change mitigation, or cultural services. Implementing these scenarios in an ESM is challenging because ESMs usually represent land use through the spatial distribution of plant functional types (PFTs), which control important ecosystem properties (e.g., vegetation structure, rooting depth, leaf reflectance, photosynthetic capacity) and processes (e.g., reflection of sunlight, evapotranspiration, and photosynthesis). Land use changes such as afforestation, cropland expansion, or land aban-donment can be expressed as changes in PFTs. However, the same land system (e.g., forest/shrub and grassland mosaic) can consist of dif-ferent PFTs depending on environmental conditions (e.g., cold cli-mates and acidic soils favour needleleaf evergreen trees, whereas temperate climates with marked seasonality and fertile soils support deciduous broadleaf trees). To obtain the PFT composition while accounting for future habitat suitability and plausible species com-position, for each land system scenario we harness high-resolution predictions of EUNIS (European Nature Information System)[56] habitat types at level 3[57] and disaggregate them into PFTs according to the species composition in >800k plots of the European Vegetation Archive (EVA)[58], the most extensive repository of vegetation records for Europe (Fig. 2 and Methods).

This approach enables us to answer the following questions: (1) How do different implementations of European climate and biodi-versity policies affect the PFT distribution across Europe? (2) At which scale and magnitude do the scenarios affect mean climate or extreme temperatures? (3) What are the underlying biogeophysical and ecosystem-related drivers? Finally, we assess if nature management focusing on biodiversity conservation, climate change mitigation, or cultural services could cause additional challenges for regional climate adaptation and mitigation.

## Results

### Effects on distributions of plant functional types

To quantify the effects of improved nature management by 2050, we compare the NfN, NfS, and NaC scenarios with the SSP1 reference, which complies with future demands and climate but does not include specific European climate and biodiversity policies. To contextualise the land system differences and their effects on PFT cover in 2050, we also analyse land use changes over time since 2015 under the different scenarios. Changes in land systems, habitat types, and PFTs are shown

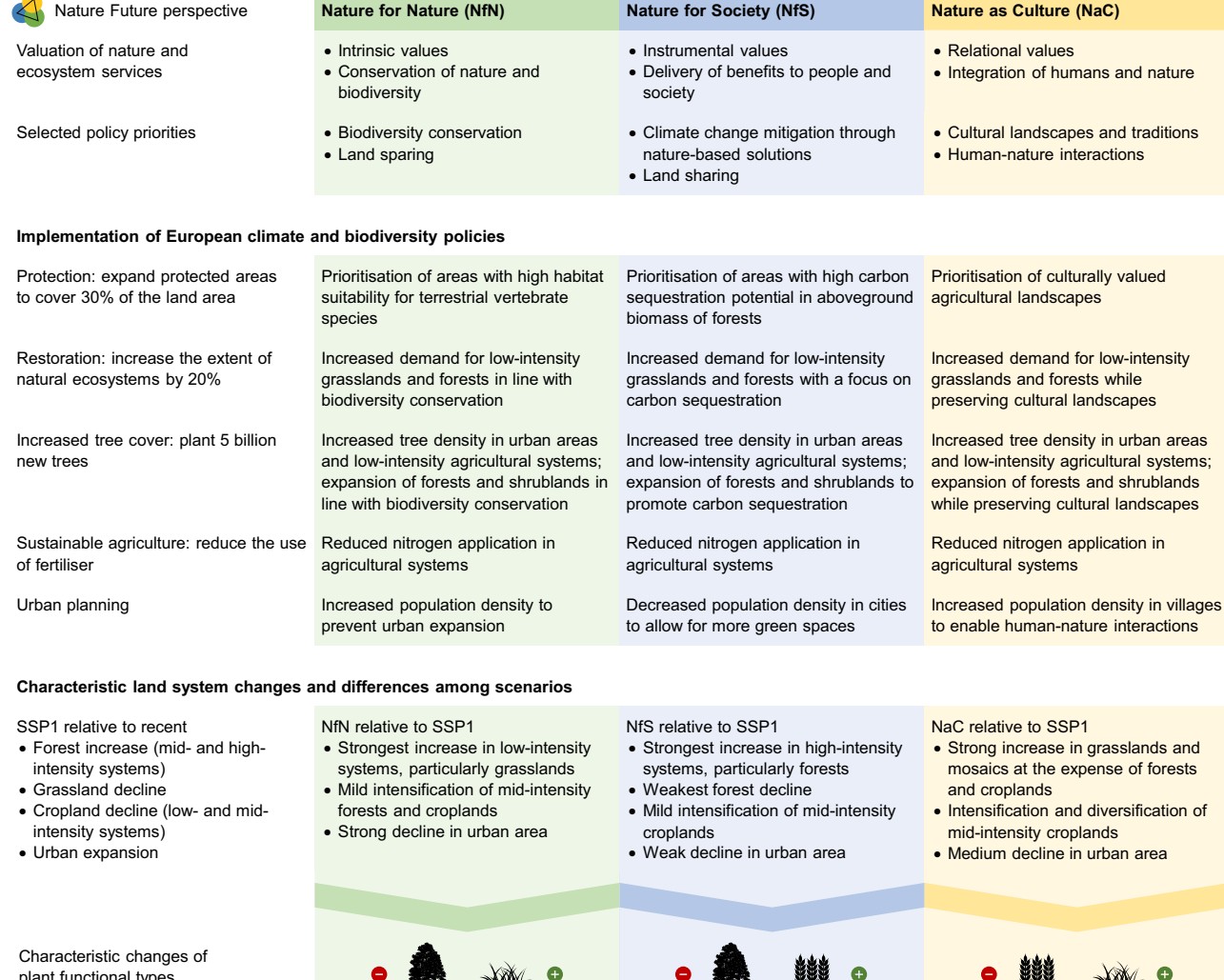

Fig. 1 | **Characteristics of the Nature Future scenarios used in this study.** The scenarios Nature for Nature (NfN), Nature for Society (NfS), and Nature as Culture (NaC) reflect three ways in which people value nature. These perspectives were utilised by Dou et al.[33] to generate land system scenarios for Europe in 2050 that implement European climate and biodiversity policies, derived from the Kunming-Montreal Global Biodiversity Framework and the European Union's Green Deal, in three different ways. The characteristics of each scenario are presented as differences to the reference (SSP1), which also represents land use in 2050 under the SSP1 "Taking the Green Road" storyline but without implementation of the European policies. Intensification means changes in land use intensity (e.g., mid to high intensity) and diversification transitions to mosaic systems (e.g., cropland to forest/shrub and cropland mosaic). Resulting changes in plant functional types are quantified in this study through the workflow illustrated in Fig. 2.

in Fig. 3a. Under SSP1, mid- and high-intensity forest systems strongly expand after 2015, and we find a corresponding increase in tree PFT cover, particularly broadleaf deciduous trees (Fig. 3a). Grassland and cropland systems decline, resulting primarily in reduced C3 grass and rainfed crop PFT cover (note that grass PFTs in ESMs represent herbs except annual crops). Mosaic systems expand with small effects on several PFTs. Although shrublands decline, shrub PFT cover increases due to gains in, e.g., forest/shrub and cropland mosaics.

Compared with the SSP1 reference in 2050, the NfN, NfS, and NaC scenarios show PFT cover differences on 10–21% of European land (see Fig. 2a for the considered region, hereafter EU+). These differences derive directly from the scenario targets and priorities (e.g., restoration in all scenarios, carbon sequestration in NfS), but also from compensatory effects to meet multiple demands (e.g., intensification in parts of the forest area to compensate for increases in more natural land systems elsewhere), regional differences in demands and supply (e.g., depending on population and yields), and environmental differences that affect the habitat and PFT composition of land systems. At the European level, all three scenarios project a substantial expansion of low-intensity land systems to meet the targets for protection, restoration, tree cover, and sustainable agriculture (see Fig. 1 and Dou et al.[33] for details). This conversion and extensification is enabled by intensification of mid-intensity forests and croplands in other regions to satisfy demands. Consequently, the scenarios hardly expand forests but increase grasslands after 2015. As a result, they have less tree PFT cover and more grass PFT cover than the reference and tend to counterbalance the changes under SSP1 in 2050 compared with recent PFT cover (Fig. 3c). The NfN scenario promoting biodiversity conservation shows the strongest increase in low-intensity land systems, particularly grasslands, while the NfS scenario promoting climate change mitigation shows the strongest increase in high-intensity land systems, particularly forests. NfN and NfS foresee only mild intensification of mid-intensity croplands, so that crop PFT cover is higher than under SSP1 because of the greater co-occuring extensification (yet cropland declines compared with 2015). Contrarily, the NaC scenario focusing on cultural landscapes foresees strong intensification and

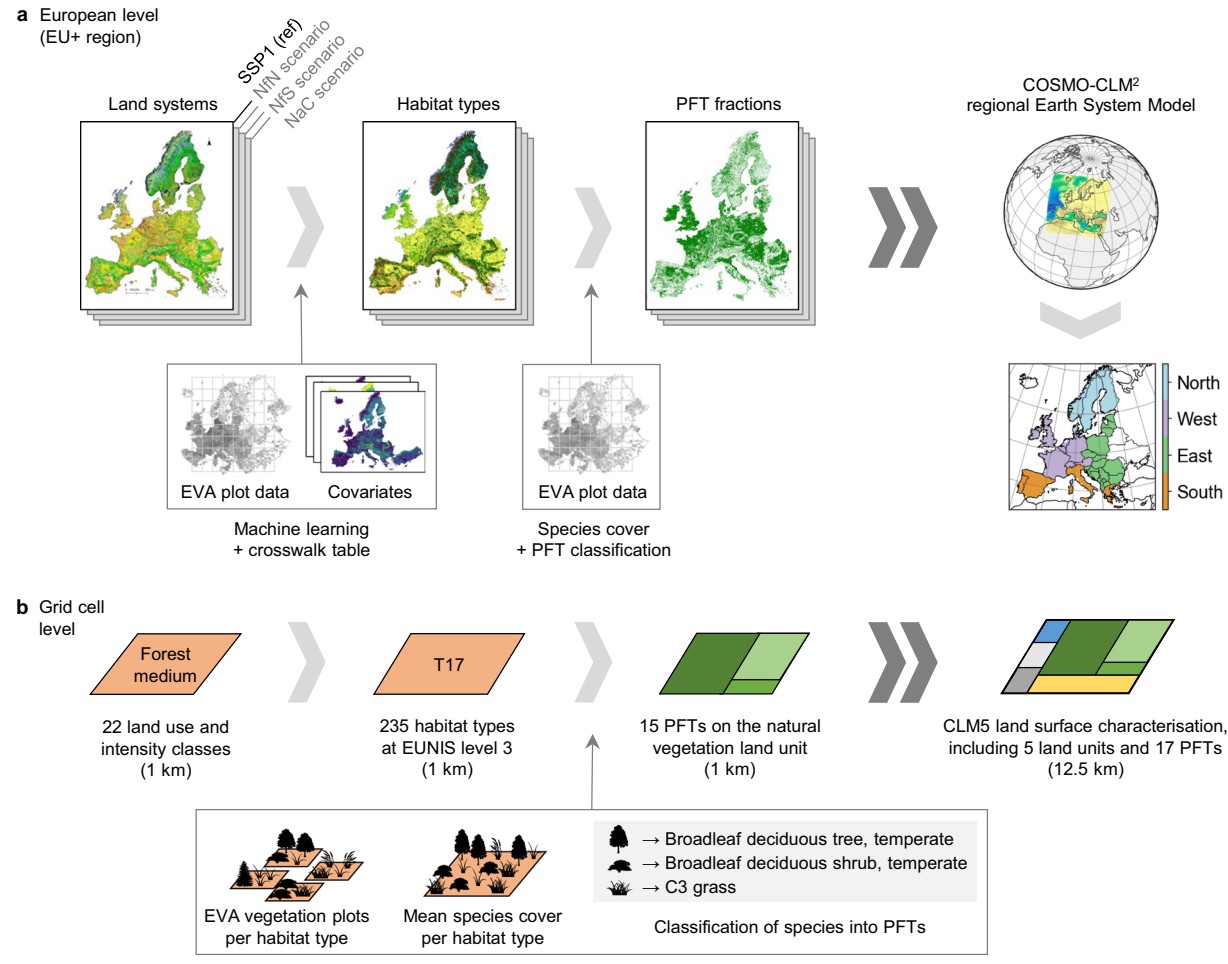

**a** European level (EU+ region)

Land systems | Habitat types | PFT fractions | COSMO-CLM² regional Earth System Model

EVA plot data | Covariates
Machine learning + crosswalk table

EVA plot data
Species cover + PFT classification

North
West
East
South

**b** Grid cell level

Forest medium | T17

22 land use and intensity classes (1 km) | 235 habitat types at EUNIS level 3 (1 km) | 15 PFTs on the natural vegetation land unit (1 km) | CLM5 land surface characterisation, including 5 land units and 17 PFTs (12.5 km)

EVA vegetation plots per habitat type | Mean species cover per habitat type

🌳 → Broadleaf deciduous tree, temperate
🌿 → Broadleaf deciduous shrub, temperate
🌾 → C3 grass

Classification of species into PFTs

Disaggregation of EUNIS level 3 habitat types into PFT fractions

**Fig. 2 | Workflow for integrating the land system scenarios in a regional Earth System Model. a** Land system maps for the scenarios Nature for Nature (NfN), Nature for Society (NfS), and Nature as Culture (NaC) and for the reference (SSP1) are translated into habitat types of the European Nature Information System (EUNIS) at level 3, using vegetation plot data of the European Vegetation Archive (EVA) and environmental covariates. Habitat types are disaggregated into plant functional types (PFTs), using species composition in EVA plots across Europe. The resulting PFT maps are used as input to the COSMO-CLM² regional Earth System Model. **b** Each 1 km grid cell contains one land system consisting of a land use type and intensity. Each land system is translated into one EUNIS level 3 habitat type, depending on the suitability of the grid cell. Habitat types are disaggregated into PFT fractions according to the mean species composition in EVA plots belonging to that habitat type. The resulting maps of PFT fractions are aggregated to 12.5 km resolution and used to modify the land cover composition of CLM5, the land component of COSMO-CLM² (i.e., the soil and vegetation-covered parts of grid cells, accounting for 95% of the area in the EU+ region). Other surfaces including urban areas, lakes, wetlands, and glaciers are not modified.

diversification (i.e., conversion to mosaic systems) of mid-intensity croplands and thus an overall cropland decline (Fig. 3a). Mosaic systems consist of various habitats including grassland, shrubland, forest, and man-made types (Supplementary Data 1), which in turn consist of various PFTs (Supplementary Data 2). As a result, we find reduced crop PFT cover compared with the reference and more strongly increased grass PFT cover than in the other scenarios (Fig. 3c).

All scenarios affect the PFT composition in over 85% of grid cells in the considered EU+ region. Most grid cells differ on a small area fraction (<5% in NfN, <6% in NfS, and <16% in NaC, measured by the median), while few differ completely (Supplementary Fig. 1). The climate forcing due to contrasting land cover is thus spatially extensive and modest in intensity over the EU+ region, but strong in greatly divergent grid cells and subregions. Most grid cells show differences in multiple PFTs due to the high-resolution (1 km) land system and habitat mapping and the subsequent disaggregation into PFTs. The PFT differences are spatially heterogeneous and opposing transitions can be observed across Europe (Fig. 3c, Supplementary Fig. 2). Only

the NaC scenario shows widespread gains in grass PFTs at the expense of crop PFTs.

NfN and NfS show comparatively weak PFT cover differences, with net transitions on 10.0% and 10.5% of the area, respectively, concentrated in the West and East subregions (Supplementary Table 1). NfN and NfS exhibit similar spatial patterns in dominant PFT transitions (Fig. 3b) and overall differences, with increased grass and crop PFT cover at the expense of tree and shrub PFTs (Fig. 3c). However, compared with NfN, NfS exhibits slightly higher crop PFT cover, lower grass PFT cover, and lower tree and shrub PFT cover (yet the promotion of forests and shrublands in areas with high aboveground carbon accumulation potential is in line with our simulated gross primary production (Supplementary Data 3)). The NaC scenario shows the most widespread PFT cover differences (21.0% of the area), dominated by the expansion of grass PFTs at the expense of crop PFTs in in the West, East, and South subregions (Fig. 3b, c). Since roughly one third of the crop PFT area is irrigated in Southern Europe, NaC causes a notable decline in irrigated crop PFT cover.

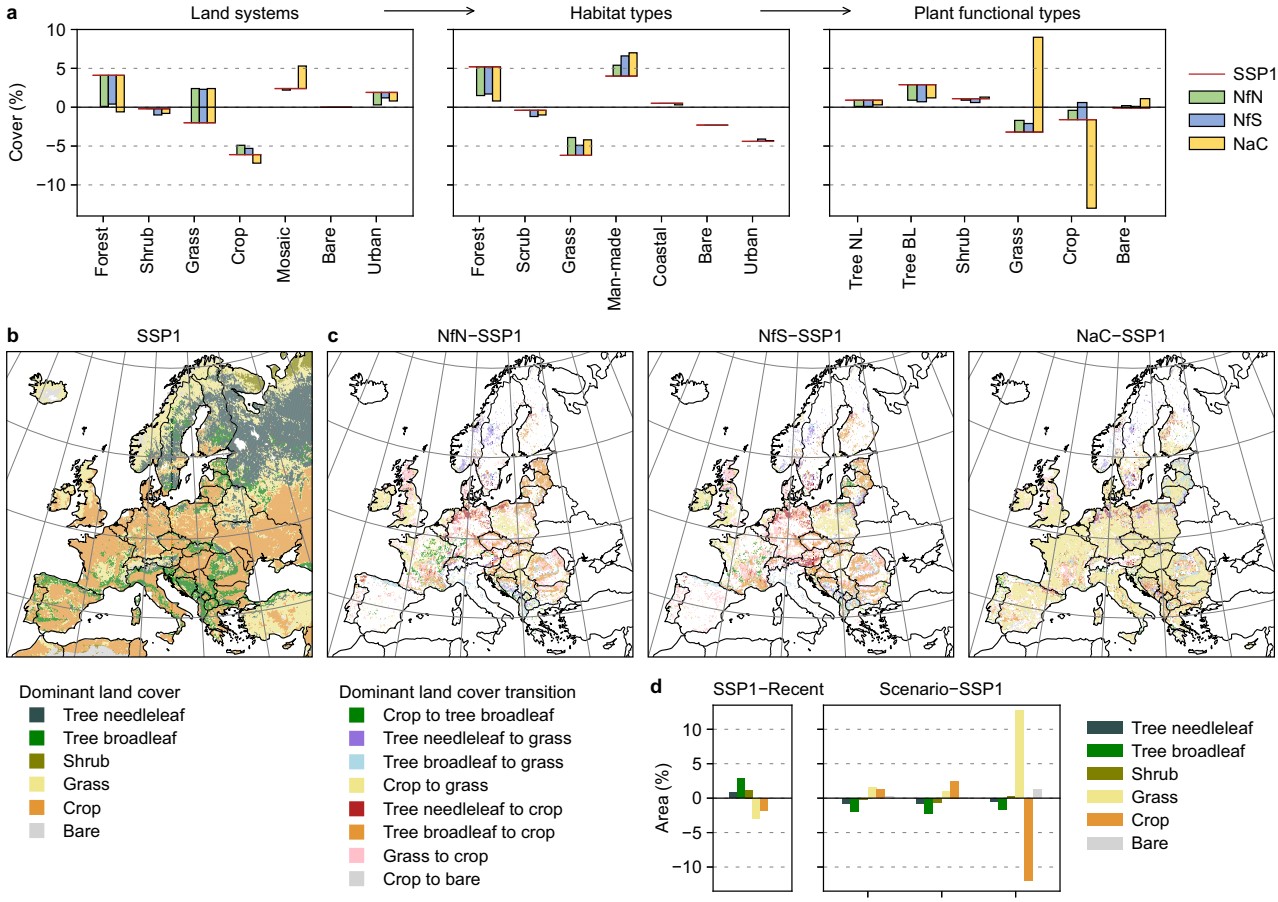

**Fig. 3 | Land cover in the scenarios compared with the reference and recent conditions. a** Land systems are translated into habitat types and plant functional types (PFTs). Differences are shown for the scenarios Nature for Nature (NfN), Nature for Society (NfS), and Nature as Culture (NaC) relative to the reference (SSP1, represented by the red baseline) and recent conditions (represented by the zero line). For instance, forest cover is lower in all scenarios than under SSP1 but higher under SSP1 than it is today. Only the NaC scenario would reduce forest cover relative to recent conditions. **b** Dominant land cover in SSP1 considering the soil and vegetation-covered parts of the grid cells. The land cover categories aggregate temperate and boreal PFT types, deciduous and evergreen tree PFTs, C3 and C4 grass PFTs (i.e., herbs excluding annual crops), and rainfed and irrigated crop PFTs (tropical variants are not present). **c** Dominant land cover transitions (i.e., main transitions in at least 5% of the grid cell area) in the scenarios relative to SSP1. Dominant transitions are determined based on maximum gains and losses per grid cell. The eight most common transitions across scenarios are shown. Supplementary Fig. 2 shows fractional changes of main PFT types. **d** Differences in land cover composition in the scenarios relative to SSP1 and in SSP1 relative to recent conditions in the EU+ region. Figure 5 shows a breakdown for European subregions.

## Effects on climate in a regional Earth System Model

Differences in the distribution of PFTs strongly influence temperature and precipitation at local to regional scales through biogeophysical processes[41,52–54]. Locally, PFT changes alter surface albedo, evaporative fraction, and roughness length (Supplementary Fig. 3), which in turn modify the surface energy balance and near-surface climate. The local effect of each PFT transition varies spatially[59,60] depending on local PFT properties (e.g., leaf area, canopy height) and background climate (e.g., radiation, temperature, water availability, snow cover). Spatially explicit assessments are thus needed. Biogeophysical processes affect the climate locally (e.g., directly through evaporative cooling, and indirectly through atmospheric feedbacks) but also non-locally through the advection of heat and moisture and altered atmospheric circulation[61,62]. While direct local effects can be estimated from observations, capturing atmospheric feedbacks and non-local effects requires coupled land-atmosphere simulations[63], like those performed here over Europe.

To assess the climate effects of the scenarios due to differences in PFTs by 2050, we analyse annual and seasonal averages (2036–2050 mean ± one standard deviation across years) of temperature, precipitation, soil moisture, wind speed, and vegetation carbon uptake

(i.e., gross primary production), and annual maximum temperature (TXx) as an extreme temperature index (Fig. 4). Results are summarised for the EU+ region, four European subregions, the 1% most affected areas per scenario, the 1% most extreme areas in the SSP1 reference (e.g., hottest areas), and the 1% most changing areas in SSP1 compared with recent climate (e.g., most heating areas) (Supplementary Data 3).

The NaC scenario causes widespread warming over Europe. Particularly in summer, near-surface (2 m) air temperature is $0.17 \pm 0.04\,°C$ higher than under the SSP1 reference (statistically significant at $P < 0.05$ unless otherwise stated). Consequently, air temperature would warm one third more under NaC than SSP1 by the middle of the century, compared with recent climate. Significant warming can be observed in the East ($0.25 \pm 0.08\,°C$), West ($0.24 \pm 0.06\,°C$), and South ($0.18 \pm 0.05\,°C$) subregions, which show strong PFT differences. The warming reaches $0.68 \pm 0.05\,°C$ in the most affected areas. The hottest areas under SSP1 would experience $0.23 \pm 0.08\,°C$ additional temperature rise under NaC, but the most heating areas would not get significantly warmer under NaC than SSP1. The warming effect is particularly strong during the hottest times of the years (TXx $+0.33 \pm 0.14\,°C$) and reaches $+1.86 \pm 0.29\,°C$ in the most

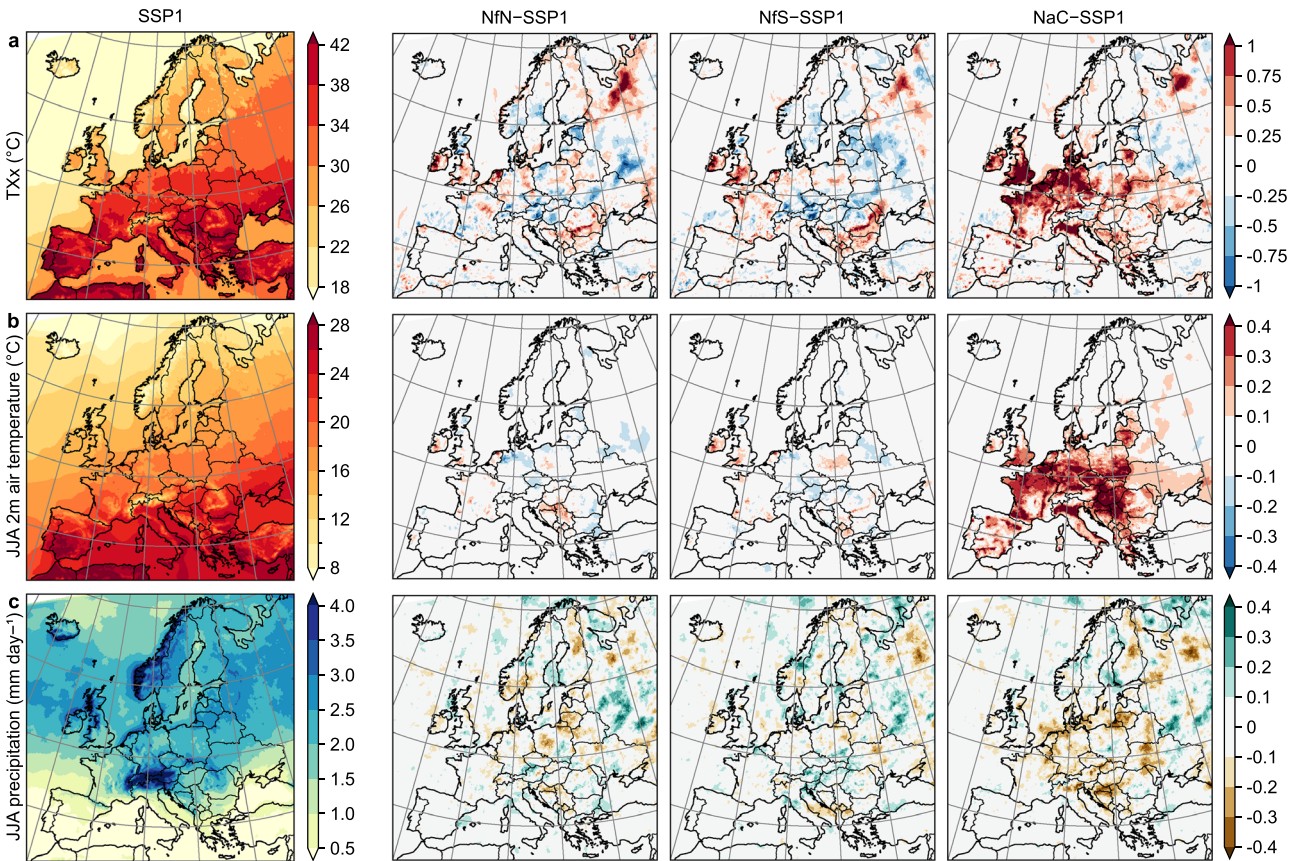

**Fig. 4 | Climate response to Nature Future scenarios.** Climate (2036–2050 mean) in the reference (SSP1, left) and responses in the scenarios Nature for Nature (NfN), Nature for Society (NfS), and Nature as Culture (NaC) relative to SSP1 (right). **a** Annual maximum temperature at 2 m (TXx). **b** Summer (JJA) temperature at 2 m. **c** Summer precipitation. Supplementary Fig. 4 shows summer soil temperature at 0–10 cm, summer wind speed at 10 m, summer soil moisture at 0–10 cm, and annual gross primary production.

affected areas. In contrast, the NfN and NfS scenarios show weaker and spatially more heterogeneous temperature responses (Fig. 4a, b) owing to weaker and more heterogeneous PFT differences. Temperatures do not differ significantly from those under SSP1 at the European scale. A small but robust cooling of summer temperatures can be observed in the most heating areas under NfN. Locally in the most affected grid cells, hot temperatures can increase by roughly 1 °C (TXx +0.98 ± 0.33 °C in NfN and +1.06 ± 0.50 °C in NfS), which is substantial but less than under NaC. Overall, 57.8% of European land would experience significantly different summer temperatures in the NaC scenario, but only 1.6% under NfN and 3.9% under NfS. The spatial extent of temperature differences is thus substantially larger than that of PFT differences under NaC, but much smaller under NfN and NfS.

The NaC scenario also shows the strongest effects on precipitation in summer, with decreases of 0.05 ± 0.03 mm day$^{-1}$ over the EU+ region, and similar declines over the East and West. Nevertheless, soil moisture would not decline at the regional and subregional scales thanks to reduced evapotranspiration (see also Supplementary Figs. 3, 4). The precipitation response is spatially heterogeneous (Fig. 4c) and not statistically significant at the grid cell level. Robust effects at the subregional scale might be explained by non-local effects due to atmospheric moisture transport and precipitation recycling at larger spatial scales[64,65].

Overall, the NfS scenario would be associated with no significant negative impacts in terms of warming, drying, or reduced ecosystem carbon uptake over Europe and with the least trade-offs in all subregions (Fig. 5). The NfN scenario may be preferable if the goal was to prevent further warming in the most heating areas or to reduce

summer temperatures in the South ($P = 0.055$), but it could lead to drier soils in the East and lower ecosystem carbon uptake in North, West, and South. For the NaC scenario, we find significant negative impacts in most regards (except soil moisture drying) over Europe and all subregions except for the North.

Our results show a clear relationship between grid cells' level of PFT perturbation and responses in TXx summer air temperature, and summer precipitation (Fig. 6), indicating strong control of the implemented PFT changes on local to regional climate. In the NaC scenario, stronger PFT changes are associated with greater temperature increases in all European subregions. For similar perturbation levels, summer temperature is more sensitive in the West, East, and South than in the North, while TXx is particularly sensitive in the West. This is likely caused by differences in PFT transitions, background climate including energy and water availability[66], and other influences such as large-scale circulation and Atlantic weather that can dampen or exacerbate the local effects of land cover changes[67]. Even for precipitation, our results show stronger declines over greatly modified grid cells in the West, East, and North, but not in the South. The NfN and NfS scenarios show weaker responses at the European and subregional scales due to compensatory effects between opposing PFT transitions.

**Drivers of the temperature response**

To investigate the causes of the simulated temperature responses, we focus on the summer season and decompose the mean (2036–2050) surface temperature differences into contributions of surface energy balance components (Methods). Surface temperature differences may

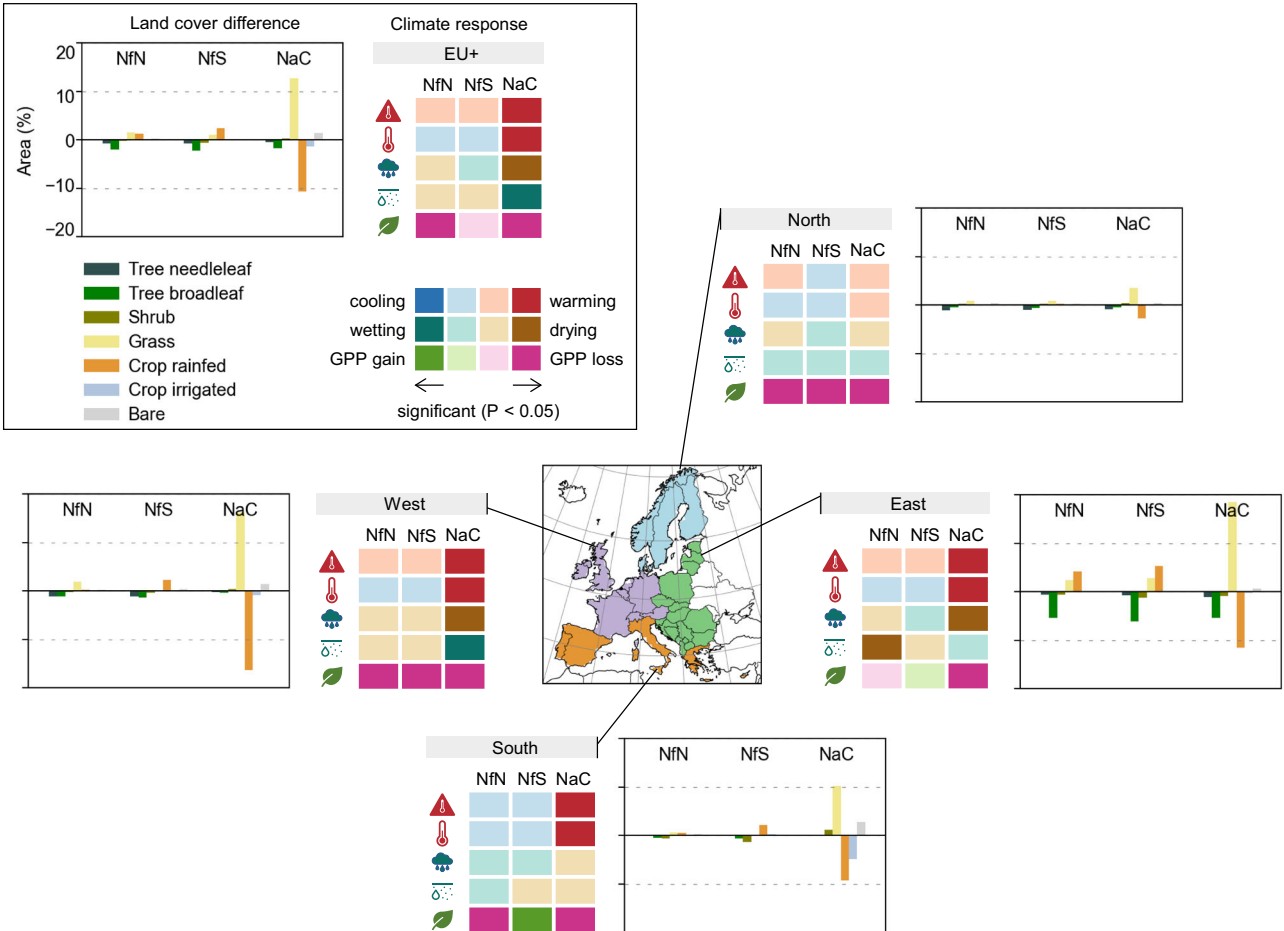

**Fig. 5 | Land cover differences linked to climate responses over Europe (EU+) and in subregions.** Each panel shows one region and differences in the scenarios Nature for Nature (NfN), Nature for Society (NfS), and Nature as Culture (NaC) to the reference (SSP1). The land cover categories aggregate temperate and boreal plant functional types (PFTs), deciduous and evergreen tree PFTs, and C3 and C4 grass PFTs (i.e., herbs excluding annual crops). The climate response is shown for annual maximum temperature at 2 m (TXx), summer temperature at 2 m, summer precipitation, summer soil moisture at 0–10 cm, and annual gross primary production. Colours indicate the direction of the response (e.g., blue for cooling, red for warming) and darker colours indicate statistical significance at the 95% confidence level.

be caused by changes at the land surface (with increases in latent heat flux, sensible heat flux, ground heat flux, and albedo cooling the surface temperature) or feedbacks from the overlying atmosphere (with increases in downwelling shortwave and longwave radiation warming the surface temperature). Surface (skin) temperature further influences the near-surface air and is thus a useful proxy for the 2 m air temperature response. Regional summaries are presented in Fig. 7 and maps in Supplementary Figs. 6–8.

The NaC scenario shows an important surface warming contribution from reduced latent heat flux (Fig. 7c). This reduction in latent heat flux is consistent with lower surface roughness due to lower canopy height, and with lower evaporative fraction due, e.g., to lower leaf area. The surface warming effect is partly compensated by increased sensible heat flux, ground heat flux, and albedo. The net surface warming is enhanced by strong atmospheric feedbacks (primarily increased downwelling shortwave radiation resulting from reduced humidity and cloudiness) at the European scale and in the West, East, and South subregions. The NfN and NfS scenarios show warming contributions from declines in both latent and sensible heat fluxes at the European scale and in most subregions, partly compensated by cooling from increased albedo (Fig. 7a, b). The net surface warming is outweighed by cooling from atmospheric feedbacks over EU+ and South, resulting in cooler surface temperatures. The South shows an important cooling contribution from reduced downwelling

longwave radiation resulting from reduced atmospheric emissivity and temperature.

These results suggest that the summer temperature differences between the scenarios and SSP1 are more strongly influenced by atmospheric feedbacks (predominantly changes in downwelling shortwave radiation) than surface effects, especially in the NaC scenario. Surface effects are dominated by reduced turbulent energy transfer to the lower atmosphere (i.e., through latent and sensible heat fluxes), which is in line with the reduced surface roughness in all scenarios and subregions. Nevertheless, the relative importance of drivers varies spatially and seasonally, depending on the pattern of PFT changes, local and seasonal PFT properties, and environmental conditions. For instance, the most warming areas show surface warming and reinforcing atmospheric feedbacks of similar magnitude. In winter, surface effects are dominated by albedo changes, and atmospheric feedbacks by changes in downwelling longwave radiation (not shown).

## Contribution of individual land cover transitions

To assess which PFT changes are important for the temperature response in each scenario, we focus again on the mean (2036–2050) summer temperature and estimate the contributions of individual land cover transitions. After computing net transitions between seven land cover categories per grid cell (i.e., tree needleleaf, tree broadleaf, shrub, grass, crop rainfed, crop irrigated, and bare, with PFT variants

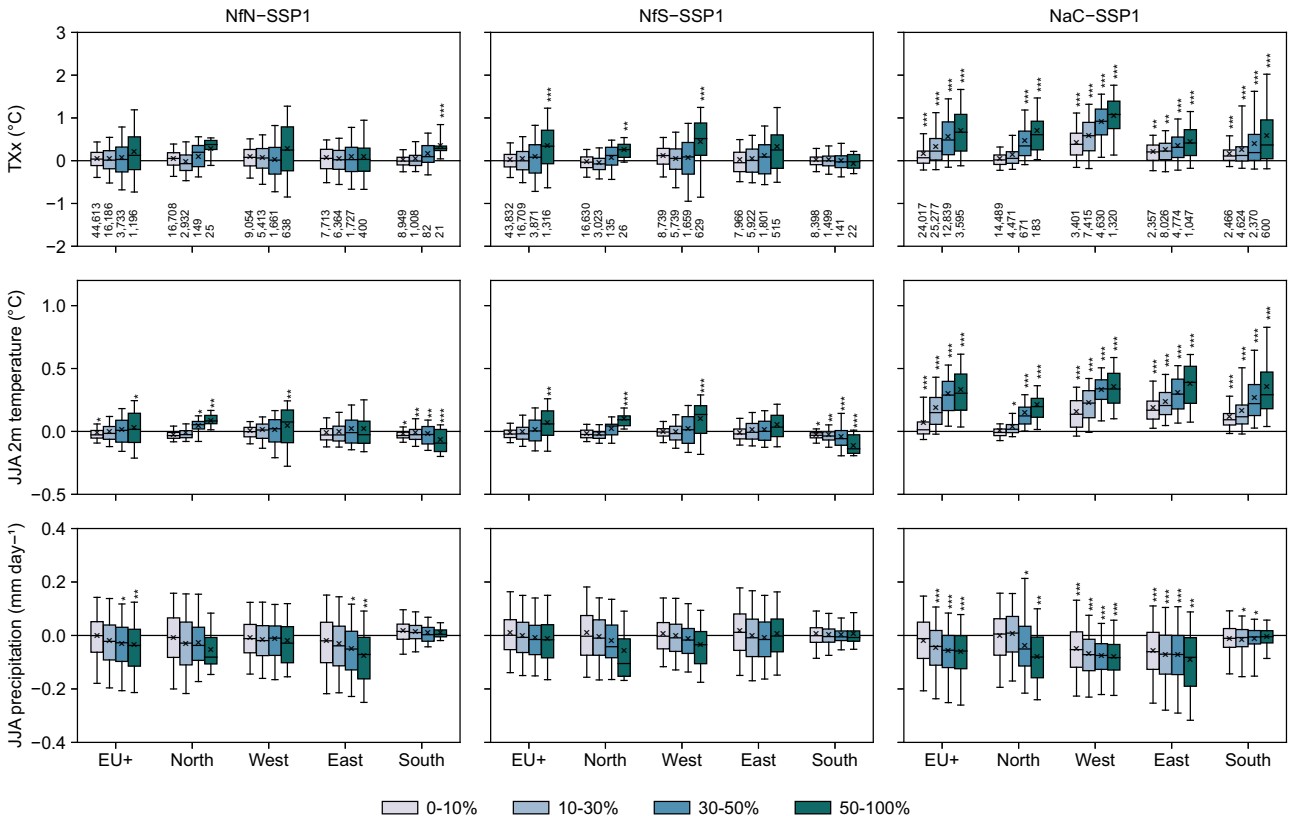

**Fig. 6 | Climate sensitivity to changes in plant functional types over Europe (EU +) and in subregions.** For the scenarios Nature for Nature (NfN), Nature for Society (NfS), and Nature as Culture (NaC) relative to the reference (SSP1), responses in the 2036–2050 mean annual maximum temperature at 2 m (TXx), summer (JJA) temperature at 2 m, and summer precipitation at the grid cell level are grouped by the proportion of changes in plant functional types (0–10%, 10–30%, 30–50%, or 50–100% of the grid cell). Numbers in the bottom row indicate the number of grid cells included per group to generate the boxplot. Boxes indicate the first quantile, the third quantile, and the median of grid cell level data, and whiskers extend from the 5th to the 95th percentile. Crosses indicate the area-weighted mean. Statistical significance is based on two-sided Wilcoxon signed-rank tests, with p-values adjusted for multiple comparisons across groups and regions (*$P < 0.1$, **$P < 0.05$, ***$P < 0.01$). Supplementary Fig. 5 shows summer soil temperature at 0–10 cm, summer wind speed at 10 m, summer soil moisture at 0–10 cm, and annual gross primary production.

aggregated like in Fig. 5), we fit a linear regression between the temperature response and 20 different net transition types across grid cells (Methods). This allows us to extract the temperature sensitivity to individual land cover transitions at the grid cell scale (i.e., local effects) from scenario simulations in which multiple PFTs are changed simultaneously. Conceptually similar approaches have been used to extract land cover change signals from simulations with multiple forcings[41,68].

At the European scale, the NfN and NfS scenarios show important contributions to cooler summer temperatures from transitions of needleleaf tree and shrub PFTs to rainfed crop and grass PFTs (Fig. 8a, b). Warming contributions are linked to transitions of rainfed crop to grass PFTs and bare soil, and broadleaf tree to grass PFTs. The NaC scenario shows similar effects, but the temperature response is dominated by a much stronger warming contribution from transitions of crop to grass PFTs (Fig. 8c). Differences between scenarios and subregions result primarily from the scale of individual transitions. For example, in the NaC scenario in the South transitions of irrigated crop to grass PFTs are associated with substantial warming, whereas in the other scenarios and subregions irrigated crops are negligible. The estimated temperature sensitivity (i.e., the temperature response per percent transition) is broadly consistent across scenarios. In all scenarios, broadleaf tree and irrigated crop PFTs are associated with the strongest cooling, followed by rainfed crop and grass PFTs, and finally needleleaf tree and shrub PFTs, which only show cooling effects in transitions from bare soil (Supplementary Fig. 10).

Regressions performed at the subregional scale can result in varying sensitivity estimates, particularly for transitions between tree and non-tree PFTs. For instance, transitions of broadleaf tree to rainfed crop PFTs are linked with cooling in the North and warming in the South in all scenarios, and with diverging and often uncertain effects in the West and East (Supplementary Figs. 11–14). This might be due to differences incoming shortwave radiation and water availability between North and South and within the West and East subregions. Such differences in background climate influence the relative importance of changes in albedo vs. evapotranspiration efficiency for temperature impacts, which also explains latitudinal differences in the response to deforestation (i.e., cooling in high latitudes and warming in the tropics)[69]. Although we focus on near-surface air temperature, we note that temperature sensitivities and contributions of individual land cover transitions are similar in relative terms for surface (skin) temperature but associated with lower uncertainty (Supplementary Fig. 15).

## Discussion

### Different implementations of European climate and biodiversity policies have important consequences for the distribution of plant functional types

Substantial transformations in land systems are needed to meet the projected demands under the SSP1 storyline and to achieve additional policy targets that help alleviate the climate and biodiversity crises. Existing policies for climate and biodiversity protection could lead to

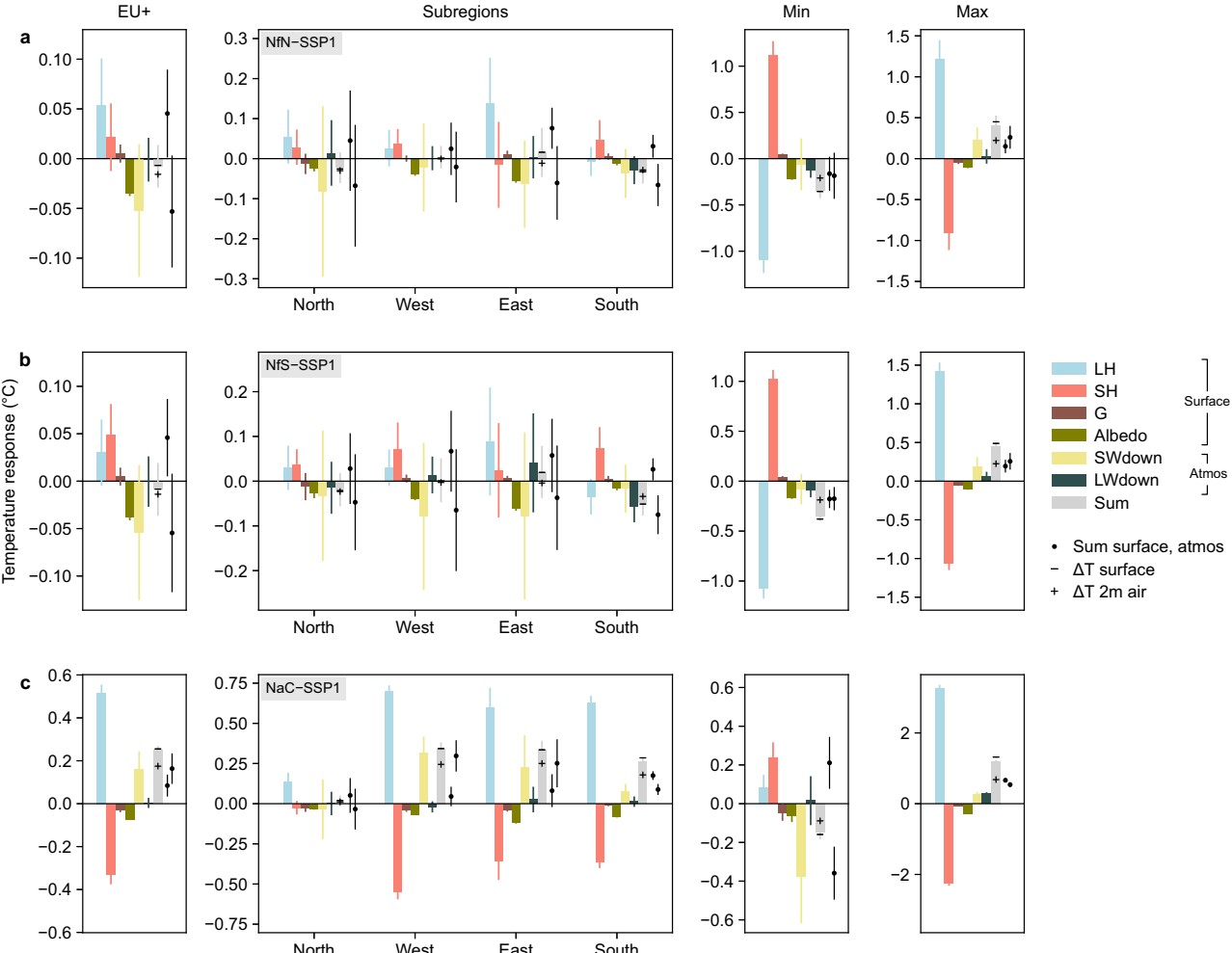

**Fig. 7 | Contributions of surface energy balance components to the surface temperature response in summer.** Results are shown for Europe, the subregions, and the 1% most cooling (min) or warming (max) areas in terms of near-surface (2 m) air temperature, for **a** the Nature for Nature (NfN) scenario, **b** the Nature for Society (NfS) scenario, and **c** the Nature as Culture (NaC) scenario. Contributions at the land surface include latent heat flux (LH), sensible heat flux (SH), ground heat flux (G), and albedo, whereas those of atmospheric feedbacks are given by downwelling shortwave radiation (SWdown) and downwelling longwave radiation (LWdown). The sum of surface and atmospheric contributions are shown as dots. The grey bar indicates the sum of all contributions, which is similar to the simulated surface (skin) temperature response (−) and linked to the near-surface (2 m) air temperature response (+). Bars and markers show the 2036–2050 mean and error bars extend to the 95% confidence interval across years. Group sizes in number of grid cells are 65,849 (EU+), 19,876 (North), 16,768 (West), 16,221 (East), 10,060 (South), and ~6500 (1% most cooling or warming areas). Supplementary Figs. 6–8 show maps of the components for each scenario.

land system changes on almost 30% of European land in 2050, and on 20% of European land the transitions would differ depending on whether for the implementation focuses on intrinsic, instrumental, or relational values following the Nature Futures Framework[33]. By modelling effects on the distribution of PFTs, we find that existing climate and biodiversity policies could lead to different PFT cover on 10.0% of European land (49 Mha) in an NfN scenario focusing on biodiversity conservation and land sparing, 10.5% (51 Mha) in an NfS scenario promoting climate change mitigation and land sharing, and 21.0% (103 Mha) in an NaC scenario prioritising cultural landscapes and human-nature interactions. The scenarios pursue nature protection and restoration in alternative ways, but differences in PFT cover are also influenced by land use intensity (e.g., extensification in NfN, intensification in NfS), multi-functionality (e.g., diversification in NaC), and compensatory effects to ensure that multiple demands can be met.

### Climate effects are strongest locally but also relevant on subregional and regional scales

By performing ESM simulations, we demonstrate that PFT cover differences on such a scale can have profound effects on land surface characteristics and thus on local to regional climate. In the 1% most affected areas, we find seasonal differences between the scenarios and the SSP1 reference in temperature of up to 0.68 ± 0.05 °C warming in summer in NaC and 0.29 ± 0.20 °C cooling in spring in NfN, and in precipitation of up to +0.32 ± 0.31 mm day⁻¹ in autumn in NfN and −0.35 ± 0.10 mm day⁻¹ in summer in NaC. These differences are comparable in magnitude with those found in extreme land cover change experiments over Europe[36,70]. Effects at the European and subregional scales are weaker and comparable with those of recent historical land cover changes in 1992–2015, affecting 70 Mha[41]. This can be explained by the limited area converted and compensatory effects between opposing transitions. Nevertheless, 57.8% of European land would experience significantly different summer temperatures under the NaC scenario, compared with only 1.6% under NfN and 3.9% under NfS. Effects on summer precipitation are small and variable (both temporally and spatially) and are thus not statistically significant in individual grid cells. However, precipitation declines significantly at the regional scale in the NaC scenario, likely in response to a large-scale reduction in evaporative fraction[71].

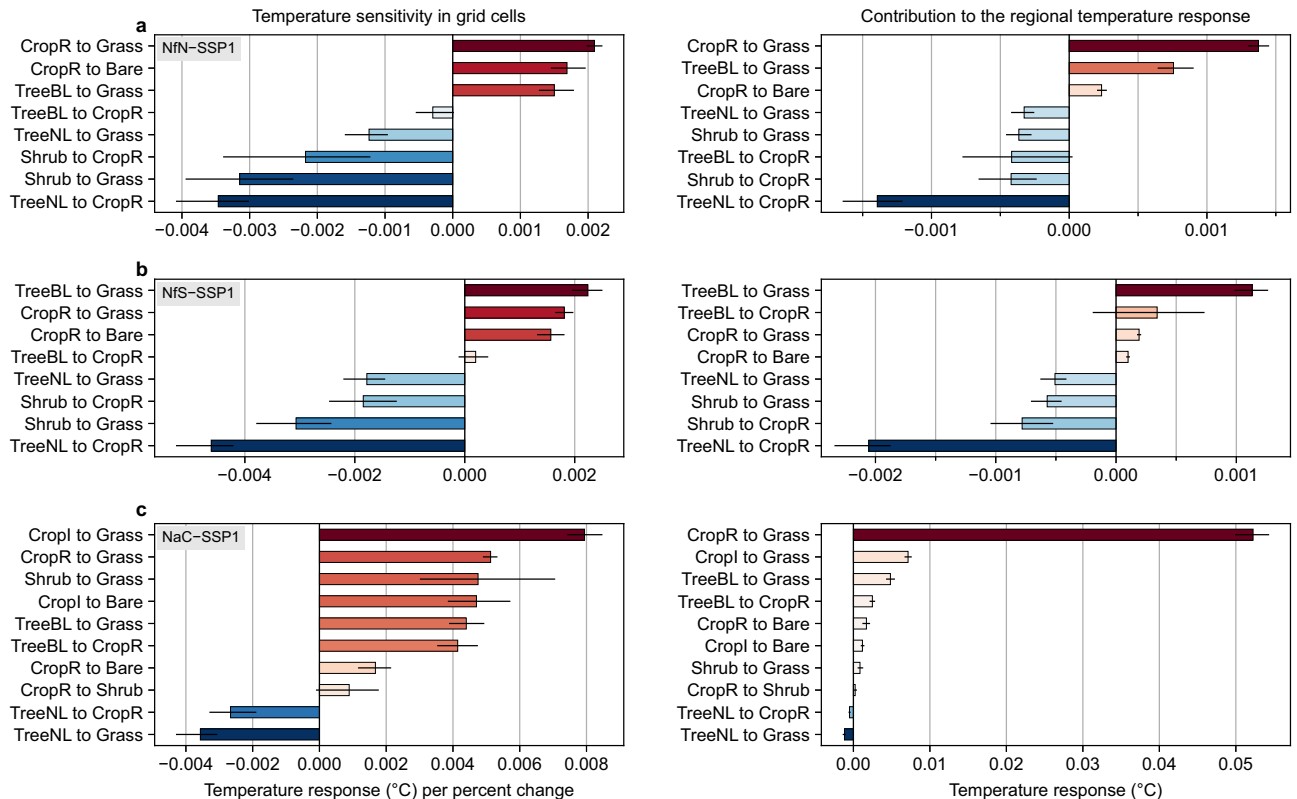

**Fig. 8 | Contribution of individual land cover transitions to the 2 m air temperature response in summer over Europe.** Regressions are performed on the 2036–2050 mean temperature response. Sensitivities are given by the unstandardised regression coefficients and contributions by the sensitivities multiplied by the percentage change over the region. Results are shown for **a** the Nature for Nature (NfN) scenario, **b** the Nature for Society (NfS) scenario, and **c** the Nature as Culture (NaC) scenario. For each scenario, land cover transitions are shown that occur on >0.2% of the area and in >20% of grid cells. Land cover transitions are named by the dominant direction per scenario (e.g., shrub to crop rainfed in NfN, crop rainfed to shrub in NaC) and sorted by strength of effect. Bars show the mean across 200 linear regression fits and error bars extend from the 2.5th to the 97.5th percentile of the bootstrap sample. Results for subregions are shown in Supplementary Figs. 11–14 and results for surface skin temperature in Supplementary Fig. 15.

## Turbulent fluxes and atmospheric feedbacks drive the surface temperature response in summer

Our results suggest that the temperature response in summer is strongly influenced by changes in turbulent fluxes at the surface and atmospheric feedbacks. In the NfN and NfS scenarios warming from reduced latent and/or sensible heat flux is outweighed by cooling from atmospheric feedbacks, resulting in cooler surface temperatures. In contrast, the NaC scenario shows warmer surface temperature because the net warming from reduced turbulent fluxes is reinforced by atmospheric feedbacks. The strong influence of atmospheric feedbacks could have important implications for regional land use management, because atmospheric feedbacks can affect nearby or remote areas (i.e., non-local effects)[61,62].

Surface temperature decomposition is a robust physics-based approach, but we note some caveats that should be considered in the interpretation of the results. First, the considered surface energy balance components are not independent due to process coupling (e.g., energy availability for latent and sensible heat flux is influenced by downwelling shortwave radiation and albedo) and the underlying biogeophysical mechanisms cannot be separated. Alternative methods focus on disentangling the effects of changes in surface albedo, evaporative fraction, and roughness length, but assume no atmospheric feedbacks[72,73], independent effects[74], or linear effects[36]. Second, surface temperature and near-surface air temperatures are tightly coupled and their responses correlated ($r > 0.8$ in more than 90% of grid cells, with lower correlations over rough terrain, see Supplementary Fig. 9). However, air temperature responses are typically weaker (see Fig. 7 and refs. 53,75–77) and influenced more strongly by atmospheric

processes. This might explain the different sign of surface and near-surface air temperature response in the East subregion in NfN and NfS.

## Two land cover transition types contribute most strongly to cooler and warmer summer temperatures

Regressions over Europe reveal that the small cooling in NfN and NfS is mostly explained by transitions of needleleaf tree to crop PFTs, due to high temperature sensitivity and high spatial coverage particularly in the West. This cooling effect could be due to higher albedo and evapotranspiration of crop PFTs during the growing season in our model, but there is contradictory evidence from models and observations on the summer temperature difference between needleleaf forests and croplands[53,59,60,70,78]. Nevertheless, the estimated cooling effect is consistent in our regression models across scenarios, most subregions, and for both temperature variables (i.e., near-surface air and surface skin temperature).

The strong warming in NaC is mostly explained by widespread transitions of crop to grass PFTs, which dominates the temperature response in all subregions. Based on spatial patterns and simple correlations, we speculate that the warming is due to lower evaporative cooling of grass PFTs in the model, which in turn could result from lower leaf area index and vegetation height, and from the lack of irrigation in the South. The simulated effects agree with observational evidence of differences between cropland and grassland, showing that non-radiative mechanisms dominate and that temperatures are warmer over grasslands than rainfed and irrigated croplands in summer (and other seasons except winter) over Europe[59] and the US[53]. However, results from another regional climate model indicated spatial

variability and uncertainty about the sign of the annual mean temperature response to transitions between cropland and grassland[41]. Since model benchmarking and evaluation has mostly focused on forest cover change[34,36,79] (and more recently also on cropland expansion and irrigation[64,80,81]), more research is needed to corroborate the signal in ESMs and to represent the variety of cropland and grassland types. The ongoing second phase of the CORDEX Flagship Pilot Study LUCAS systematically investigates inter-model differences in the response to realistic land use changes across Europe and could provide new insights in this context.

### Nature management focusing on cultural services could cause additional challenges for regional climate adaptation and mitigation

Our results show that the NaC scenario would have the strongest detrimental effects on climate. Higher annual maximum temperatures, higher summer temperatures, and lower summer precipitation under NaC than SSP1 indicate trade-offs with local climate impacts and the resulting need for adaptation. The European continent has warmed by more than 2 °C since the pre-industrial period and is expected to further warm in the coming decades[82]. Even small increases in temperature can increase the intensity and frequency of extreme weather and affect ecosystems and human health[5,83,84]. An additional warming and drying could thus further threaten local biodiversity and human well-being, particularly in hot spot areas for heat and drought in Southern Europe but also in the West and East. A detailed assessment of losses in species climatic range size[9,85] or health impacts[83,86] was, however, outside the scope of this study. Reduced vegetation carbon uptake ($-11 \pm 0.4$ %) suggests lower potential for ecosystem-based mitigation. However, dedicated simulations with interactive biogeochemistry would be needed to quantify the net response of the land carbon cycle (i.e., additionally considering respiration, disturbances, and soil carbon dynamics) and to represent the underlying biogeochemical processes. In contrast, the NfS scenario shows no significant negative impacts in terms of warming, drying, or reduced ecosystem carbon uptake over Europe and the least trade-offs in all subregions. The NfN scenario could result in lower vegetation carbon uptake but would cool summer temperatures in the most warming areas. Trade-offs with climate adaptation and mitigation are thus low, although local compromises exist in all scenarios. Based on these findings, new scenarios could be developed that promote biodiversity while helping stabilise the climate.

### Advances and limitations regarding the inclusion of Nature Future scenarios in Earth System Models

This study demonstrates an approach to represent nature management guided by different value perspectives in a regional ESM, with the aim to explore potential biogeophysical effects on the regional climate. While land system allocation, habitat suitability, and species composition are addressed with dedicated models, we use the ESM to study climate processes, sensitivities, and feedbacks. Our method thus includes comprehensive normative, social, and ecological scenarios as well as intricate interactions between land and atmosphere, without the ESM itself simulating social-ecological complexity. Although additional evaluation and model benchmarking is needed, the translation of land systems into PFTs based on spatially explicit habitat and species information could be an attractive alternative to spatially uniform crosswalk rules commonly used to generate new PFT maps from land cover observations[87,88].

Our implementation based on existing PFTs captures important land use-driven forcings, but it does not address all characteristics of the land system scenarios (e.g., land use intensity in terms of fertiliser input, irrigation, and wood production[89]) and the corresponding habitat predictions (e.g., functional diversity[90]). Yet, for regional to global scale ESMs, identifying and parameterising an optimal set of PFTs is still challenging and alternative representations of plant diversity are not mature[91–93]. A future version of this analysis could attempt to include land use intensity and plant diversity by, e.g., introducing new PFTs, geographically varying PFT parameters, or management/habitat-specific vegetation structure (i.e., height and leaf area index). Specific attention could be given to the representation of mosaic land systems, which are an important diversification component in the SSP1 storyline and account for ~20% of vegetated land in the land system scenarios of Dou et al.[33]. Ideally, the requirements and capabilities of ESMs should be considered during scenario design and modelling by, e.g., coordinating the definitions of land systems and PFTs, or evaluating scenario robustness with respect to PFT composition.

We acknowledge simplifications in our model setup, such as the lack of interannual vegetation development as we model a new land use state and not a transition over time. Furthermore, prescribing leaf area index and canopy height based on observations reduces uncertainties related to the reliable prediction of vegetation structure but omits certain aspects of vegetation-climate coupling. Specifically, this approach does not capture the effects of interannual variability and climate change on vegetation growth and phenology and their consequences for biogeophysical processes and carbon uptake. Further uncertainty results from the reliance on boundary conditions from one global climate model that performs well over Europe, MPI-ESM1-2-HR. A single model cannot represent the range of plausible outcomes under the SSP1-2.6 scenario, so we cannot assess the sensitivity of the climate response to the large-scale background climate simulated by different global climate models (see Supplementary Note 4 for a comparison of candidate CMIP6 models). This aspect is still less explored than the substantial uncertainty resulting from the representation of land use and land cover change processes in different regional and global climate models[36,40] (but see refs. [42,94]).

### Advances and limitations regarding land use scenario modelling

Land-based measures to achieve climate and biodiversity goals will be a major challenge in the future. In fact, agreed sustainability targets in Europe will only be feasible if consumption is reduced sufficiently, as assumed under the SSP1 scenario, and if land systems are transformed on a large scale[33]. Various maps have been produced to identify priority areas for conservation, but the identified sites vary widely depending on the methods used to prioritise biodiversity[95], and the inclusion of additional factors such as nature's contributions to people[31,96,97], climate change[98,99], and land use change[100]. The land system scenarios employed here consider multiple measures for nature management jointly with different value perspectives (based on the Nature Futures Framework), projected future climate (based on RCP2.6) and human demands (based on SSP1), which are important steps to move from priority maps towards integrative scenarios[27,32]. Nevertheless, several indicators would be needed to capture the values and stakeholder views associated with each perspective of the Nature Futures Framework[33] (e.g., based on existing data[24,97,101] NfS could be represented by additional indicators for flood control, air and water quality regulation, pollination and biogeophysical climate regulation; and NaC could be represented by additional indicators for heritage forests, wild foods, access to nature, and nature tourism). Given the breadth of the Nature Futures Framework, its operationalisation can vary greatly in land use scenario modelling[45] and to date, only few studies have developed normative nature-positive storylines[102]. Further steps have recently been made in a new land system elaboration[103]. Future implementations of the Nature Futures Framework could represent each value perspective by multiple indicators. Moreover, scenarios could be developed that integrate several value perspectives (e.g., considering synergies among value perspectives[33,104] and their coexistence in society[45]). However, the development of increasingly complex scenarios will make it more

difficult to disentangle drivers and understand their effects. Therefore, distinct and diverse scenarios have advantages for exploring the regional climate sensitivity to possible futures, like in this study.

The land system scenarios employed in this study provide a positive view of the future, in which European climate and biodiversity policies are implemented. The underlying modelling technique of Dou et al.[33] imposes the same commodity demands and policy targets in all three scenarios, but the land system changes required to fulfil them differ depending on the priorities for nature management (i.e., biodiversity conservation, climate change mitigation, or cultural services). The scenario outcomes reflect different processes across Europe, which can be opposing due to contextual differences or compensatory effects. Particularly at the local scale, differences between scenarios can be difficult to interpret or seem counterintuitive. The results are path-dependent and complex, and the link to the initial scenario assumptions is not always evident (e.g., NfN requires more cropland than SSP1 to compensate for extensification, NfS can afford less forest expansion than SSP1 so carbon accumulation must be achieved by intensification and strategic placement of forests in productive areas). Such emergent outcomes allow a realistic representation of how land systems may change across Europe and deviate from overall non-spatial policy targets and visions. Considering the implemented land system allocation rules, it is unlikely that the scenarios outcomes go against their premises. However, the scenarios are driven by policy means and not their outcomes, so they are not necessarily efficient at achieving policy goals (e.g., NfS promotes forests in areas with high carbon sequestration potential but does not maximise carbon sequestration). An in-depth evaluation is outside the scope of this study and the reader is referred to Dou et al.[33] for details on the scenarios and their limitations.

The scenarios employed here reflect plausible developments under given scenario assumptions, and our ESM simulations offer a comprehensive and spatially explicit assessment of consequences for climate regulation on local to regional scales. Additionally, other modelling techniques can help inform landscape planning in Europe. For instance, spatial optimisation has been used to simulate where restoration, conservation, and production could be allocated to maximise biodiversity and carbon benefits while meeting other future land demand[105]. Similar approaches can be applied to co-optimise the delivery of various ecosystem services (see, e.g., refs. 106,107). Integrating diverse nature values (i.e., intrinsic, instrumental, and relational values) and valuation methods (i.e., ways to quantify relevant indicators) could improve decision-making towards just and sustainable futures[108].

## Summary and implications

This study assesses potential regional climate effects of scenarios consistent with the Kunming-Montreal Global Biodiversity Framework and the IPBES Nature Futures Framework, which describe alternative pathways towards the vision of "living in harmony with nature by 2050". Considering detailed information on habitats, species composition, and climate processes in a regional ESM, we show that nature management focusing on cultural landscapes and human-nature interactions (NaC scenario) could lead to further temperature increase in Europe. In contrast, focusing on biodiversity conservation and land sparing (NfN scenario) or climate change mitigation and land sharing (NfS scenario) would not cause additional challenges for regional climate adaptation. We conclude that existing policies for climate and biodiversity can be implemented in quite different ways, with substantially different outcomes and inherent trade-offs: not only because of the initial goals, but also because of subsequent feedbacks from land to the atmosphere. Despite uncertainty about future policy implementation and related land use changes, our results highlight the importance of quantifying potential climate effects of scenarios that transform the land surface on a large scale. Crucially, measures to

address the human-induced climate and biodiversity crises do not necessarily lead to improvements in local to regional climate and adaptation to climate change, as shown previously, e.g., for the climate effects of forestation[24,54,109]. These findings call for nuanced and targeted policy implementation in Europe to navigate trade-offs and optimise overall outcomes. Such optimised policy implementation could benefit from elements of all three scenarios. To support policy development and spatial planning, further research is needed to develop joint climate-stabilising and biodiversity-positive scenarios.

## Methods

### Integrative land system scenarios

We build on four integrative land system scenarios for Europe in 2050, which were generated using the spatially explicit land systems allocation model CLUMondo[33]. All four scenarios comply with the projected population, demands (e.g., for housing, crops, livestock, and wood), and climate under the SSP1 "Taking the Green Road" storyline paired with the greenhouse gas forcing of RCP2.6. The SSP1 storyline for the land use sector includes strong environmental regulation (e.g., reduced deforestation and emissions from land use), increased agricultural productivity, and little growth in food demand including low-meat diets[43,44]. These conditions are reflected in CLUMondo by taking outputs of the Integrated Assessment Model GLOBIOM for SSP1 as an input. Due to changes in sectoral demands and trade simulated by GLOBIOM, land use projections for Europe under SSP1 by 2050 indicate a strong decline of pastures and annual crops, which enables the pervasive expansion of forests and (semi-)natural vegetation[33,43].

The reference in this study (SSP1) reflects how projected demands for agricultural commodities and urbanisation according to GLOBIOM would lead to land system change from 2015 to 2050 according to CLUMondo, based on land system suitability estimated from historic land system patterns. Besides the sustainability assumptions embedded in the SSP1 storyline in terms of demand, production, and trade, the reference does not represent specific climate and biodiversity policies in Europe. The three other scenarios additionally implement targets of the Kunming-Montreal Global Biodiversity Framework and the European Union's Green Deal related to nature conservation (30% of the land area protected by 2030), restoration (1% annual increase in natural areas 2030–2050), increased tree cover (3 billion new trees by 2030, followed by an additional 2 billion new trees by 2050), reduced fertiliser application in agriculture (−20% by 2030, followed by an additional −10% by 2050), and urban planning (improved access to green spaces) (Fig. 1). In each of these scenarios, the land system allocation procedures are modified to represent these targets and one value perspective of the Nature Futures Framework. Policies are implemented through changes in (1) demands (e.g., increased demand for natural ecosystems and trees); (2) supply of goods and services (e.g., increased tree cover in urban systems); (3) land system conversion order and resistance (e.g., to prioritise high-intensity cropland for crop production or to protect natural ecosystems); and (4) spatial weights (e.g., to prioritise areas for protection). CLUMondo models were parameterised and run independently for four European sub-regions (North, West, East, South). Together, the resulting land system maps cover the European Union, the United Kingdom, Norway, Switzerland, and the Western Balkans at 1 km resolution.

### Translation of land systems into plant functional types

We implement the three scenarios (NfN, NfS, and NaC) and the SSP1 reference in the COSMO-CLM[2] regional ESM. Because CLM5 represents land use through the spatial distribution of PFTs, we translate the land system scenarios into maps of PFT coverage and prescribe these instead (Fig. 2). First, we build on a recent effort to model thematic habitats at 1 km resolution across Europe, following the four land system scenarios and climate under RCP2.6[57]. The habitat predictions are generated using ensemble multi-class models (Supplementary

Note 1 and Supplementary Fig. 16), which link EUNIS level 3 habitat types to environmental variables (Supplementary Table 2). These models are used together with crosswalk rules aligned with the land system types (Supplementary Data 1) to generate wall-to-wall habitat maps at EUNIS level 3.

Second, we disaggregate the EUNIS level 3 habitat types into PFTs. For this purpose, we use the species cover and habitat classification[110,111] at EUNIS level 3 from 819,232 vegetation plots of the European Vegetation Archive (EVA)[58] (Supplementary Note 2 and Supplementary Fig. 17). By assigning each plant species to a PFT of CLM5 and averaging across plots, we obtain the average cover of individual PFTs for 235 habitat types present in our dataset (Supplementary Data 2). These data form the basis for updating the PFT fractions of CLM5. However, in CLM5 fractions of (semi-)natural vegetation PFTs must sum to 100% and only the highest vegetation layer is considered for flux calculations[112]. Therefore, we classify the PFTs into vertical vegetation layers and adjust their fractions proportionally. The adjusted PFT fractions per habitat type are then used to replace the predicted habitats, before aggregating the layers to the 0.1° grid of CLM5. At the hierarchical level of CLM5 land units, the relative cover of crops and natural vegetation is modified based on the EUNIS habitat types for cropland (224–236) and semi-natural vegetation (all other types except ice (218) and urban (223)). Changes in ice and urban areas are not considered.

## Regional Earth System Model simulations

The COSMO-CLM² regional ESM includes the non-hydrostatic limited-area atmospheric model COSMO in Climate Mode (COSMO-CLM) and the Community Land Model (CLM), using OASIS3-MCT as a coupler. The model was developed to simulate land-atmosphere exchanges at high resolution, while capturing the complexity of physical, chemical, and biological processes that govern terrestrial states and fluxes[55]. CLM includes detailed representations of sub-grid (i.e., within grid cells) surface heterogeneity, vegetation physiology, and soil hydrology. It is thus well suited for investigating the effects of land use and land cover changes on climate across spatial and temporal scales[113]. In CLM each grid cell can be composed of up to five different land units (i.e., (semi-)natural vegetation, crop, urban, lake, and glacier), and the natural vegetation and crop land units can be composed of several different PFTs. Processes are simulated independently for each PFT, but the atmosphere is assumed horizontally homogeneous within each grid cell[112].

COSMO-CLM² over the European domain of the Coordinated Regional Climate Downscaling Experiment (EURO-CORDEX)[114] has been continuously validated against observations[55,115,116] and benchmarked against other regional and global climate models[36,117]. Here we employ an updated version of COSMO-CLM², including the COSMO-Model 6[118], CLM5[112], and OASIS3-MCT4[119]. COSMO-Model 6 is parameterised for climate applications over Europe[120]. CLM5 is configured in satellite phenology mode, i.e., for each grid cell and PFT, monthly leaf area index, stem area index, and canopy height are prescribed based on a repeated monthly climatology obtained from satellite data[113]. This means that the PFT-specific vegetation structure varies seasonally and geographically according to observations, but it does not respond to simulated environmental conditions as in the prognostic biogeochemistry mode of CLM5. Nevertheless, even in satellite phenology mode evapotranspiration and gross primary production are calculated as functions of leaf area and PFT-specific rates for stomatal conductance and photosynthesis, which both respond to environmental conditions including temperature, radiation, humidity, $CO_2$ concentration, day length, and water availability[112,113]. Surface albedo varies with leaf and stem area, PFT-specific optical properties (i.e., reflectance and transmittance of leaves and stems in the visible and near-infrared spectra), leaf orientation, solar zenith angle, and incoming direct and diffuse radiation from COSMO. Surface roughness

is a function of leaf and steam area, canopy height, and aerodynamic parameters, and thus also varies in space and time. Here, we produce the spatially explicit input data from high-resolution (0.05°) observations using CLM5 tools. With the crop land unit and irrigation active, our configuration represents terrestrial ecosystems as a combination of up to 17 different PFTs of which one is bare ground.

The model is set up at ~12.5 km horizontal resolution, corresponding to 0.11° on the rotated grid of COSMO and 0.1° on the regular grid of CLM. To optimise the usage of computational resources, COSMO runs on GPU while CLM uses the idling CPU processors on Piz Daint (1 GPU device and 12 CPUs). For model validation we perform a historical run driven with boundary conditions derived from ERA5 reanalysis[121]. The years 2004–2010 are used for model spin-up and 2011–2015 for comparison with observations (Supplementary Note 4). For the future scenarios, COSMO-CLM² is driven with boundary conditions derived from the global climate model MPI-ESM1-2-HR under SSP1-2.6, using the first realisation (r1i1p1f1)[122]. MPI-ESM1-2-HR is chosen due to good performance over Europe[123], medium equilibrium climate sensitivity, and high spatial resolution (~100 km for land and atmosphere). For these reasons, MPI-ESM1-2-HR is also used for CORDEX-CMIP6 simulations over Europe[124] and the CORDEX Flagship Pilot Study LUCAS. The years 2034–2035 are used for model spin-up and 2036–2050 for analysis. The four simulations (SSP1, NfN, NfS, and NaC) have identical parameters except for the PFT composition.

## Analysis of drivers

To investigate how land cover changes affect the simulated temperature, differences in surface (skin or radiometric) temperature can be attributed to components of the surface energy balance[52,109,125]. The surface energy balance equation states that energy inputs from net radiation (left hand side) are balanced by energy outputs (right hand side):

$$\underbrace{(1-\alpha)SW_{down}}_{net\ SW} + \underbrace{LW_{down} - \varepsilon\sigma T_S^4}_{net\ LW} = \underbrace{LH+SH}_{turbulent\ fluxes} + G \quad (1)$$

where $\alpha$ is surface albedo, SW shortwave radiation (down for downwelling), LW longwave radiation, $\varepsilon$ surface emissivity, $\sigma$ the Stefan-Boltzmann constant, $T_s$ surface temperature, LH latent heat flux, SH sensible heat flux, and $G$ ground heat flux. By taking the partial derivative of Eq. (1) and assuming constant surface emissivity, the surface temperature response can be expressed as:

$$\Delta T_S = \frac{1}{4\varepsilon\sigma T_S^3}\left(\underbrace{-\Delta LH - \Delta SH - \Delta G - SW_{down}\Delta\alpha}_{surface\ contribution} + \underbrace{(1-\alpha)\Delta SW_{down} + \Delta LW}_{atmospheric\ feedbacks}\right)$$

$$(2)$$

where $\frac{1}{4\varepsilon\sigma T_S^3}$ (K Wm⁻²) is the surface temperature sensitivity. Following Luyssaert et al.[109], we summarize the cooling effects of in latent heat flux, sensible heat flux, ground heat flux, and albedo as direct contribution from changes at the land surface, and the warming effects of downwelling shortwave and longwave radiation as indirect contribution caused by changes in the overlying atmosphere. Possible mechanisms behind atmospheric feedbacks include changes in atmospheric humidity, cloud cover, temperature, and emissivity. The inputs to Eq. (2) are obtained from model output, except for $\varepsilon$, which is set to 1 and $G$, which is calculated as the residual of the surface energy balance per grid cell (by definition, the energy balance is closed in COSMO-CLM²).

## Analysis of climate effects

For the analysis, outputs of the COSMO-Model are interpolated to the 0.1° regular grid of CLM using bilinear regridding. We focus on the EU+ region, which includes all land areas in COSMO-CLM² that are within

the extent of the land system maps (Fig. 2a), corresponding to 65,849 grid cells covering ~4.9 × 10$^6$ km$^2$. The EU+ region is further dis-aggregated into four European subregions (North, West, East, South) (Fig. 2a). These subregions are independent in terms of policy imple-mentation and land use (i.e., each subregion could pursue a different value perspective of the Nature Futures Framework and demands would be satisfied internally, see ref. 33) and share some environ-mental and climatic characteristics.

We consider six ESM outputs that are sensitive to land cover change and directly relevant for natural and human systems: near-surface air temperature at 2 m, soil temperature in the top 10 cm, near-surface wind speed at 10 m, precipitation, soil moisture in the top 10 cm, and gross primary production. We report annual and seasonal climatological values (2036–2050 mean ± one standard deviation across years) for these variables and the annual maximum of daily maximum temperature at 2 m (TXx) as an extreme temperature index. Results are provided for the modified EU+ region, the four European subregions, the 1% most affected areas per scenario, the 1% most extreme areas in the SSP1 reference, and the 1% most changing areas in SSP1 compared with recent climate.

Statistical significance of the climate response per scenario (i.e., scenario minus SSP1) is tested by applying a two-sided Wilcoxon signed-rank test. The test statistic indicates if the means of two sce-narios are equal or not under similar yearly conditions, so that year-to-year variation is eliminated as a confounding factor given the strong temporal correlation between RCM simulations driven with identical boundary conditions. Statistical significance of changes in SSP1 com-pared with recent climate (i.e., SSP1 minus Recent) is tested by applying a two-sided Mann–Whitney U rank test. The obtained $p$-values are adjusted using the Benjamini–Hochberg procedure, which controls the false discovery rate in multiple hypothesis testing[126].

## Analysis of the contribution of individual land cover transitions

To assess the contribution of PFT changes to the simulated temperature response, we aggregate the PFTs into seven categories: tree nee-dleleaf (including evergreen temperate, evergreen boreal, and deciduous boreal), tree broadleaf (including evergreen temperate, deciduous temperate, and deciduous boreal), shrub (including broadleaf evergreen temperate, broadleaf deciduous temperate, and broadleaf deciduous boreal), grass (i.e., herbs excluding annual crops, including C3 arctic, C3, and C4), crop rainfed, crop irrigated, and bare ground. PFT changes are enabled by opposing changes in another PFT, and the temperature response depends on both PFTs involved in a land cover transition (e.g., grass replacing tree PFTs). However, in fractional PFT maps and in the COSMO-CLM$^2$ model, PFTs do not occupy specific parts of grid cells, so it is not clear which PFT replaced another PFT. To resolve this, we compute net transitions per grid cell iteratively using a greedy approach that matches the most increased and decreased land cover categories in each iteration until all PFT changes are captured. Net transitions are named after the dominant direction per scenario across Europe (e.g., if crop to grass transitions are larger than grass to crop, the net transition is named crop to grass). This results in 20 different net transition types, because transitions between rainfed and irrigated crops occur in none of the scenarios.

To estimate the sensitivity of the local (i.e., grid cell scale) tem-perature response to each of the m = 20 possible land cover transi-tions, we fit a multiple linear regression across $i$ = 1, 2, … $n$ grid cells in the EU+ region or within subregions:

$$\Delta T_i = \beta_0 + \beta_1 x_{i1} + \beta_2 x_{i2} + \ldots + \beta_p x_{ip} + \varepsilon_i \tag{3}$$

where ΔT is the temperature response (i.e., the difference between a scenario and the SSP1 reference), $x_p$ is the fraction of land cover transition $p$, and $\beta_p$ is the corresponding temperature sensitivity. Conceptually, this approach attributes the local temperature response

to land cover transitions in the same grid cell, whereas non-local effects originating from land cover changes in other grid cells contribute to the grid cell-specific error $\varepsilon$ or the intercept $\beta_0$ if the effects are spatially uniform.

Linear regression models with many predictors often suffer from multicollinearity (i.e., two or more predictors are correlated), resulting in erroneously high coefficient estimates. To prevent this, we employ ridge regression, a variant of multiple linear regression that includes a regularisation term in the calculation of the residual sum of squares to shrink coefficients in proportion to their initial size and effectively penalise high coefficient estimates. To determine an appropriate reg-ularisation strength per model fit, we test logarithmically spaced values between 0.1 and ~2000 and perform cross-validation to choose the best value. Mean coefficients and their confidence intervals are estimated based on 200 iterations of the pipeline for model fitting and evaluation (i.e., train-test split, feature standardisation, and ridge regression with cross-validation for hyperparameter tuning). The 200 models are trained and evaluated using spatial block cross-validation to increase the independence of data used for training (75% of blocks) and testing (25% of blocks). For spatially structured data, blocked sampling yields higher but more realistic error estimates[127]. In our application, blocking increases the confidence interval (calculated as 2.5th to 97.5th percentiles of the bootstrap sample) while the means are robust and insensitive to the chosen block size.

We note that the inclusion of spatial predictors like latitude and longitude or background temperature and precipitation improves model performance scores (e.g., mean absolute error, $R^2$), but hardly affects the contribution of individual land cover transitions. Since we focus on explaining contributions and not on predictions, and a clear interpretation of such independent effects is missing, we do not include additional predictors in the final model formulation.

## Reporting summary

Further information on research design is available in the Nature Portfolio Reporting Summary linked to this article.

## Data availability

The ESM data generated in this study have been deposited in the ETH Research Collection at https://doi.org/10.3929/ethz-c-000795598[128]. The land system scenarios used in this study are available at https://doi.org/10.34894/NWGCBY[33]. The habitat projections used in this study are available at https://doi.org/10.5281/zenodo.15307415[57]. The raw vegetation plot data belong to the owners or contributors of each vegetation database and can be requested at https://euroveg.org/eva-database/obtaining-data.

## Code availability

The code used to generate the presented data and figures is available at https://github.com/pesieber/Sieber-etal-2026_NCOMMS.git [https://doi.org/10.5281/zenodo.18511016][129]. The translation of EUNIS habitats to PFTs is available at https://gitlabext.wsl.ch/karger/eunis2pft[130]. The COSMO-CLM$^2$ regional ESM consists of three com-ponents. The source code of COSMO-Model 6 is available to research institutions free of charge under an institutional license from https://www.cosmo-model.org/content/default.htm (last access: October 2022). CLM5 is a publicly released version of the Community Land Model available from https://github.com/ESCOMP/CTSM (last access: October 2022). The OASIS coupler is a community software available from https://oasis.cerfacs.fr/en/home/ (last access: January 2020).

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

## Acknowledgements

This research was funded through the 2019-2020 BiodivERsA joint call for research proposals, under the BiodivClim ERA-Net COFUND program, and with the funding organisations Swiss National Science Foundation SNF (project: FeedBaCks, 193907), Agence nationale de la recherche (ANR-20-EBI5-0001), the Swedish Research Council for Sustainable Development (Formas 2020-02360), the German Research Foundation (DFG BR 1698/21-1, DFG HI 1538/16-1), and the Technology Agency of the Czech Republic (SS70010002). The model simulations were supported by a grant from the Swiss National Supercomputing Centre (CSCS) under project ID s1256. We thank Jonas Jucker (C2SM) and Eric Maisonnave (CERFACS) for support in updating COSMO-CLM[2], and Lukas Gudmundsson (ETH) for helpful discussions during the analysis.

## Author contributions

P.S., D.N.K., N.E.Z., and S.I.S. conceived the study. S.S.M. and W.T. provided the EUNIS habitat predictions. D.N.K. and G.M. developed the translation of EUNIS habitats to PFTs with input from P.S., I.A., and M.C. Z.S., I.B., B.G., and J.D. contributed to data curation within important EVA databases. P.H.V. helped describe and discuss the land system scenarios. P.S. and M.L. updated the COSMO-CLM[2] model. P.S. implemented the scenarios and ran the simulations. P.S. performed the data analysis, created the figures, and drafted the manuscript with input from J.S., S.I.S., and D.N.K. All mentioned authors, further including E.L.D., S.K., H.B., T.H., and J.D., provided comments and contributed to revising the manuscript.

## Competing interests

The authors declare no competing interests.

## Additional information

[1]Institute for Atmospheric and Climate Science, ETH Zurich, Zurich, Switzerland. [2]Swiss Federal Institute for Forest, Snow and Landscape Research WSL, Birmensdorf, Switzerland. [3]University of Grenoble Alpes, University of Savoie Mont Blanc, CNRS, LECA, Grenoble, France. [4]Department of Spatial Sciences, Faculty of Environmental Sciences, Czech University of Life Sciences Prague, Praha-Suchdol, Czech Republic. [5]Department of Botany and Zoology, Faculty of Science, Masaryk University, Brno, Czech Republic. [6]Institute for Environmental Planning, Leibniz Universität Hannover, Hannover, Germany. [7]Alpine Environment and Natural Hazards, WSL Institute for Snow and Avalanche Research SLF, Davos Dorf, Switzerland. [8]Wyss Academy for Nature, University of Bern, Bern, Switzerland. [9]Climate and Environmental Physics, Physics Institute, University of Bern, Bern, Switzerland. [10]Oeschger Centre for Climate Change Research, University of Bern, Bern, Switzerland. [11]Center for Climate Systems Modeling, ETH Zurich, Zurich, Switzerland. [12]Institute of Biology/Geobotany and Botanical Garden, Martin Luther University Halle-Wittenberg, Halle, Germany. [13]German Centre for Integrative Biodiversity Research (iDiv) Halle-Jena-Leipzig, Leipzig, Germany. [14]Senckenberg Biodiversity and Climate Research Centre, Frankfurt am Main, Germany. [15]Department of Physical Geography, Goethe University, Frankfurt am Main, Germany. [16]Faculty of Geotechnical Engineering, University of Zagreb, Varaždin, Croatia. [17]Department of Plant Biology and Ecology, University of the Basque Country UPV/EHU, Bilbao, Spain. [18]Biology Education, Dokuz Eylul University, Izmir, Turkey. [19]Vegetation Ecology Research Group, Institute of Natural Resource Management (IUNR), Zurich University of Applied Sciences (ZHAW), Wädenswil, Switzerland. [20]Bayreuth Center of Ecology and Environmental Research (BayCEER), University of Bayreuth, Bayreuth, Germany. [21]Institute for Environmental Studies, Vrije Universiteit Amsterdam, Amsterdam, Netherlands. ✉e-mail: petra.sieber@env.ethz.ch

