## [Peer Review file · Nature Communications]

Climate response to Nature Future scenarios in a regional Earth System Model

Corresponding Author: Dr Petra Sieber

Version 0:

Reviewer comments:

Reviewer #1

(Remarks to the Author)

This study shows an approach to represent nature management guided by different value perspectives (nature for nature, nature for society and nature for culture) in a regional Earth system model to explore potential effects on climate. The paper shows that nature management focusing on cultural landscapes (Nature for culture scenario) could lead to additional temperature increase in Europe but focusing on species protection (Nature for nature scenario) or carbon accumulation (Nature for society scenario) would not present major trade-offs with climate adaptation and mitigation. The study also highlights that existing policies for climate and biodiversity have implications to both of them also due to the subsequent feedback from land to the atmosphere that those policies will lead to.

The paper is original and important for current discussions on future land management, but I missed clear explanations on the relevance of this study for the broader audience, on the implications of this study and how it can help with a better definition of those scenarios or with future land management. Also, the main text (and the methods) should include a few sentences that explain the rationale in focusing on those aspects for each of the 3 visions. For example, the Nature for society focuses on climate mitigation, but it would be nice to hear why this was the benefit selected given that many others exist, including climate adaptation. Also, given that the focus was on climate, shouldn't the scenario be called Nature for climate mitigation? I am struggling to understand the Nature for culture one (as it is not clear why culturally valued agriculture landscapes were the ones selected, and how that was done), so a sentence in the main text (and in the methods) that explains what was considered in this scenario, and why, would help the reader to better follow the paper. So, in summary, the authors need to justify and defend the aspects that were used to define each vision and make sure that only those 3 aspects of the visions were addressed. Finally, the paper would be a lot easier to read if the authors removed some of the acronyms.

I like the paper and can be a great contribution to the field if the comments above are considered. Below are some specific suggestions on how to make it more clear and relevant to the general audience:

Suggest cutting some acronyms as it is too hard to read as is. One needs to go back to the introduction to remember what the acronyms refer to. For example, I suggest that you use "Nature for Nature" instead of NFN and the same for the other 2 visions. Likewise, in the results, I suggest that you don't use too many acronyms (LAI, Txx, EF) as the paper is hard to read as is. Those may be very obvious for some readers that work on the field, but difficult for the general audience.

line 52: suggest to add ecosystems that are important for climate adaptation of people as you have mentioned adaptation in line 109, but not before that.

line 40-41: this sentence could be a bit more clear. For example, it could be: "The development of scenarios that enhance biodiversity while stabilizing the climate should take into account????"

line 56: suggest you also add one example of how a biodiversity goal may threaten a climate goal, like you did in lines 55-56.

Line 90: is tree planting afforestation? is that why you distinguished it from nature protection and restoration? If so, it may not

be a correct approach to include afforestation in that scenario as it can lead to decreases in diversity especially in non-forested areas. Also, not clear why you decided to focus on carbon in the NFS society given that there are so many other ecosystems services/nature's contributions to people? Along the same lines, if you focus on carbon accumulation potential, is then not that surprising that it did not present major trade-offs with climate mitigation? Likewise, among all the potential cultural values of nature, why the decision was to focus on culturally valued agricultural landscapes? and what does that mean? maybe that is clear for Europeans, but it may not be for people in other regions.

I recommend therefore that you add a few sentences in the introduction to explain a bit the rationale to focus on those aspects for each of the 3 visions.

Line 116: please explain how the scenarios, especially NFS and NFN, would lead the replacement of low intensity to high intensity systems.

Line 120: please explain how the scenario that promotes species conservation (nature for nature) led to increased crop cover given that this scenario promotes biodiversity and ecosystems? What does that mean, that the land transformations needed to promote species conservation is actually not having the expected impact? Or that the scenario that you have built is actually not good in promoting species conservation?

On a more general sense, current discussions focus on the need to manage the land with multiple benefits in mind. That would mean, in the case of this study, that we should implement specific actions and in places where intrinsic, instrumental and relational values could be maximized. Based on your results, do you think it would be possible to maximize all 3 visions? In any case, it would be nice to add a sentence or two on how your study could help in moving those discussions forward.

(Remarks on code availability)

Reviewer #2

(Remarks to the Author)

The manuscript "Climate response to Nature Future scenarios in a regional Earth System Model" addresses an important challenge - considering IPCC linked SSP1 and IPBES linked Nature for Future-based scenarios in regional (Europe focussed) Earth system modelling. The idea behind the inclusion of the Nature for Future idea is to consider human and societal values and human-nature relations in the modelling.

The idea is interesting and highly relevant. The modelling itself is comprehensive, impressive and well explained.

However, what is not clear enough is:

- how "comprehensive socio-ecological information in the design of ESM scenarios" are integrated. What is actually understood by "comprehensive socio-ecological information"? More detail needs to be provided here beyond mentioning the KM GBF and the Green Deal.
- in the description of the results, how socio-ecological information used in the design of ESM scenarios effect scenario outcomes. With this, it is not clear enough how "different value perspectives" are integrated and which potential outcome they have.
- how Nature for Society and Nature as Culture are integrated in the scenarios, understood in the modelling and then impact on the modelling results although this is argued to be the main benefit of this kind of analysis in the introduction.

It should be explained how "different value perspectives translate into and guide nature management" and what the "different perspectives" here are.

The Discussion of the results of the NAC scenario should be more precise when mentioning "trade-offs".  Which trade-offs are meant when arguing that "Higher annual maximum temperatures, higher summer temperatures, and lower summer precipitation under NAC than SSP1 indicate trade-offs with climate adaptation."?  The trade-offs and climate adaptation measures should be explained in more detail providing also references.

Methodology

"The three other scenarios additionally implement targets of the Kunming-Montreal Global Biodiversity Framework and the European Union's Green Deal related to nature conservation (30% of the land area by 2030), restoration, reduced fertiliser application in agriculture, increased tree cover, and improved access to nature." Please explain and provide more detail.

How is "access to nature" considered in the scenario development and in the modelling?

"In NFS, priority is given to areas with high aboveground carbon accumulation potential to explore the consequences of ecosystem-based climate mitigation (although ideally, a wider range of ecosystem services should be considered)."  Which are those areas? And which ecosystem services are meant? Needs explanation.

"In NAC, priority is given to culturally valued agricultural landscapes, considering (low) management intensity, landscape heterogeneity, and landscape value and meaning."
 More detail is needed to explain what is meant by "more priority is given" and which landscapes are exactly considered and how?

Abstract:

Given my previous comments, it is hard to follow the conclusions from the abstract. "Using Earth System Modelling and detailed land use, habitat, and species projections, we show that nature management guided by relational values (Nature As Culture) could lead to additional temperature increases further threatening local biodiversity and human well-being. Conversely, promoting nature for intrinsic values (Nature For Nature) or ecosystem services (Nature For Society) would not present major trade-offs with climate adaptation and mitigation due to land-atmosphere feedbacks."

This is a little hard to conclude because it is not clear enough how actually human values or ecosystem services have been addressed in the modelling. More detail and information is needed in the manuscript. Actually, the ecosystem services and trade-off concept is just briefly mentioned but the argument is not fully spelled out.

Minor comment

Authors should reconsider their use of abbreviations. Given the high number of abbreviations, it is hard to follow the flow of the manuscript text.

(Remarks on code availability)

Reviewer #3

(Remarks to the Author)

Review, Nature Communications

Overall, this is an important modelling study, showing how the different natural futures scenarios might impact land cover and some environmental variables in Europe.

Not sure the "As" in "Nature As Culture" should be capitalized. Same for the "For" in "Nature For Nature".

I find the inclusion of both NFF (as Nature Futures Framework) and NFN (Nature For Nature) as acronyms, sometimes in the same paragraph, confusing. I recommended that you just don't use NFF, always writing this phrase out (which doesn't occur that often). There is no need to say "NFF scenarios", just say "scenarios" most of the time!

For the same reason, consider writing out PFTs rather than having that as an acronym. In general, if an acronym is used only a few times, consider just writing it out. For instance, "EF" seems like it could just be "evaporative fraction".

Introduction- Good clear well written introduction.

In the discussion of tradeoffs in land management to achieve multiple global goals, you might consider citing the following (although it is cited later in the paper I see):

Tallis, H. M., P. L. Hawthorne, S. Polasky, J. Reid, M. W. Beck, K. Brauman, J. M. Bielicki, S. Binder, M. G. Burgess, E. Cassidy, A. Clark, J. Fargione, E. T. Game, J. Gerber, F. Isbell, J. Kiesecker, R. McDonald, M. Metian, J. L. Molnar, N. D. Mueller, C. O'Connell, D. Ovando, M. Troell, T. M. Boucher, and B. McPeck. 2018. An attainable global vision for conservation and human well-being. *Frontiers in Ecology and the Environment* 16:563-570.

I find this sentence a bit overstated "Until recently, the scientific community lacked a framework to account for the ways society values nature...". Rather, there have been a number of frameworks (Sustainable development, MEA, etc.), with the IPBES one being the latest and a distinct improvement in a number of technical ways!

The Introduction might be improved if, instead of ending with a set of research questions, it ended with a set of hypotheses that are being tested here. That is, from past research, the authors must have had a sense of what the answers to their research questions might be...

A bit more info is needed in the Introduction on how the scenarios were defined. In particular, it isn't clear to the non-European reader why the NAC scenario involves an increase of grassland and a decline in cropland, for instance (I get the cultural context, since I live in Europe, but readers from the US or Australia, for instance, might not).

Results:

Overall, it would be good if the results, instead of presenting a set of scenario results by variable, were better tied to the overarching questions laid out in the Introduction. Perhaps in sub-sections?

Could results be represent relative to current land cover, in addition to SSP1 reference scenario in 2050?

Please say a word about why Ridge Regression rather than regular regression.

Discussion:

Overall, this is a well-written section. But I do wish that it talked more broadly about comparisons among the NFF Scenarios. That is, rather than focusing the discussion on primarily how the modeling found out different implications for the different scenarios, I found myself wanting a more normative comparison among scenarios. What should Europe do in managing, it's landscape? So of in the form of "If Europe's goal was to... then the implication is..."

Methods

I am not qualified to judge all of these methods, as some are from other fields.

Some summary figures/tables in the Results (perhaps with little zoomed in example maps?) would be helpful to tell people the key implications of the different NFF scenarios for land cover. Almost all of the model results follow from this, so it seems key to clearly show (for instance) tree cover and grass cover and crop cover under the various scenarios. I realize Figure 2 goes into this, but I found myself really squinting at Figure 2 and trying to see more detail about what was going on in specific countries. Clearly, a lot of detail is in the Supplementary Figures, but I would suggest elevating one of the figures from that to the main manuscript.

(Remarks on code availability)

Reviewer #4

(Remarks to the Author)

This study employs a regional Earth system model, fed with detailed land use and land cover change scenario data derived from the Nature Futures Framework (NFF), to assess the potential impacts of three nature management scenarios (NFF, NFS, and NAC) on regional climate in Europe. The key findings of this study include that some nature management could incur trade-offs among societal values, ecosystem services and climate mitigation (potential warming effects), particularly when the management is guided by relational values (NAC). While this could offer some insights into sustainable development policies in Europe, the methodology and analysis of this manuscript currently suffer from several flaws. In the broader theme, the effects of land use and land cover change on regional or global climate is a very old topic. Exploring the trade-offs among different nature values (including societal, ecosystem/climate services and biodiversity) under different land use scenarios are also not new (Smith et al; Fastré et al.; Thompson et al.; see references below). The novelty of this work is mainly in the implementation of NFF-based scenarios in a regional ESM, but the current analysis is not sufficiently robust, and the findings of climate effects and their drivers are not surprising, which are mostly expected (well-known mechanisms) from existing knowledge in this field (see below comments). Other existing studies have offered more in-depth investigation of land management effects on regional climate in Europe (Luyssaert et al. 2018 and Naudts, K. et al 2016). I hope the following major and minor comments could help improve this work.

Major comments:

Methodological issues:

1. Model setup:

1) The authors used satellite phenology mode for CLM5 in the regional climate model COSMO-CLM. This mode prescribes monthly LAI and canopy height derived from MODIS for historical simulations. However, how did you prescribe future LAI and canopy height for the period 2034-2050? No matter how, there will be substantial uncertainties in future vegetation status even though the PFT covers are sophisticatedly derived from NFF. Furthermore, in the satellite phenology mode, BGC is fully off which means no interactive vegetation growth and carbon-nitrogen biogeochemistry and fertilization effects. In this context, your simulations can hardly capture future land-atmosphere interactions and the full biogeophysical and biogeochemical aspects of land use change effects on climate (see Bala et al., Zhu et al, Betts et al.).

2) Your climate forcing is only from one climate model "MPI-ESM1-2-HR" for the future period 2034-2040 and is not bias-corrected. In this case, there would be substantial uncertainty in the future climate. Even if fully bias-corrected, the many CMIP6 models still have a large spread in future climate projections for any given climate scenario. Therefore, most modelling studies in this field employ multiple CMIP6 models (often >10) as climate forcing so that you can show the ensemble mean and uncertainties, instead of a single value.

2. Analysis:

1) In the analysis of climate effects, the authors mentioned six climate variables, but GPP and SM are eco-hydrological, not strictly climate variables. It is better to focus on temperature and precipitation and other climate variables. GPP is useless in

this context, especially given that it is output from the satellite phenology mode of CLM (directly related to prescribed LAI), which tells little about the carbon cycle. You can turn on the BGC mode and look at NPP, NEP or NEE, so that you can really talk about the climate service of NFF scenarios in terms of their carbon cycle values such as carbon sequestration and carbon stocks.

2) In the analysis of drivers, there are two issues. First, the regression mixed physical variables (albedo, EF, and roughness length) from NAC, NFN and NFS scenarios with climate variables (air temperature, precipitation, and wind speed) from the reference scenario (SSP1) as confounding factors. The validity of this approach is suspicious, as can be seen from the divergent/unstable results presented in Fig. 5a, Fig. S7a and Fig. S9a (and low R2 in some regions and scenarios). The regression should use climatological means of temperature, precipitation and wind (and specific humidity, also important) from the historical period (ERA5) or the SSP1-2.6 forcing data of MPI-ESM1-2-HR to represent climate background effects. Instantaneous values of these variables from COSMO-CLM SSP1 are not appropriate and could introduce bias to the regression analysis of NAC, NFN and NFS scenarios. More importantly to validate this approach, the authors should present a sensitivity analysis, that is, using your regression coefficients to show temperature responses to unit differences or unit changes in predictor variables under each scenario. We should expect similar feature importance rank and sign and size of effect of each variable among different scenarios (that is, the sensitivity of air temperature to each relevant predictor/driver should be relatively stable across these land use scenarios). If the current plots show total effect, instead of unit effect, please clarify. Please also clarify if you differenced NAC, NFN and NFS from SSP1 and used the differences in each variable in the regression analysis. You could also focus on the 1% most affected areas to do sensitivity analysis. If the sensitivity plots are not consistent among scenarios, the authors should consider alternative attribution approaches (see below).

Second, the two predictor sets used in ridge regression include intrinsic physical variables (albedo, EF, and roughness length; Fig. 5a) and upstream surface properties (PFT fractions, vegetation height, LAI, and irrigated area) together with vegetation characteristics (broadleaf tree, needleleaf tree, grass, and crop; Fig. 5b). The second set used for Fig. 5b are highly correlated and not suitable to be mixed in regression; they can all be decomposed to the three intrinsic variables in Fig. 5a (and unconsidered ones like photosynthetic rate, stomatal conductance, water use efficiency, C:N ratios, etc.). Although ridge regression can partly mitigate multicollinearity, they cannot fully separate the contributions of these highly correlated variables and properly infer causality as you mentioned (lines 406–407). Therefore, analysis for Fig. 5b and related ones are not meaningful. Furthermore, ridge regression assumes linearity, while temperature and dynamics are highly nonlinear (e.g., the effect of EF on temperature may exhibit threshold effects). The authors can consider using machine learning methods (e.g., Random Forest, XGBoost) to capture nonlinear relationships, or combine them with causal inference methods, (e.g., Structural Equation Modeling (SEM)), to quantify the independent contributions of each driver.

Finally, I recommend the authors to consider using robust physically based attribution methods to quantify the contribution of intrinsic physical variables related to land-use change (such as changes in albedo, EF, roughness, or surface and aerodynamic conductance, or ET/latent and sensible heat fluxes) to surface climate (see Betts et al. 1997, Zeng et al. 2017, He et al. 2020, Chen & Dirmeyer (also used CLM)).

Results and discussion:

1) The main findings appear to be theoretical inferences based on existing eco-climatological and physical mechanisms (e.g., line 187: “We expect warmer temperatures from albedo decrease, EF decrease, and roughness decrease”), rather than causal relationships that cannot be established by ridge regression itself (e.g., lines 190–191: EF is the dominant predictor for cooler temperatures in all scenarios and subregions; line 193: albedo is also mostly linked to cooling as expected; line 207: the decline in irrigated area contributes strongly to warmer temperatures). The strong signal of EF is apparent, but the effect of albedo is far from clear from the Fig. 5a, Fig. S7a and Fig. S9a (inconsistent signs and sizes of albedo effect across regions and scenarios). The authors explain that some discrepancies exist because albedo differences are very small in summer. But the differences are even smaller under NFN and NFS scenarios than under NAC, then why can you derive that albedo is more important in Alpine subregion in NFN and NFS? Although the seemingly important role of albedo in Alpine regions is expected, which is consistent with the dominant role of albedo in high-latitude regions (similarly high snow cover to Alpine regions), the unstable performance of the regression analysis across regions and scenarios currently prevents a robust understanding of the drivers behind temperature changes and warrants further refinement of the analysis (see above).

2) The study analyzed land use change effects on annual maximum temperatures and precipitation. The Discussion could briefly consider heat stress and droughts, particularly in Southern Europe, where such impacts could be severe. This would enhance the study’s relevance to climate adaptation and risk management. Readers will wonder the current small effects on temperature and precipitation may not incur significant ecological and social-economic damages compared to the reference scenario. Moreover, the effect of irrigation on air temperature is only the tip of iceberg of heat impact (irrigation may increase heat stress due to increased humidity which is currently not included in the analysis; see Mishra, V. et al, Lobell et al).

3) The authors state that precipitation changes at the subregional scale are driven by non-local effects of atmospheric moisture transport (lines 154–158). Although relevant studies were cited in support of this claim, this mechanism is not directly analyzed in the study. Therefore, it should be framed as a hypothesis rather than a confirmed conclusion. It is recommended that the Discussion section should consider alternative factors, including: (1) Scale effect – vegetation-precipitation or ET-climate feedbacks may be weak at small local scales but stronger at regional scales; (2) Time-lag effects – the impact of PFT changes on precipitation may have seasonal delays.

4) Your manuscript mentioned “biodiversity” and its tradeoff with climate goals many times, but your current methods couldn’t

quantify the effect on biodiversity under different scenarios. Merely saying “additional warming and drying could threaten biodiversity” (e.g., lines 238-240) is insufficient without quantitative analysis of biodiversity metrics and climate-biodiversity relationships (and the projected changes in temperature and precipitation are very small). There are relevant works that have quantified the tradeoffs between biodiversity conservation and climate mitigation services under scenarios of nature-based solutions, which are not cited yet (Smith et al; Fastré et al.).

5) The Discussion currently focuses on statistical correlations without deeply analyzing underlying physical mechanisms. Therefore, it is recommended that the authors consider causal inference or physically based attribution methods and expand discussion on the physical mechanisms driving temperature changes (there is a broad literature to compare). Note that the statement in line 212 “reduced turbulence (negative correlation with roughness)” is wrong.

6) In the Discussion, the authors propose additional indicators to assess the value of different NFF scenarios (lines 274–278), which aligns closely with existing ecosystem service valuation frameworks. The authors could cite relevant literature on established ecosystem valuation systems, such as The Economics of Ecosystems and Biodiversity (TEEB), and Natural Capital Accounting (NCA), and discuss how the NFF framework can leverage these approaches to enhance its practical applicability and policy implications.

Minor comments:

Line 103: no need to cite unpublished work.

Lines 119-120: this is surprising. Nature conservation scenarios should normally promote tree diversity instead of crop expansion.

Line 185: “emergent” is a popular word, but it should refer to properties not directly apparent or arising from complex interactions between simpler components. Are these variables mentioned really emergent characteristics? At least, albedo is an intrinsic physical property, but not “emergent”.

Line 202: It is unclear whether this is your hypothesis or the analysis outcome. If you already knew it would not perform well, why still including all these inter-correlated variables?

Line 212: surface turbulence should be positively correlated with roughness.

Line 228: NAC highly values agricultural landscape. Then, why there is large conversion of cropland to grassland in this scenario?

Lines 238-239: it is unconvincing that such small additional warming and drying compared with recent climate could further threaten local biodiversity and human well-being over large parts of Europe. Only in 1% of most affected grid cells, there could be some additional climate impacts when warming exceeds 0.5 degC (lines 220-221)

Line 247: In the discussion, the readers would expect a bit more elaboration about specific cultural/value drivers behind the land use changes under different NFF scenarios, although they have been described in methods. Some unexpected land use transition mentioned above should also be clarified.

Lines 256-258: and more limitations in models, for example, tree age and phenological development after land use transitions and plant adaptation to heat and drought stresses are not represented yet in most ESMs including CLM5.

Line 294: the scale of land cover changes in these NFF scenarios are actually rather small, compared to historical land use changes across many parts of the world since a century ago caused by agricultural expansion and urbanization.

Lines 295-296: These studies (Luyssaert et al. 2018 and Naudts et al. 2016) and others (Zhang et al. 2024) about unexpected effects of afforestation and deforestation also suggest that the estimated/expected climate effects shown in this study may be subject to uncertainties without fully consider the biogeochemical and biophysical effects of land surface change (see major comments on other variables and attribution methods to consider).

Lines 361-362: The satellite phenology mode will disable two-way vegetation-climate feedback effects and biogeochemical cycles, which are important for the analysis of land use impact on climate. And please clarify how LAI and canopy height are prescribed for future scenarios after the observational period?

Lines 373-374: Climate forcing from one model “MPI-ESM1-2-HR” and not bias-corrected would have substantial uncertainty.

References:

1. Jeffrey R. Smith et al. Variable impacts of land-based climate mitigation on habitat area for vertebrate diversity. *Science* 387,420-425(2025). DOI:10.1126/science.adm9485
2. Fastré, C., Possingham, H.P., Strubbe, D. et al. Identifying trade-offs between biodiversity conservation and ecosystem services delivery for land-use decisions. *Sci Rep* 10, 7971 (2020). <https://doi.org/10.1038/s41598-020-64668-z>
3. Thompson, J. R. et al. The consequences of four land-use scenarios for forest ecosystems and the services they provide. *Ecosphere* 7, e01469 (2016).
4. Naudts, K. et al. Europe's forest management did not mitigate climate warming. *Science* 351, 597–600 (2016).
5. Zhu, L. et al. Comparable biophysical and biogeochemical feedbacks on warming from tropical moist forest degradation. *Nat. Geosci.* 1–6 (2023) doi:10.1038/s41561-023-01137-y.
6. Bala, G. et al. Combined climate and carbon-cycle effects of large-scale deforestation. *PNAS* 104, 6550–6555 (2007).
7. Betts, R. A., Cox, P. M., Lee, S. E. & Woodward, F. I. Contrasting physiological and structural vegetation feedbacks in climate change simulations. *Nature* 387, 796–799 (1997).
8. Bonan, G. *Ecological Climatology: Concepts and Applications*. (Cambridge University Press, Cambridge, 2015). doi:10.1017/CBO9781107339200.
9. Zeng et al. Climate mitigation from vegetation biophysical feedbacks during the past three decades. *Nature Climate Change* 7, 432–436 (2017).
10. He, M. et al. Amplified warming from physiological responses to carbon dioxide reduces the potential of vegetation for climate change mitigation. *Commun Earth Environ* 3, 1–10 (2022).
11. Chen, L. & Dirmeyer, P. A. Distinct Impacts of Land Use and Land Management on Summer Temperatures. *Frontiers in*

Earth Science 8, (2020).

12. Mishra, V. et al. Moist heat stress extremes in India enhanced by irrigation. Nat. Geosci. 13, 722–728 (2020).

14. Lobell, D. B., Bonfils, C. J., Kueppers, L. M. & Snyder, M. A. Irrigation cooling effect on temperature and heat index extremes. Geophysical Research Letters 35, (2008).

15. Zhang, Y. et al. Asymmetric impacts of forest gain and loss on tropical land surface temperature. Nat. Geosci. 1–7 (2024) doi:10.1038/s41561-024-01423-3.

(Remarks on code availability)

Reviewer #5

(Remarks to the Author)

(Remarks on code availability)

Version 1:

Reviewer comments:

Reviewer #2

(Remarks to the Author)

Thank you for revising your manuscript and thoroughly addressing my comments. The clarity of the conceptual sections, particularly regarding trade-offs and ecosystem services, has improved significantly. I also greatly appreciate the addition of the new Figure 1.

However, there remain some ambiguities in the scenarios and assumptions depicted in Figure 1. It is unclear where some of these assumptions originate from, such as the increase of 5 billion trees, urban planning, and others. Could you elaborate on how you developed the rules for the modeling? While Figure 1 is indeed useful, the implementation details of these policies are still somewhat obscure. For instance, what do terms like "Prioritization of areas with high habitat" or "Increased tree density in urban areas" entail? How did you arrive at the figure of 5 billion new trees, and what method governs their distribution? All assumptions underlying the scenarios should be thoroughly justified and explained, especially concerning the species selection for increased tree cover, which significantly impacts temperature outcomes. Moreover, the details and justification of the "Urban planning" component, along with its assumptions, are missing completely. I could not find any explanation, justification, or link to the Nature Future scenarios within the main text.

Each assumption translated into the single Nature Futures concept scenarios must be well-explained and justified. E.g, why is "Increased demand for low-intensity grassland" categorized under Nature as Culture value?

I recommend providing a table of scenario assumption rules, including specific numbers, justifications, and references. This table could also clarify how each rule or assumption aligns with the intrinsic, instrumental, or relational values within the Nature for Future framework.

Finally, in the abstract, while you now mention "land use management," you continue to refer to "nature management." Please ensure that terms are defined and used consistently throughout the manuscript.

(Remarks on code availability)

Reviewer #3

(Remarks to the Author)

Rereview of Nature Communications NCOMMS-24-79005A

The changes made by the authors to clarify the scenarios used have improved the clarity of the manuscript. I find now Figure 1 a fairly helpful summary of the different scenarios.

They have also done a good job trying to make clearer how land cover changes from current conditions to SSP1 and thence to their three scenarios. The new Figure 3 does a good job with that. I would suggest the authors edit the figure legend for part 3A though to make it clearer, since the "baseline of the hat graph" may not be clear to readers. It might be possible, for instance, to draw the baseline in another color like red, and then have the figure caption say something like "Differences are shown for the scenarios relative to the reference (SSP1, represented by the red baseline) and recent conditions (represented by the zero line). For instance, forest cover increases by 4% [or whatever] in SSP1 relative to current conditions but is reduced below the SSP1 scenario by all three of our scenarios, and in the case of the NaC scenario forest cover would actually be less than under current conditions." (I believe that is the correct interpretation of Figure 3A, but I am not sure!). The authors shouldn't feel bound by my suggestion, it is just one of I am sure multiple ways to make the caption clearer.

Overall, they have made a good faith effort to respond to most of my specific comments, although I will confess I have not had time to carefully review all of them.

It would be helpful if the authors found time to explain, in the Methods or Results or Discussion, more why (if I am understanding Figure 3 correctly) that all three Nature Future Framework scenarios decreased forest cover, relative to SSP1. In particular, I don't understand (and didn't go to Dou et al. to check their methods) why under the NfS scenario, there wouldn't be MORE forest cover than under SSP1. As someone who works for an organization that promotes (where appropriate) reforestation in part for its climate mitigation benefits, it is surprising for me that the NfS scenario has less tree cover than the SSP1 scenario. After all, allowing forest to naturally regenerate after agricultural abandonment (which I assume is part of what is happening under SSP1) could be a way to achieve significant carbon sequestration, as it was in the eastern US after agricultural abandonment and subsequent reforestation (and an increase in some rare biodiversity).

(Remarks on code availability)

Reviewer #4

(Remarks to the Author)

Thanks for authors' thorough revisions and responses, which have considerably improved the manuscript. The methodological clarifications, restructuring of the driver analysis, and strengthened discussion represent major progress compared to the original submission. Most of my initial concerns have been adequately addressed. Nevertheless, a few important issues remain that require further clarification or minor additions to ensure transparency and robustness of the study's conclusions. The detailed comments are as follows:

1. The use of the satellite phenology (SP) mode in CLM5 is still a key limitation and the justification for carbon sequestration results can still be improved. I am not asking for redoing BGC simulations at this stage but choosing SP mode definitely limited the robustness of your findings regarding carbon and climate responses of nature management scenarios. Computational cost is not a strong motivation nowadays for this choice. It is partly true that this simplification reduces uncertainties related to the seasonality (monthly phenology) of vegetation structure, but you lost the big picture of carbon cycle responses to land use changes. Any carbon cycle variable (including GPP) generated in the SP mode and its interannual dynamics would be hardly comparable with observations (you can try a comparison with FLUXNET or one of MODIS-based GPP products for example) because the nitrogen cycle, soil biogeochemistry and carbon cycle-climate feedbacks are all turned off in SP and only monthly climatology of LAI, SAI and canopy height are prescribed, without interannual variations. Given this key limitation, it's good that GPP figures are moved to the supplement, but the following concerns should be addressed. In the revised methods, it remains unclear how future vegetation structure for 2034-2050 is specifically prescribed, i.e., whether monthly LAI and canopy height are repeated from historical satellite climatology? This is very likely the case, as SSP scenarios prescribe land cover change in the future according to LUH2 but hardly know the future vegetation structure and phenology. Thus, the potential implications of this static assumption and neglect of interannual and decadal variations in vegetation structure should be discussed. Even under the low-emission SSP1-2.6 scenario, such a static prescription could overestimate afforestation benefits (afforested grid cells or land-units would abruptly attain the climatology LAI and canopy height without undergoing a lengthy growing stage from seedling to mature as in the full BGC mode) and underestimate deforestation and degradation impacts by ignoring perturbations to soil carbon pools that are only interactively simulated in the BGC mode. So on. The actual differences between SP and BGC modes are huge – SP means only biogeophysical modules such as water and energy fluxes (including stomatal response) are active but the vegetation structure is not responsive to climate (no interannual vegetation dynamics), whereas the prognostic BGC mode involves full above- and below-ground biogeochemical cycles and their interaction with climate (including feedback loops between vegetation dynamics, carbon cycle and the climate system). Simply putting "The only difference lies in whether LAI is prescribed or simulated, and the prescribed LAI might be less uncertain than the predicted LAI" is not convincing and misleading. Please also remove the CLM aspects of biogeochemical processes and biological processes such as "Biogeophysical and biogeochemical processes are simulated independently for each PFT, but the atmosphere is assumed horizontally homogeneous within each grid cell" in Methods because they are only for the BGC mode. Also note that nitrogen cycle and its effects on carbon cycle are fully turned off in the SP mode of CLM5, so please remove 'nitrogen' from the new sentence: "stomatal conductance and photosynthesis, which both respond to environmental conditions including temperature, radiation, humidity, CO₂ concentration, day length, and availability of water and nitrogen". I recommend (a) explicitly stating in the Methods whether LAI and canopy height are based on a repeated historical satellite climatology, and (b) expanding the Discussion to address the potential biases in carbon sequestration findings from this static assumption and the limitations of not enabling BGC and not considering carbon cycle-climate feedback associated with land use change.

2. The authors' rationale for using MPI-ESM1-2-HR is now convincing, and the comparison with other CMIP6 models in the supplementary notes and additional sensitivity analysis of regional climate response to the background climate are very helpful. However, please also acknowledge in the Discussion or the relevant Methods section that the study relies on a single uncorrected GCM, that inter-model spread remains an important source of uncertainty, and that boundary conditions were not bias-corrected. Then you can add your argument that "we explore possible future scenarios (so predictability is not of major importance) and we investigate differences between scenarios under identical background climate (so all scenarios face the same forcing biases)."

3. The new results "Drivers of temperature response" based on two separate attribution analyses are very good. But in the

text, "The warming effect is partly compensated by increased sensible heat flux, ground heat flux," may need some revision or clarification. Currently, Fig. 7 shows the individual effects of changes in these fluxes on surface/skin temperature. But such description of increased sensible heat together with decreased latent heat is a typical phenomenon associated with surface air warming and the increased ground heat flux is also the driver of soil temperature increase. Thus, it may be better to differentiate the responses of surface temperature and 2-m air temperature here.

4. The addition of results for warming and drying in hottest/dryest and most heating/drying areas of Europe increases the policy relevance of the study. The definition of "most heating areas" is not very clear at the moment, especially in this sentence "but the most heating areas under SSP1 would not get significantly warmer" which is confusing. Although the authors say that detailed heat impact analysis is outside the scope of this study, the following sentence is not accurate: "The additional warming and drying compared with recent climate could further threaten local biodiversity and human well-being, particularly in hot spot areas for heat stress and droughts in Southern Europe but also in the West and East", because drying and reduced humidity due to conversion of irrigated cropland to grassland can mitigate (rather than further exacerbate) heat stress caused by dry-bulb temperature alone. I suggest that the authors explicitly clarify in the Discussion that the study focuses on temperature and precipitation only, and that interactive humidity effects on heat stress and human well-being were not assessed. This will help readers correctly interpret the limitations of the study regarding health impacts of different scenarios.

(Remarks on code availability)

Reviewer #5

(Remarks to the Author)

(Remarks on code availability)

Version 2:

Reviewer comments:

Reviewer #2

(Remarks to the Author)

Thank you for the detailed clarification of my concerns. The manuscript has improved. No further comments from my side.

(Remarks on code availability)

Reviewer #3

(Remarks to the Author)

This document is a review of a revision (NCOMMS-24-79005B), in response to comments from myself and other reviewers. Overall, the paper has improved upon revision, I do not have any other major comments or critiques.

Small comments:

Note that the letter to reviewers has a formatting error at a few points where it says "Reference source not found". I assume this is just a formatting error from citation software after copying this text from the manuscript document to the letter for reviewers, but the authors might want to check that all the citations are correct in the main manuscript text.

I appreciate the additional info on the Dou et al. scenarios added to the Introduction it has made the paper clearer. Figure 1 is very helpful. I also appreciate the text in the letter to reviewers, describing the Dou et al. methods more. In a sense, the authors have done what they can to make the Dou et al. methods clear. I disagree with some of the assumptions baked into the Dou et al. scenarios regarding reforestation, but it is not that those assumptions are wrong per se (there are multiple future scenarios possible), and they are descriptions of one plausible future.

I appreciate the changes to Figure 3 and its caption, which have improved its readability.

(Remarks on code availability)

Reviewer #4

(Remarks to the Author)

Thanks for the authors' careful responses. I don't have further comments.

(Remarks on code availability)

Reviewer #5

(Remarks to the Author)

(Remarks on code availability)

Dear Reviewers,

We appreciate the insightful comments and suggestions to improve our manuscript. They have been invaluable in guiding the revision, particularly to clarify the scenarios, their effects on land cover, and the implications of our work. We addressed the methodological concerns and added additional analyses to corroborate our results. The most important improvements concern:

- A. Description of scenario design (introduction, new Fig. 1)
- B. Description of the scenario effects on land systems and PFTs (first results subsection)
- C. Addition of a physics-based approach (third results subsection)
- D. Revision and sensitivity analysis of the attribution approach (fourth results subsection)
- E. Implications of the study and limitations of the modelling approach (discussion)

Please find our point-by-point response to the comments below. All references to numbered items (i.e., figures and supplementary figures and tables) refer to the revised versions of the manuscript and supplementary materials. Because questions about points A-B were raised by several reviewers, we provide common answers below and refer to them in the point-by-point responses.

Thank you for your time and consideration. We look forward to your feedback.

Sincerely,

Petra Sieber (on behalf of all co-authors)

A. Description of scenario design

Several reviewer comments indicate that there were unclarities in our description of the scenario design. We therefore expanded the scenario description in the introduction (copied below) and added a new figure that summarizes the characteristics of each scenario (Fig. 1). We also clarified that the scenarios are adopted from Dou et al., who operationalised European policies and the Nature Future Framework perspectives in a land use model to generate land system maps. In this study, we used these land system maps as an input to model the spatial distribution of PFTs, which could then be implemented in the Earth System Model to simulate the climate response.

The Nature Futures Framework was recently used as a lens for developing plural land system scenarios for Europe in 2050³³: a reference that complies with the projected population, commodity demands, and climate under a sustainable low-emissions scenario (SSP1-2.6), and three scenarios that additionally implement targets of the Kunming-Montreal Global Biodiversity Framework and the European Union's Green Deal. These climate and biodiversity policies are implemented while focusing on either intrinsic, instrumental, or relational values according to the Nature Futures Framework (Fig. 1). Specifically, Dou et al.³³ prioritised biodiversity conservation and land sparing in the NfN scenario, climate change mitigation and land sharing in the NfS scenario, and cultural landscapes and human-nature interactions in the NaC scenario. In an NfS perspective, various regulating and provisioning ecosystem services could be considered⁴⁵, and climate change mitigation was chosen as one key function in the climate crisis. Several provisioning services (e.g., production of crops, livestock, and wood) are guaranteed in all scenarios to meet human demands under the SSP1 storyline. Biodiversity is represented by habitat suitability for terrestrial vertebrate species, climate change mitigation by

carbon sequestration potential in forests⁵², and cultural value by agricultural heritage landscapes (i.e., rural areas characterised by traditions, low management intensity, landscape heterogeneity, and high value and meaning)⁵³. The resulting land system maps differ in the area covered by various land systems (e.g., greatest proportions of low-intensity systems in NfN, high-intensity forests in NfS, and mosaic systems in NaC) and their spatial distribution across Europe. The scenarios diverge from the SSP1 reference on ~30% of European land, and among each other on ~20% of European land³³. Land use changes on such a scale could have profound impacts on land surface processes, with particularly strong effects on local to regional climate through changes in albedo and evapotranspiration (i.e., biogeophysical effects)^{41,54-56}.

Fig. 1. Characteristics of the Nature Futures Framework perspectives and scenario implementation in Dou et al. utilised in this study. Intensification means changes in land use intensity (e.g., mid to high intensity) and diversification transitions to mosaic systems (e.g., cropland to forest/shrub and cropland mosaic). Resulting changes in plant functional types are quantified in this study through the workflow illustrated in Fig. 2.

B. Description of the scenario effects on land systems and PFTs

Several reviewer comments indicate that there were unclarities in our description of the scenario effects on land systems and PFTs. We therefore expanded the first results subsection by first describing changes under the reference compared with recent land cover, before describing the effects of nature management (i.e., scenarios compared with the reference). We added Fig. 3a to show how changes in land systems propagate to changes in habitat types and PFTs (both SSP1-Recent and scenario-SSP1). In the text, we made clear links between the land system changes found by Dou et al. and the PFT cover changes quantified in this study. We also clarify potentially counterintuitive land system or PFT changes in the scenarios by explaining the underlying drivers.

Under SSP1, mid- and high-intensity forest systems expand compared to 2015, and we find a corresponding increase in tree PFT cover, particularly broadleaf deciduous trees (Fig. 3a). Grassland and cropland systems decline, resulting primarily in reduced C3 grass and rainfed crop PFT cover (note that grass PFTs in ESMs represent herbs except annual crops). Mosaic systems expand with small effects on several PFTs. Although shrublands decline, shrub PFT cover increases due to gains in, e.g., forest/shrub and cropland mosaics.

Compared with the SSP1 reference, the NfN, NfS, and NaC scenarios show PFT cover differences on 10-21% of European land (see Fig. 2a for the considered region, hereafter EU+). These differences derive directly from the scenario targets and priorities (e.g., restoration in all scenarios, carbon sequestration in NfS), but also from compensatory effects to meet multiple demands (e.g., intensification to compensate for increases in natural ecosystems), regional differences in demands and supply (e.g., depending on population and yields), and environmental differences that affect the habitat and PFT composition of land systems. At the European level, all three scenarios project a substantial expansion of low-intensity systems to meet the targets for protection, restoration, tree cover, and sustainable agriculture (see Fig. 1 and Dou et al.³³ for details). This conversion and extensification is enabled by intensification of mid-intensity forests and croplands to satisfy demands. As a result, all scenarios have less tree PFT cover and more grass PFT cover than the reference, and thereby tend to counterbalance the changes under SSP1 in 2050 compared with recent PFT cover (Fig. 2c). The NfN scenario promoting biodiversity conservation shows the strongest increase in low-intensity systems, particularly grasslands, while the NfS scenario promoting climate change mitigation shows the strongest increase in high-intensity systems, particularly forests. NfN and NfS foresee only mild intensification of mid-intensity croplands, so that crop PFT cover increases because of the greater co-occurring extensification (yet cropland declines compared with recent land cover). Contrarily, the NaC scenario focusing on cultural landscapes foresees strong intensification and diversification (i.e., conversion to mosaic systems) of mid-intensity croplands and thus an overall cropland decline (Fig. 2a). Mosaic systems consist of various habitats including grassland, shrubland, forest, and man-made types (Supplementary Table 1), which in turn consist of various PFTs (Supplementary Table 2). As a result, we find reduced crop PFT cover compared with the reference and more strongly increased grass PFT cover than in the other scenarios (Fig. 2c).

Fig. 2. Land cover in the scenarios compared with the reference (SSP1) and recent conditions. **a**, Land systems are translated into habitat types and plant functional types (PFTs). Differences are shown for the scenarios relative to the reference (SSP1, represented by the baseline of the bar graph) and recent conditions (represented by the zero line). **b**, Dominant land cover in the reference (SSP1) considering the soil and vegetation-covered parts of the grid cells. The land cover categories aggregate temperate and boreal PFT types, deciduous and evergreen tree PFTs, C3 and C4 grass PFTs (i.e., herbs excluding annual crops), and rainfed and irrigated crop PFTs (tropical variants are not present). **c**, Dominant land cover transitions (i.e., main transitions in at least 5% of the grid cell area) in the scenarios relative to SSP1. Dominant transitions are determined based on maximum gains and losses per grid cell. The eight most common transitions across scenarios are shown. Supplementary Fig. 2 shows fractional changes of main PFT types. **d**, Differences in land cover composition in the scenarios relative to SSP1 and in SSP1 relative to recent conditions in the EU+ region. Fig. 5 shows a breakdown for European subregions.

Reviewer #1 (Remarks to the Author):

This study shows an approach to represent nature management guided by different value perspectives (nature for nature, nature for society and nature for culture) in a regional Earth system model to explore potential effects on climate. The paper shows that nature management focusing on cultural landscapes (Nature for culture scenario) could lead to additional temperature increase in Europe but focusing on species protection (Nature for nature scenario) or carbon accumulation (Nature for society scenario) would not present major trade-offs with climate adaptation and mitigation. The study also highlights that existing policies for climate and biodiversity have implications to both of them also due to the subsequent feedback from land to the atmosphere that those policies will lead to.

The paper is original and important for current discussions on future land management, but I missed clear explanations on the relevance of this study for the broader audience, on the implications of this study and how it can help with a better definition of those scenarios or with future land management. Also, the main text (and the methods) should include a few sentences that explain the rationale in focusing on those aspects for each of the 3 visions. For example, the Nature for society focuses on climate mitigation, but it would be nice to hear why this was the benefit selected given that many others exist, including climate adaptation. Also, given that the focus was on climate, shouldn't the scenario be called Nature for climate mitigation? I am struggling to understand the Nature for culture one (as it is not clear why culturally valued agriculture landscapes were the ones selected, and how that was done), so a sentence in the main text (and in the methods) that explains what was considered in this scenario, and why, would help the reader to better follow the paper. So, in summary, the authors need to justify and defend the aspects that were used to define each vision and make sure that only those 3 aspects of the visions were addressed. Finally, the paper would be a lot easier to read if the authors removed some of the acronyms.

I like the paper and can be a great contribution to the field if the comments above are considered. Below are some specific suggestions on how to make it more clear and relevant to the general audience:

Thank you for the thoughtful and constructive feedback. We sincerely appreciate your suggestions and believe the revisions greatly strengthened the manuscript. Please see detailed responses below.

Overall, we improved the description of scenario design (introduction, new Fig. 1), the description of the scenario effects on land systems and PFTs (first results subsection), and the discussion to highlight the implications of the results and which insights it provides for future land management, modelling, and scenario design. The discussion is now structured as follows:

- Different implementations of European climate and biodiversity policies have important consequences for the distribution of plant functional types
- Climate effects are strongest locally but also relevant on subregional and regional scales
- Turbulent fluxes and atmospheric feedbacks drive the surface temperature response in summer
- Two land cover transition types contribute most strongly to cooler and warmer summer temperatures

- Nature management focusing on cultural services could cause additional challenges for regional climate adaptation and mitigation
- Advances and limitations regarding the inclusion of Nature Future scenarios in Earth System Models
- Advances and limitations regarding land use scenario modelling

Suggest cutting some acronyms as it is too hard to read as is. One needs to go back to the introduction to remember what the acronyms refer to. For example, I suggest that you use "Nature for Nature" instead of NFN and the same for the other 2 visions. Likewise, in the results, I suggest that you don't use too many acronyms (LAI, Txx, EF) as the paper is hard to read as is. Those may be very obvious for some readers that work on the field, but difficult for the general audience.

We removed the acronym for Nature Futures Framework (previously "NFF"). The "NFF scenarios" are now called "scenarios", and the "SSP1 reference scenario" is called "SSP1 reference" for better distinction. We kept the scenario acronyms for readability but changed the spelling as suggested by reviewer 3 (i.e., NfN, NfS, NaC). We further removed the acronyms for LAI, EF, and GPP in the text (TXx was kept for readability).

line 52: suggest to add ecosystems that are important for climate adaptation of people as you have mentioned adaptation in line 109, but not before that.

Thank you for the suggestion, we added the following sentence: *"Furthermore, trees and forests can contribute to climate change adaptation by cooling temperatures locally, particularly during daytime and the boreal summer."*

line 40-41: this sentence could be a bit more clear. For example, it could be: "The development of scenarios that enhance biodiversity while stabilizing the climate should take into account????"

Indeed, this could be clearer. The sentence now reads *"These results highlight the need for integrative scenarios that enhance biodiversity and stabilise the climate, while considering biogeophysical effects on local to regional temperatures."*

line 56: suggest you also add one example of how a biodiversity goal may threaten a climate goal, like you did in lines 55-56.

We added the following sentence: *"Conversely, protecting important habitats for biodiversity may restrict areas available for renewable energy or agricultural production."*

Line 90: is tree planting afforestation? is that why you distinguished it from nature protection and restoration? If so, it may not be a correct approach to include afforestation in that scenario as it can lead to decreases in diversity especially in non-forested areas. Also, not clear why you decided to focus on carbon in the NFS society given that there are so many other ecosystems services/nature's contributions to people? Along the same lines, if you focus on carbon accumulation potential, is then not that surprising that it did not present major trade-offs with climate mitigation? Likewise, among all the potential cultural values of nature, why the decision was to focus on culturally valued agricultural landscapes? and what does that mean? maybe that is clear for Europeans, but it may not be for people in other regions. I recommend therefore that you add a few sentences in the introduction to explain a bit the rationale to focus on those aspects for each of the 3 visions.

Several reviewer comments indicate that there were unclarities in our description of the scenario design. We therefore improved the introduction and added a new figure (Fig. 1), see **point A**.

In Dou et al., tree planting includes (1) increasing tree density in urban areas and low intensity agricultural systems; and (2) expanding forests and shrublands in line with the policy priorities (i.e., biodiversity conservation, carbon sequestration, or cultural landscapes). Consequently, tree planting is possible in the NfN scenario in ways that is compatible with biodiversity conservation. Protection means preventing land system conversion and intensification, and nature restoration means promoting low intensity grasslands and forests. This information is included in the new Fig. 1.

For the NfS scenario, we now explain in the introduction that *“In the NfS perspective, various regulating and provisioning ecosystem services could have been considered and climate change mitigation was chosen as one key function in the climate crisis. Several provisioning services (e.g., production of crops, livestock, and wood) are guaranteed in all scenarios to meet human demands under the SSP1 storyline.”* This means that the NfS scenario does consider several NCP but not exclusively, and it does not consider all possible NCP. In the discussion section, we acknowledge this limitation and suggest additional indicators: *“based on existing data NfS could be represented by additional indicators for flood control, air and water quality regulation, pollination, and biogeophysical climate regulation”*.

For the NaC scenario, we added in the introduction: *“cultural values are represented by agricultural heritage landscapes (i.e., rural areas characterised by traditions, low management intensity, landscape heterogeneity, and high value and meaning)”*.

Line 116: please explain how the scenarios, especially NFS and NFN, would lead the replacement of low intensity to high intensity systems.

Thank you for pointing this out. All scenarios have a stronger tendency for intensification than SSP1 to satisfy demands in light of the co-occurring de-intensification (i.e., mid-intensity systems are replaced by both high-intensity and low-intensity systems). We clarified this in the text and refer to Dou et al. for details. In line 116, there was a mistake, which we corrected to **“the replacement of medium-intensity by high-intensity forests”**.

Several reviewer comments indicate that there were unclarities in our description of the scenario effects on land systems and PFTs. We therefore expanded the first results section, see **point B**.

Line 120: please explain how the scenario that promotes species conservation (nature for nature) led to increased crop cover given that this scenario promotes biodiversity and ecosystems? What does that mean, that the land transformations needed to promote species conservation is actually not having the expected impact? Or that the scenario that you have built is actually not good in promoting species conservation?

We agree that this sounds counterintuitive. See **point B** for improvements regarding the description of scenario effects on land systems and PFTs.

Cropland declines in all scenarios relative to recent conditions (year 2015), but the declines are smaller under NfN and NfS than under the SSP1 reference. Compared with SSP1, NfN and NfS show stronger extensification than intensification, so that crop PFT cover increases. We clarified this in the text: *“NfN and NfS foresee only mild intensification of mid-intensity croplands, so that crop PFT cover*

increases because of the greater co-occurring extensification (yet cropland declines compared with recent land cover)."

Cropland expansion is possible in the NfN scenario in ways compatible with species conservation, particularly if species can be protected in other (more valuable) areas. So, there is not necessarily a contradiction between the PFT changes and the scenario intention. We added the following clarification in the discussion: *"Particularly at the local scale, changes can be difficult to interpret or seem counterintuitive. The results are path-dependent and complex, and the link to the initial scenario assumptions is not always evident (e.g., NfN projects cropland expansion to compensate for extensification, NfS projects forest decline enabled by intensification). Such emergent outcomes allow a realistic representation of how land systems may change across Europe and deviate from overall non-spatial policy targets and visions. Considering the implemented land system allocation rules, it is unlikely that the scenarios outcomes go against their premises. However, the scenarios are driven by policy means and not their outcomes, so they are not necessarily efficient at achieving policy goals (e.g., NfS promotes forests in areas with high carbon sequestration potential but does not maximise carbon sequestration). An in-depth evaluation is outside the scope of this study and the reader is referred to Dou et al.³³ for details on the scenarios and their limitations."*

On a more general sense, current discussions focus on the need to manage the land with multiple benefits in mind. That would mean, in the case of this study, that we should implement specific actions and in places where intrinsic, instrumental and relational values could be maximized. Based on your results, do you think it would be possible to maximize all 3 visions? In any case, it would be nice to add a sentence or two on how your study could help in moving those discussions forward.

This is a very interesting and relevant question. In this study, we used existing scenarios generated through a scenario approach that does not aim at maximising outcomes. Instead, it imposes demands and policies as drivers and generates possible (not necessarily optimal) outcomes given various rules for land system transitions (e.g., resistances). This is comparable with real-world developments, but it cannot be used directly to inform spatial planning like optimisation approaches. We included a paragraph in the discussion section:

The land systems scenarios employed in this study provide a positive view of the future, in which European climate and biodiversity policies are implemented. In the modelling technique used by Dou et al.³³, the same commodity demands and policy targets are imposed in all three scenarios, but the land system changes required to fulfil them differ depending on the priorities for nature management (i.e., biodiversity conservation, climate change mitigation, or cultural services). The scenario outcomes reflect different processes across Europe, which can be opposing due to contextual differences or compensatory effects. Particularly at the local scale, changes can be difficult to interpret or seem counterintuitive. The results are path-dependent and complex, and the link to the initial scenario assumptions is not always evident (e.g., cropland increase in NfN due to extensification, or forest decline in NfS enabled by intensification and in NaC by diversification). Such emergent outcomes allow a realistic representation of how land system changes may vary across Europe and deviate from overall non-spatial visions. Considering the land system allocation rules, it is unlikely that the outcomes go against the premises of the scenarios. However, the scenarios are driven by policy means and not their outcomes, so they are not necessarily most efficient at achieving policy goals (e.g., the NfS scenario promotes but does not maximise carbon sequestration) and ex-

post assessment is needed to evaluate the scenario outcomes. An in-depth evaluation is outside the scope of this study and the reader is referred to Dou et al.³³ for details on how the scenarios were constructed and their limitations. To support integrated spatial planning, spatial optimisation techniques can be used to develop scenarios that maximise target variables (e.g., biodiversity and carbon sequestration⁸¹) under given constraints. Such methods could be used to co-optimize the delivery of various ecosystem services, but the comprehensive quantification of aspects important to intrinsic, instrumental, and relational nature values is still challenging.

Reviewer #2 (Remarks to the Author):

The manuscript "Climate response to Nature Future scenarios in a regional Earth System Model" addresses an important challenge - considering IPCC linked SSP1 and IPBES linked Nature for Future-based scenarios in regional (Europe focussed) Earth system modelling. The idea behind the inclusion of the Nature for Future idea is to consider human and societal values and human-nature relations in the modelling. The idea is interesting and highly relevant. The modelling itself is comprehensive, impressive and well explained.

Thank you for the insightful comments and suggestions, they were very helpful for revising the manuscript. Please see detailed responses below.

However, what is not clear enough is:

- how "comprehensive socio-ecological information in the design of ESM scenarios" are integrated. What is actually understood by "comprehensive socio-ecological information"? More detail needs to be provided here beyond mentioning the KM GBF and the Green Deal.

Thank you for pointing this out. Several reviewer comments indicate that there were unclarities in our description of the scenario design. We therefore improved the introduction and added a new figure (Fig. 1), see **point A**. As shown in the new Fig. 1, each scenario implements specific policy targets of the KM GBF and the Green Deal in different ways. In this study, we used these land system maps as an input to model the spatial distribution of PFTs, which could then be implemented in the Earth System Model to simulate the climate response.

The phrase "comprehensive socio-ecological information in the design of ESM scenarios" may cause confusion, as it sounds like we designed new scenarios in this study. To improve clarity, the sentence now reads: "*This study is, to our knowledge, the first to implement Nature Future scenarios in an ESM and to incorporate comprehensive ecological information in the translation of land systems to PFTs.*" Comprehensive ecological information refers to the newly developed method which considers habitat predictions and species composition across Europe, as now described in the introduction.

- in the description of the results, how socio-ecological information used in the design of ESM scenarios effect scenario outcomes. With this, it is not clear enough how "different value perspectives" are integrated and which potential outcome they have.

The three value perspectives of the Nature Futures Framework are built into the NfN, NfS, and NaC scenarios of Dou et al. (i.e., they are considered in the land systems maps that we use as inputs). We revised the text in the introduction to make this clearer. Furthermore, in the new Fig. 1 we describe

the scenarios characteristics, how European climate and biodiversity policies are implemented, and how this leads to land system changes and differences among scenarios. See **point B** for improvements regarding the description of scenario effects on land systems and PFTs.

- how Nature for Society and Nature as Culture are integrated in the scenarios, understood in the modelling and then impact on the modelling results although this is argued to be the main benefit of this kind of analysis in the introduction. It should be explained how "different value perspectives translate into and guide nature management" and what the "different perspectives" here are.

See replies to earlier comments, this is addressed by **points A and B**.

The Discussion of the results of the NAC scenario should be more precise when mentioning "trade-offs".  Which trade-offs are meant when arguing that "Higher annual maximum temperatures, higher summer temperatures, and lower summer precipitation under NAC than SSP1 indicate trade-offs with climate adaptation."?  The trade-offs and climate adaptation measures should be explained in more detail providing also references.

Thank you for pointing this out. With "trade-offs with climate adaptation [goals]" we referred to local/regional temperature reduction itself, not adaptation measures. We agree that this formulation is imprecise and difficult to understand. Throughout the paper, we replaced the formulation with "additional challenges for local adaptation", or "trade-offs with local climate impacts and the resulting need for adaptation".

Methodology

"The three other scenarios additionally implement targets of the Kunming-Montreal Global Biodiversity Framework and the European Union's Green Deal related to nature conservation (30% of the land area by 2030), restoration, reduced fertiliser application in agriculture, increased tree cover, and improved access to nature." Please explain and provide more detail. How is "access to nature" considered in the scenario development and in the modelling?

We expanded the scenario description in the main text to provide better background how each value perspective of the Nature Futures Framework is incorporated in the land system scenarios of Dou et al., which served as an input for our ESM implementation. See **point A** for improvements regarding the description of scenario design. We revised the methods section accordingly and provided additional details how the policies are implemented in the modelling approach of Dou et al.: "*Policies are implemented through changes in (1) demands (e.g., increased demand for natural ecosystems and trees); (2) supply of goods and services (e.g., increased tree cover in urban systems); (3) land system conversion order and resistance (e.g., to prioritise high-intensity cropland for crop production or to protect natural ecosystems); and (4) spatial weights (e.g., to prioritise areas for protection).*"

We agree that "access to nature" is unclear. We now call it "urban planning", meaning the following (included in the new Fig. 1):

- NfN: Increased population density to prevent urban expansion (land sparing)
- NfS: Decreased population density in cities to allow for more green spaces (land sharing)
- NaC: Increased population density in villages to enable human-nature interactions

"In NFS, priority is given to areas with high aboveground carbon accumulation potential to explore the consequences of ecosystem-based climate mitigation (although ideally, a wider range of ecosystem services should be considered)."  Which are those areas? And which ecosystem services are meant? Needs explanation.

We improved the scenario description in the introduction, see **point A**. For the NfS scenario, we now explain in the introduction that *"In the NfS perspective, various regulating and provisioning ecosystem services could have been considered and climate change mitigation was chosen as one key function in the climate crisis. Several provisioning services (e.g., production of crops, livestock, and wood) are guaranteed in all scenarios to meet human demands under the SSP1 storyline."* This means that the NFS scenario does consider several NCP but not exclusively, and it does not consider all possible NCP. In the discussion section, we acknowledge this limitation and suggest additional indicators (*"based on existing data NfS could be represented by additional indicators for flood control, air and water quality regulation, pollination, and biogeophysical climate regulation"*).

"In NAC, priority is given to culturally valued agricultural landscapes, considering (low) management intensity, landscape heterogeneity, and landscape value and meaning."  More detail is needed to explain what is meant by "more priority is given" and which landscapes are exactly considered and how?

We improved the scenario description in the introduction, accompanied by Fig. 1. For the NaC scenario, we now explain in the introduction that *"Dou et al.33 prioritised ... cultural services and human-nature interactions in the NaC scenario. ... cultural values are represented by agricultural heritage landscapes (i.e., rural areas characterised by traditions, low management intensity, landscape heterogeneity, and high value and meaning)54"*.

The spatial prioritisation was needed in the method of Dou et al., to ensure that the implementation of protection, restoration, and increased tree cover is in line with the policy priority of delivering cultural values. This is now clarified in Fig. 1 (e.g., culturally valued agricultural landscapes are prioritised for protection and exempt from forestation). To represent culturally valued agricultural landscapes, Dou et al. obtained a map of agricultural heritage landscapes from Tieskens et al., which is referenced in the introduction (ref 54). In the discussion, we acknowledge that the NaC scenario could consider additional indicators (*"NaC could be represented by additional indicators for heritage forests, wild foods, access to nature, and nature tourism"*).

Abstract:

Given my previous comments, it is hard to follow the conclusions from the abstract. "Using Earth System Modelling and detailed land use, habitat, and species projections, we show that nature management guided by relational values (Nature As Culture) could lead to additional temperature increases further threatening local biodiversity and human well-being. Conversely, promoting nature for intrinsic values (Nature For Nature) or ecosystem services (Nature For Society) would not present major trade-offs with climate adaptation and mitigation due to land-atmosphere feedbacks." This is a little hard to conclude because it is not clear enough how actually human values or ecosystem services have been addressed in the modelling. More detail and information is needed in the manuscript. Actually, the ecosystem services and trade-off concept is just briefly mentioned but the argument is not fully spelled out.

We expanded the scenario description to explain how the three value perspective of the Nature Futures Framework were considered. Furthermore, the new Fig. 1 describes the scenario characteristics, how European climate and biodiversity policies are implemented, and how this leads to land system changes and differences among scenarios. While the abstract does not provide the space for such details, Fig. 1 should provide sufficient clarification. Regarding ecosystem services, we now specify the services considered in each scenario in the introduction and acknowledge additional services that could have been considered in the discussion (see replies to previous comments).

Minor comment

Authors should reconsider their use of abbreviations. Given the high number of abbreviations, it is hard to follow the flow of the manuscript text.

We agree. We kept the scenario acronyms for readability but changed their spelling as suggested by reviewer 3 (i.e., NfN, NfS, NaC) and removed the acronym for Nature Futures Framework (NFF). The “NFF scenarios” are now called “scenarios”, and the “SSP1 reference scenario” is called “SSP1 reference”. We removed the acronyms for LAI, EF, and GPP in the text (TXx was kept for readability).

Reviewer #3 (Remarks to the Author):

Overall, this is an important modelling study, showing how the different natural futures scenarios might impact land cover and some environmental variables in Europe.

Thank you for the constructive comments. We sincerely appreciate your feedback and believe the revisions greatly strengthened the manuscript. Please see detailed responses below.

Not sure the "As" in "Nature As Culture" should be capitalized. Same for the "For" in "Nature For Nature". I find the inclusion of both NFF (as Nature Futures Framework) and NFN (Nature For Nature) as acronyms, sometimes in the same paragraph, confusing. I recommended that you just don't use NFF, always writing this phrase out (which doesn't occur that often). There is no need to say "NFF scenarios", just say "scenarios" most of the time!

Thank you for the suggestion. We changed the spelling of the scenario acronyms as suggested (i.e., NfN, NfS, NaC) and removed the acronym for Nature Futures Framework (NFF). The “NFF scenarios” are now called “scenarios”, and the “SSP1 reference scenario” is called “SSP1 reference”.

For the same reason, consider writing out PFTs rather than having that as an acronym. In general, if an acronym is used only a few times, consider just writing it out. For instance, "EF" seems like it could just be "evaporative fraction".

We kept the acronym for PFTs because it is used ~100 times in the paper and common in the field (e.g., we wanted to specify “grass PFT” every time rather than writing grasses/grasslands, which can include various plant types). We removed other acronyms like LAI, EF, and GPP (TXx was kept for readability).

Introduction- Good clear well written introduction.

Thank you.

In the discussion of tradeoffs in land management to achieve multiple global goals, you might consider citing the following (although it is cited later in the paper I see):

Tallis, H. M., P. L. Hawthorne, S. Polasky, J. Reid, M. W. Beck, K. Brauman, J. M. Bielicki, S. Binder, M. G. Burgess, E. Cassidy, A. Clark, J. Fargione, E. T. Game, J. Gerber, F. Isbell, J. Kiesecker, R. McDonald, M. Metian, J. L. Molnar, N. D. Mueller, C. O'Connell, D. Ovando, M. Troell, T. M. Boucher, and B. McPeck. 2018. An attainable global vision for conservation and human well-being. *Frontiers in Ecology and the Environment* 16:563-570.

Thank you for the suggestion, we included the reference already in the introduction: *"However, land systems are multifunctional and complex, so nature management for climate and biodiversity goals may compete, and it may affect current livelihoods25-27."*

I find this sentence a bit overstated "Until recently, the scientific community lacked a framework to account for the ways society values nature...". Rather, there have been a number of frameworks (Sustainable development, MEA, etc.), with the IPBES one being the latest and a distinct improvement in a number of technical ways!

We agree and rephrased as follows: *"To support the development of such nature-focused scenarios, the Intergovernmental Science-Policy Platform on Biodiversity and Ecosystem Services (IPBES) developed the Nature Futures Framework."*

The Introduction might be improved if, instead of ending with a set of research questions, it ended with a set of hypotheses that are being tested here. That is, from past research, the authors must have had a sense of what the answers to their research questions might be...

We improved the introduction by highlighting several research gaps in individual paragraphs (i.e., development of integrative scenarios, inclusion of integrative scenarios in ESM modelling, and translating land use scenarios into ESM input). The scenarios are exploratory and contain spatially very heterogeneous land use changes, so it is difficult to construct useful hypothesis that do not misguide the reader. For instance, one could hypothesise that a NfS scenario focusing on climate change mitigation would increase forest cover and thus cool temperatures locally/regionally, but this is not the case for reasons rooted in the scenario modelling approach by Dou et al. rather than in our ESM implementation (see additions in the first results section and in the discussion). Therefore, we decided to stick to the presentation as research questions, but we improved the wording of the final paragraph of the introduction (and structured the discussion according to these questions):

"This approach enables us to answer the following questions: (1) How do different implementations of European climate and biodiversity policies affect the PFT distribution across Europe? (2) At which scale and magnitude do the scenarios affect mean climate or extreme temperatures? (3) What are the underlying biogeophysical and ecosystem-related drivers? Finally, we assess if nature management focusing on biodiversity conservation, climate change mitigation, or cultural services could cause additional challenges for regional climate adaptation and mitigation. This study is, to our knowledge, the first to implement Nature Future scenarios in an ESM and to incorporate comprehensive ecological information in the translation of land systems to PFTs."

A bit more info is needed in the Introduction on how the scenarios were defined. In particular, it isn't clear to the non-European reader why the NAC scenario involves an increase of grassland and a

decline in cropland, for instance (I get the cultural context, since I live in Europe, but readers from the US or Australia, for instance, might not).

Several reviewer comments indicate that there were unclarities in our description of the scenario design. We therefore expanded the scenario description in the introduction and added a new figure that summarizes the characteristics of each scenario (Fig. 1), see **point A**. For the NaC scenario, we now explain that “cultural values are represented by agricultural heritage landscapes (i.e., rural areas characterised by traditions, low management intensity, landscape heterogeneity, and high value and meaning)”.

Results:

Overall, it would be good if the results, instead of presenting a set of scenario results by variable, were better tied to the overarching questions laid out in the Introduction. Perhaps in sub-sections?

We organised the results in subsections that are linked to the research questions formulated in the final paragraph of the introduction: (1) Effects on distributions of plant functional types, (2) Effects on climate in a regional Earth System Model, (3) Drivers of the temperature response, (4) Contribution of individual land cover transitions

Furthermore, we revised and expanded the section “Effects on climate in a regional Earth System Model” to provide an integrated assessment per scenario considering all variables. See new Fig. 5 and accompanying assessment: *“Overall, the NfS scenario would be associated with no significant negative impacts in terms of warming, drying, or reduced ecosystem carbon uptake over Europe and with the least trade-offs in all subregions (Fig. 5). The NfN scenario may be preferable if the goal was to reduce summer temperatures (not significant) or to prevent further warming in the most heating areas, but it could lead to drier soils in the East and lower ecosystem carbon uptake in North, West, and South. For the NaC scenario, we find significant negative impacts in most regards (except soil moisture drying) over Europe and all subregions except for the North.”*

During this revision, we modified our definition of the subregions (originally five subregions based on climate and environmental similarity following EURO-CORDEX) to match that of the land system scenarios of Dou et al. Since these regions were modelled independently by Dou et al., this allows us to evaluate which policy would be best per subregion: *“The EU+ region is further disaggregated into four European subregions (North, West, East, South) (Fig. 2a). These subregions are independent in terms of policy implementation and land use (i.e., each subregion could pursue a different value perspective of the Nature Futures Framework and demands would be satisfied internally, see ref33) and share some environmental and climatic characteristics.”*

Fig. 3. Land cover difference and climate response in each scenario relative to the SSP1 reference at the European scale (EU+) and in subregions. The land cover categories aggregate temperate and boreal PFT types, deciduous and evergreen tree PFTs, and C3 and C4 grass PFTs (i.e., herbs excluding annual crops). The climate response is shown for annual maximum temperature at 2 m (TXx), summer temperature at 2 m, summer precipitation, summer soil moisture at 0-10 cm, and annual gross primary production. Colours indicate the direction of the response (e.g., blue for cooling, red for warming) and darker colours indicate statistical significance at the 95% confidence level.

Could results be represent relative to current land cover, in addition to SSP1 reference scenario in 2050?

Thank you for the suggestion. We added a paragraph in the beginning of the first results subsection, describing the changes in land systems and PFTs in the SSP1 reference compared with recent land cover. This is helpful to understand that several increases relative to SSP1 are actually decreases relative to today, albeit less strong than in SSP1. We included the comparison between SSP1 and recent land cover in Fig. 3.

In terms of climate, we added results for recent conditions and differences between SSP1 and recent conditions in Supplementary Table 4. This makes it easy to calculate differences between scenarios and recent conditions if that is of interest (without duplicating the results for each scenario). Furthermore, we used these results to analyse effects of the scenarios in areas that change the most under SSP1 compared with recent climate (e.g., most heating areas). This adds an interesting perspective to the analysis.

Please say a word about why Ridge Regression rather than regular regression.

We added the following explanation in the methods section: *“Linear regression models with many predictors often suffer from multicollinearity (i.e., two or more predictors are correlated), resulting in erroneously high coefficient estimates. To prevent this, we employ ridge regression, a variant of multiple linear regression that includes a regularisation term in the calculation of the residual sum of squares to shrink coefficients in proportion to their initial size and effectively penalise high coefficient estimates.”*

Discussion:

Overall, this is a well-written section. But I do wish that it talked more broadly about comparisons among the NFF Scenarios. That is, rather than focusing the discussion on primarily how the modeling found out different implications for the different scenarios, I found myself wanting a more normative comparison among scenarios. What should Europe do in managing, it's landscape? So of in the form of "If Europe's goal was to... then the implication is..."

Thank you for this suggestion. In the results, we now provide an integrated assessment per scenario considering all variables, accompanied by the new Fig. 5 (see previous comment). In the discussion, we dedicated a new subsection to this:

Nature management focusing on cultural services could cause additional challenges for regional climate adaptation and mitigation:

Overall, our results show that the NaC scenario would have the strongest detrimental effects on climate. Higher annual maximum temperatures, higher summer temperatures, and lower summer precipitation under NaC than SSP1 indicate trade-offs with local climate impacts and the resulting need for adaptation. The European continent has warmed by more than 2 °C since the pre-industrial period and is expected to further warm in the coming decades⁸⁶. Even small increases in temperature can increase the intensity and frequency of extreme weather and affect ecosystems and human health^{5,87,88}. The additional warming and drying under NaC could thus further threaten local biodiversity and human well-being, particularly in hot spot areas for heat stress and droughts in Southern Europe but also in the West and East. A detailed assessment of losses in species climatic range size (see refs.^{9,89}) or health impacts (see refs.^{87,90}) was, however, outside the scope of this study. Reduced vegetation carbon uptake (-11 ± 0.4 %) suggests lower potential for ecosystem-based mitigation, but dedicated simulations would be needed to quantify the net response of the land carbon cycle (i.e., additionally considering respiration, disturbances, and soil carbon dynamics). In contrast, the NfS scenario shows no significant negative impacts in terms of warming, drying, or reduced ecosystem carbon uptake over Europe and the least trade-offs in all subregions. The NfN scenario could result in drier soils and lower vegetation carbon uptake but would cool summer temperatures in the South. Trade-offs with climate adaptation and mitigation are thus low, although local compromises exist in all scenarios. Based on these findings, new scenarios could be developed that promote biodiversity while helping stabilise the climate.

Methods

I am not qualified to judge all of these methods, as some are from other fields.

Some summary figures/tables in the Results (perhaps with little zoomed in example maps?) would be helpful to tell people the key implications of the different NFF scenarios for land cover. Almost all of

the model results follow from this, so it seems key to clearly show (for instance) tree cover and grass cover and crop cover under the various scenarios. I realize Figure 2 goes into this, but I found myself really squinting at Figure 2 and trying to see more detail about what was going on in specific countries. Clearly, a lot of detail is in the Supplementary Figures, but I would suggest elevating one of the figures from that to the main manuscript.

Thank you for pointing this out. We improved the description of the scenarios (e.g., summary of characteristic land system changes and differences in the new Fig. 1) and the resulting changes in PFTs (first results subsection) to show the key implications of each scenario for land cover over Europe. For the subregional scale, we now summarise PFT changes per subregion as part of Fig. 5 (this was previously in the supplement) and link it to climate effects.

A breakdown of Fig. 3c into individual PFTs can be found in supplementary Fig. S2. We did not elevate this figure to the main manuscript because it provides additional details rather than highlighting the key implications. Country level results are not the focus of this study (rather European and subregional scale), but interested readers will be able to zoom into the high-resolution maps in Fig. 3c and Fig. S2.

Reviewer #4 (Remarks to the Author):

This study employs a regional Earth system model, fed with detailed land use and land cover change scenario data derived from the Nature Futures Framework (NFF), to assess the potential impacts of three nature management scenarios (NFN, NFS, and NAC) on regional climate in Europe. The key findings of this study include that some nature management could incur trade-offs among societal values, ecosystem services and climate mitigation (potential warming effects), particularly when the management is guided by relational values (NAC). While this could offer some insights into sustainable development policies in Europe, the methodology and analysis of this manuscript currently suffer from several flaws. In the broader theme, the effects of land use and land cover change on regional or global climate is a very old topic. Exploring the trade-offs among different nature values (including societal, ecosystem/climate services and biodiversity) under different land use scenarios are also not new (Smith et al; Fastré et al.; Thompson et al.; see references below). The novelty of this work is mainly in the implementation of NFF-based scenarios in a regional ESM, but the current analysis is not sufficiently robust, and the findings of climate effects and their drivers are not surprising, which are mostly expected (well-known mechanisms) from existing knowledge in this field (see below comments). Other existing studies have offered more in-depth investigation of land management effects on regional climate in Europe (Luyssaert et al. 2018 and Naudts, K. et al 2016). I hope the following major and minor comments could help improve this work.

Thank you for the detailed and constructive feedback. We sincerely appreciate your suggestions and believe the revisions greatly strengthened the manuscript. Please see detailed responses below.

Generally, we agree that the main novelty of our work lies in the implementation of NFF-based scenarios in an ESM (using new methods to consider detailed habitat and species information) and the investigation of the resulting land use/management effects on European climate. We asked whether such mixed and modest land use/cover changes would have statistically significant effects in an ESM at all, and at what spatial scale and magnitude. Then, we analyse the climate effects while considering various land use/cover changes, whereas Luyssaert et al. and Naudts et al. focused on

past and future forest management only. Here, our aim was not to understand how individual land cover changes/transitions affect the climate in general (for this purpose, we would have performed idealised simulations that allow disentangling the effects of individual land cover changes by design), but to understand their contributions to the temperature response in a plausible future scenario with mixed, spatially heterogeneous land use/cover changes. To better communicate our aims, we revised the last paragraph of the introduction as follows:

This approach enables us to answer the following questions: (1) How do different implementations of European climate and biodiversity policies affect the PFT distribution across Europe? (2) At which scale and magnitude do the scenarios affect mean climate or extreme temperatures? (3) What are the underlying biogeophysical and ecosystem-related drivers? Finally, we assess if nature management focusing on biodiversity conservation, climate change mitigation, or cultural services could cause additional challenges for regional climate adaptation and mitigation. This study is, to our knowledge, the first to implement Nature Future scenarios in an ESM and to incorporate comprehensive ecological information in the translation of land systems to PFTs.

Major comments:

Methodological issues:

1. Model setup:

1) The authors used satellite phenology mode for CLM5 in the regional climate model COSMO-CLM. This mode prescribes monthly LAI and canopy height derived from MODIS for historical simulations. However, how did you prescribe future LAI and canopy height for the period 2034-2050? No matter how, there will be substantial uncertainties in future vegetation status even though the PFT covers are sophisticatedly derived from NFF. Furthermore, in the satellite phenology mode, BGC is fully off which means no interactive vegetation growth and carbon - nitrogen biogeochemistry and fertilization effects. In this context, your simulations can hardly capture future land-atmosphere interactions and the full biogeophysical and biogeochemical aspects of land use change effects on climate (see Bala et al., Zhu et al, Betts et al.).

We agree that this point requires additional clarification in the manuscript. We revised the methods by adding: *“CLM5 is configured in satellite phenology mode, i.e., for each grid cell and PFT, monthly leaf area index, stem area index, and canopy height are prescribed as a monthly climatology obtained from satellite data¹¹¹. This means that the PFT-specific vegetation structure varies seasonally and geographically according to observations, but it does not respond to simulated environmental conditions as in the prognostic biogeochemistry mode of CLM5. This simplification reduces computational costs (e.g., by turning off carbon and nitrogen cycling) and uncertainties related to the reliable prediction of vegetation structure. It omits potential changes in vegetation structure in a future climate (e.g., due to elevated CO₂ concentration and warmer temperatures), but this effect is likely small under SSP1-2.6 mid-century. Nevertheless, even in satellite phenology mode evapotranspiration and gross primary production are calculated as functions of leaf area and PFT-specific rates for stomatal conductance and photosynthesis, which both respond to environmental conditions including temperature, radiation, humidity, CO₂ concentration, day length, and availability of water and nitrogen^{110,111}. Surface albedo varies with leaf and stem area, PFT-specific optical properties (i.e., reflectance and transmittance of leaves and stems in the visible and near-infrared*

spectra), leaf orientation, solar zenith angle, and incoming direct and diffuse radiation from COSMO. Surface roughness is a function of leaf and stem area, canopy height, and aerodynamic parameters, and thus also varies in space and time."

We agree that the effects of future climate on LAI and canopy height would be important to consider in a high-forcing scenario like RCP8.5 at the end of the century, where we expect considerable effects of elevated CO₂ concentration and warming on vegetation structure. However, the differences between recent climate and the employed low-forcing scenario (RCP2.6) in the middle of the century are comparatively small, so the error from emitting the effect of this difference on vegetation structure is likely smaller than the prediction uncertainty in the prognostic biogeochemistry mode. Please note that the simplification resulting from satellite phenology mode only concerns the vegetation structure (i.e., LAI and canopy height), whereas effects of elevated CO₂ and climate change on stomatal conductance and photosynthesis rates are considered. Therefore, we believe that our analysis does not miss crucial aspects of land-atmosphere interactions.

2) Your climate forcing is only from one climate model "MPI-ESM1-2-HR" for the future period 2034-2040 and is not bias-corrected. In this case, there would be substantial uncertainty in the future climate. Even if fully bias-corrected, the many CMIP6 models still have a large spread in future climate projections for any given climate scenario. Therefore, most modelling studies in this field employ multiple CMIP6 models (often >10) as climate forcing so that you can show the ensemble mean and uncertainties, instead of a single value.

We appreciate the comment and it motivated us to evaluate differences across potential driving GCMs from CMIP6 for SSP1-2.6 during our simulation period. This evaluation is now included as Supplementary Notes and copied below. In the methods, we added the following justification for the choice of MPI-ESM1-2-HR: *"MPI-ESM1-2-HR is chosen due to good performance over Europe¹²³, medium equilibrium climate sensitivity, and high spatial resolution (~100 km for land and atmosphere). For these reasons, MPI-ESM1-2-HR is also used for CORDEX-CMIP6 simulations over Europe¹²⁴ and the CORDEX Flagship Pilot Study LUCAS."*

Supplementary Notes: Comparison of MPI-ESM1-2-HR with other CMIP6 models over Europe

To understand how MPI-ESM1-2-HR compares to other CMIP6 models used to drive regional climate simulations over Europe, we evaluate six global climate models (GCMs) selected for the EURO-CORDEX CMIP6 balanced matrix experiment¹⁵. Among EURO-CORDEX experiments, this experiment includes the largest number of GCMs and aims to cover a wide range of possible climate outcomes and uncertainties at acceptable computational cost. The included models are CMCC-CM2-SR5, CNRM-ESM2-1, EC-Earth3-Veg, MIROC6, MPI-ESM1-2-HR, and NorESM2-MM and we use the first available realisation for the SSP1-2.6 scenario. A common way to compare potential driving GCMs is to characterise them in terms of spatial and temporal means of near-surface air temperature and precipitation¹⁴, which are closely linked to the boundary fields used to drive regional climate models.

Our evaluation shows that MPI-ESM1-2-HR falls into the range of annual and seasonal mean values of temperature and precipitation simulated by the other GCMs over Europe in 2036-2050 (Supplementary Fig. 21). Only MIROC6 and CMCC-CM2-SR5 consistently project warmer annual and summer mean temperatures in all subregions. Compared with the other GCMs, MPI-ESM1-2-HR is usually neither the coldest, warmest, driest, or wettest model. The six GCMs together cover a wide

range of annual (~5 °C) and summer (~6 °C) mean temperatures across models and years, whereas a single GCM (here focusing on MPI-ESM1-2-HR) covers a range of ~1.5 °C across years (Supplementary Fig. 21). Interannual variation in MPI-ESM1-2-HR amounts to ~31% (24-48% in subregions) of multi-model interannual variation for annual mean temperature, and 56% (45-79% in subregions) for annual mean precipitation, while temperature and precipitation per model are weakly correlated. Interannual variation in temperature and precipitation of MPI-ESM1-2-HR is thus useful to test the robustness of the climate response under varying background climate, but it does not sample the full range of possible climate conditions.

A re-evaluation of the model spin-up showed that two years of spin-up are sufficient for the climate variables of interest to stabilise. Therefore, we discarded only two years for spin-up (2034-2035) and used 15 years (2036-2050) for analysis. This allows us to sample a greater range of interannual climate variability and to corroborate our results. Because total column soil moisture is not equilibrated after 2 years, we now use top (0-10 cm) soil moisture to assess soil moisture drying (results are similar in terms of direction and significance of effects).

Generally, we agree that many studies employ multiple CMIP6 models when they use secondary climate model output from community experiments coordinated under CMIP. However, we performed new climate model experiments with custom scenarios, for which secondary output is not readily available. We are not aware of other studies that used many different forcing datasets to perform new scenario experiments. Even under the comprehensive effort for regional climate downscaling, CORDEX, GCMs are carefully selected depending on the experiment purpose to limit computational costs. For instance, targeted experiments explore uncertainty due to SSP scenario, driving GCM, GCM internal variability, RCM, or RCM internal variability. These experiments use standardised protocols and do not incorporate specific scenarios like the ones explored here. The balanced matrix experiment, which aims to cover a wide range of possible climate outcomes and uncertainties by combining various GCMs and RCMs, includes six GCMs over Europe (i.e., the ones we included in our comparison). LUCAS, the CORDEX experiment focusing on land use change forcing, relies on MPI-ESM1-2-HR as the only driving GCM like our study, and explores differences among RCMs as the main source of uncertainty. In LUCAS phase 1, large inter-RCM differences have been found for extreme land use changes under historical conditions (Davin et al., 2020).

We believe that the RCM parameterisation is an important (probably the major) source of uncertainty, which we address in the discussion: *“The simulated effects agree with observational evidence of differences between cropland and grassland, showing that non-radiative mechanisms dominate and that temperatures are warmer over grasslands than rainfed and irrigated croplands in summer (and other seasons except winter) over Europe⁶³ and the US⁵⁵. However, results from another regional climate model indicated spatial variability and uncertainty about the sign of the annual mean temperature response to transitions between cropland and grassland⁴¹. Since model benchmarking and evaluation has mostly focused on forest cover change^{34,36,83} (and more recently also on cropland expansion and irrigation^{68,84,85}), more research is needed to corroborate the signal in ESMs and to represent the variety of cropland and grassland types. The ongoing second phase of the CORDEX Flagship Pilot Study LUCAS systematically investigates inter-model differences in the response to realistic land use changes across Europe and could provide new insights in this context.”*

We agree that bias-correction of GCM forcing data could potentially improve the performance of RCM simulations. However, it is common practice to drive RCMs with non-bias corrected GCM data (see CORDEX) due to inherent challenges in multi-variate bias-correction. Instead, RCM output variables are bias-corrected for specific applications in impact modelling (see ISIMIP). To our knowledge, bias correction of RCM boundary conditions is the exception not the norm (we are only aware of studies by Kim and colleagues) and most relevant for the simulation of extreme events and improved predictability according to the authors. In our study, we explore possible future scenarios (so predictability is not of major importance) and we investigate differences between scenarios under identical background climate (so all scenarios face the same forcing biases).

To understand the sensitivity of the climate response to the background climate in our simulations, we performed simple linear regressions at the grid cell level between the summer temperature response and the yearly background climate as simulated by MPI-ESM1-2-HR, regridded to the 0.1° regular grid of the RCM data. We find a significant relationship ($P < 0.05$) in very small parts of Europe, with average sensitivities of $-0.04 \text{ } ^\circ\text{C } ^\circ\text{C}^{-1}$ in NfN and NfS, and $0.03 \text{ } ^\circ\text{C } ^\circ\text{C}^{-1}$ in NaC. The highest sensitivities across grid cells and scenarios are -0.22 and $0.17 \text{ } ^\circ\text{C } ^\circ\text{C}^{-1}$, with temperature contributions of up to -0.54 and $0.44 \text{ } ^\circ\text{C}$ in the years with the greatest temperature anomalies in those grid cells. However, at the subregional scale, the highest contributions were $-0.018 \text{ } ^\circ\text{C}$ (18.3%) and $0.012 \text{ } ^\circ\text{C}$ (4.5%) in the NaC scenario, under yearly temperature anomalies of $-1.49 \text{ } ^\circ\text{C}$ in the East and $0.89 \text{ } ^\circ\text{C}$ in the South. Contributions at the European scale even were lower with up to $-0.005 \text{ } ^\circ\text{C}$ (4.1%) and $0.005 \text{ } ^\circ\text{C}$ (2.1%) in the NaC scenario. Effects of precipitation anomalies are hardly significant and weak. Based on this simple analysis, we do not expect major differences in the regional temperature response under the background climate simulated by the other potential driving GCMs. However, this analysis is greatly simplified compared with the multi-variate boundary conditions provided to regional climate models and their internal complexities. Therefore, we do not include this analysis in the revised paper or supplementary materials. Below are the results for the NaC scenario for illustration (not included):

Overall, we believe that the inclusion of additional simulation years corroborates the robustness of the climate response under varying background climate, and that the analysis of differences across potential driving GCMs helps place MPI-ESM1-2-HR as the only driving GCM in a wider range of possible (multi-variate) background climates.

2. Analysis:

1) In the analysis of climate effects, the authors mentioned six climate variables, but GPP and SM are eco-hydrological, not strictly climate variables. It is better to focus on temperature and precipitation

and other climate variables. GPP is useless in this context, especially given that it is output from the satellite phenology mode of CLM (directly related to prescribed LAI), which tells little about the carbon cycle. You can turn on the BGC mode and look at NPP, NEP or NEE, so that you can really talk about the climate service of NFF scenarios in term of their carbon cycle values such as carbon sequestration and carbon stocks.

We agree that it is better to focus on temperature and precipitation, so we kept summer temperature, TXx, and summer precipitation in the main text and moved detailed effects on GPP to the supplementary. We removed the wording “six climate variables”. We believe it is useful to keep the additional variables (soil temperature, wind speed, soil moisture, GPP) in the supplementary and to mention other relevant impacts in the main text, to avoid reducing climate to temperature and precipitation. For instance, the essential climate variables of the WMO consider “climate variables” in the broadest sense.

We agree that GPP can only serve as a proxy of carbon sequestration, and we clarify this in the discussion: *“Reduced vegetation carbon uptake ($-11 \pm 0.4\%$) suggests lower potential for ecosystem-based mitigation, but dedicated simulations would be needed to quantify the net response of the land carbon cycle (i.e., additionally considering respiration, disturbances, and soil carbon dynamics).”* Although increased GPP does not guarantee increased (net) sequestration, it is a necessary condition for carbon accumulation in ecosystems (particularly aboveground biomass) as included in the scenarios of Dou et al. (conversely, reduced GPP is unlikely to result in increased net sequestration). In CLM5, GPP is directly related to LAI in both SP and BGP mode, and in both modes the photosynthetic rate per unit leaf area responds to environmental conditions (see previous comment). The only difference lies in whether LAI is prescribed or simulated, and the prescribed LAI might be less uncertain than the predicted LAI, as explained before. As a proxy for carbon sequestration, the GPP output from our simulations is qualitatively similar to maps of carbon sequestration potential in aboveground biomass, as employed by Dou et al. and many other studies that do not use climate models. Therefore, we believe it is useful to report the simulated GPP and communicate its limitations.

2) In the analysis of drivers, there are two issues. First, the regression mixed physical variables (albedo, EF, and roughness length) from NAC, NFN and NFS scenarios with climate variables (air temperature, precipitation, and wind speed) from the reference scenario (SSP1) as confounding factors. The validity of this approach is suspicious, as can be seen from the divergent/unstable results presented in Fig. 5a, Fig. S7a and Fig. S9a (and low R2 in some regions and scenarios). The regression should use climatological means of temperature, precipitation and wind (and specific humidity, also important) from the historical period (ERA5) or the SSP1-2.6 forcing data of MPI-ESM1-2-HR to represent climate background effects. Instantaneous values of these variables from COSMO-CLM SSP1 are not appropriate and could introduce bias to the regression analysis of NAC, NFN and NFS scenarios. More importantly to validate this approach, the authors should present a sensitivity analysis, that is, using your regression coefficients to show temperature responses to unit differences or unit changes in predictor variables under each scenario. We should expect similar feature importance rank and sign and size of effect of each variable among different scenarios (that is, the sensitivity of air temperature to each relevant predictor/driver should be relatively stable across these land use scenarios). If the current plots show total effect, instead of unit effect, please clarify. Please also clarify if you differenced NAC, NFN and NFS from SSP1 and used the differences in each

variable in the regression analysis. You could also focus on the 1% most affected areas to do sensitivity analysis. If the sensitivity plots are not consistent among scenarios, the authors should consider alternative attribution approaches (see below).

Second, the two predictor sets used in ridge regression include intrinsic physical variables (albedo, EF, and roughness length; Fig. 5a) and upstream surface properties (PFT fractions, vegetation height, LAI, and irrigated area) together with vegetation characteristics (broadleaf tree, needleleaf tree, grass, and crop; Fig. 5b). The second set used for Fig. 5b are highly correlated and not suitable to be mixed in regression; they can all be decomposed to the three intrinsic variables in Fig. 5a (and unconsidered ones like photosynthetic rate, stomatal conductance, water use efficiency, C:N ratios, etc.). Although ridge regression can partly mitigate multicollinearity, they cannot fully separate the contributions of these highly correlated variables and properly infer causality as you mentioned (lines 406–407). Therefore, analysis for Fig. 5b and related ones are not meaningful. Furthermore, ridge regression assumes linearity, while temperature and dynamics are highly nonlinear (e.g., the effect of EF on temperature may exhibit threshold effects). The authors can consider using machine learning methods (e.g., Random Forest, XGBoost) to capture nonlinear relationships, or combine them with causal inference methods, (e.g., Structural Equation Modeling (SEM)), to quantify the independent contributions of each driver.

Finally, I recommend the authors to consider using robust physically based attribution methods to quantify the contribution of intrinsic physical variables related to land-use change (such as changes in albedo, EF, roughness, or surface and aerodynamic conductance, or ET/latent and sensible heat fluxes) to surface climate (see Betts et al. 1997, Zeng et al. 2017, He et al. 2020, Chen & Dirmeyer (also used CLM)).

Thank you for this critical evaluation. Due to the criticism raised, we decided to split the analysis of drivers into two parts (1) contributions of surface energy balance components (relying on surface temperature decomposition); and (2) contributions of individual land cover transitions (relying on an improved version of the ridge regression approach used before). We believe this 2-part approach reduces uncertainty where possible (i.e., by using a robust physics-based approach for surface energy balance components, as you suggested) but also goes further by disentangling effects of land cover transitions (i.e., the initial forcing in the scenarios) on near-surface air temperature.

The ridge regression approach was modified as follows: we only consider PFT fractions as possible drivers, no other vegetation-related predictors such as LAI to avoid multicollinearity; we do not use PFT fractions directly but compute PFT transitions per grid cell to avoid multicollinearity (e.g., if the main transition is grass to crop, then grass and crop fractions are necessarily correlated and the regression will not be able to disentangle the effects); we evaluate the effect of auxiliary spatial predictors but do not include them in the final model formulation as documented in the methods (see copied section below). As background temperature and precipitation we used climatological means (not instantaneous values). Whether background climate variables should be taken from MPI-ESM1-2-HR or the SSP1 reference is a question of trade-offs: The GCM variables are independent of land cover effects (which are not the major determinant of climate gradients across Europe) but have much coarser resolution than the RCM variables from the SSP1 reference and thus lack topographical effects (which are quite important for climate gradients across Europe). Therefore, we believe that

RCM variables from the SSP1 reference are likely the better choice, but we tested both and it did not substantially affect the estimated temperature sensitivity to land cover changes.

The comments indicate that the presentation of results in terms of feature importance (i.e., °C per standard deviation of the predictor) was not clear, likely because feature importance combines sensitivity and contribution into a single metric. To improve clarity, we now show separately the temperature sensitivity using the unstandardised regression coefficients (i.e., °C per unit (here percent) land cover transition) and the contribution of land cover transitions to the temperature response (i.e., °C obtained by multiplying sensitivity and actual percentage transition). We also analysed the stability of the unstandardised regression coefficients across scenarios and subregions (Supplementary Fig. 10).

Below we copy the new parts in the results section, which are accompanied by new parts in the methods and discussion sections:

Results

Drivers of the temperature response

To investigate the causes of the simulated temperature responses, we focus on the summer season and decompose the mean (2036-2050) surface temperature differences into contributions of surface energy balance components (Methods). Surface temperature differences may be caused by changes at the land surface (with increases in latent heat flux, sensible heat flux, ground heat flux, and albedo cooling the surface temperature) or feedbacks from the overlying atmosphere (with increases in downwelling shortwave and longwave radiation warming the surface temperature). Surface (skin) temperature further influences the near-surface air and is thus a useful proxy for the 2 m air temperature response. Regional summaries are presented in Fig. 7 and maps in Supplementary Figs. 6-8.

The NaC scenario shows an important warming contribution from reduced latent heat flux (Fig. 7c). This reduction in latent heat flux is consistent with lower surface roughness due to lower canopy height, and with lower evaporative fraction due, e.g., to lower leaf area. The warming effect is partly compensated by increased sensible heat flux, ground heat flux, and albedo. The net surface warming is enhanced by strong atmospheric feedbacks (primarily increased downwelling shortwave radiation resulting from reduced humidity and cloudiness) at the European scale and in the West, East, and South subregions. The NfN and NfS scenarios show warming contributions from declines in both latent and sensible heat fluxes at the European scale and in most subregions, partly compensated by cooling from increased albedo (Fig. 7a-b). The net surface warming is outweighed by cooling from atmospheric feedbacks over EU+ and South, resulting in cooler surface temperatures. The South shows an important cooling contribution from reduced downwelling longwave radiation resulting from reduced atmospheric emissivity and temperature.

These results suggest that the summer temperature differences between the scenarios and SSP1 are more strongly influenced by atmospheric feedbacks (predominately changes in downwelling shortwave radiation) than surface effects, especially in the NaC scenario. Surface effects are dominated by reduced turbulent energy transfer to the lower atmosphere (i.e., through latent and sensible heat fluxes), which is in line with the reduced surface roughness in all scenarios and subregions. Nevertheless, the relative importance of drivers varies spatially and seasonally,

depending on the pattern of PFT changes, local and seasonal PFT properties, and environmental conditions. For instance, the most warming areas show surface warming and reinforcing atmospheric feedbacks of similar magnitude. In winter, surface effects are dominated by albedo changes, and atmospheric feedbacks by changes in downwelling longwave radiation (not shown).

Fig. 4. Contributions of surface energy balance components to the surface temperature response in summer. Results are shown for Europe, the subregions, and the 1% most cooling (min) or warming (max) areas in terms of near-surface (2 m) air temperature, for **a**, the NfN scenario, **b**, the NfS scenario, and **c**, the NaC scenario. Contributions at the land surface include latent heat flux (LH), sensible heat flux (SH), ground heat flux (G), and albedo, whereas those of atmospheric feedbacks are given by downwelling shortwave radiation (SWdown) and downwelling longwave radiation (LWdown). The sum of surface and atmospheric contributions are shown as dots. The grey bar indicates the sum of all contributions, which is similar to the simulated surface (skin) temperature response (-) and linked to the near-surface (2 m) air temperature response (+). Bars and markers show the 2036-2050 mean and error bars extend to the 95% confidence interval across years. Supplementary Figs. 6-8 show maps of the components for each scenario.

Contribution of individual land cover transitions

To assess which PFT changes are important for the temperature response in each scenario, we focus again on the mean (2036-2050) summer temperature and estimate the contributions of individual land cover transitions. After computing net transitions between seven land cover categories per grid cell (i.e., tree needleleaf, tree broadleaf, shrub, grass, crop rainfed, crop irrigated, and bare, with PFT variants aggregated like in Fig. 5), we fit a linear regression between the temperature response and

20 different net transition types across grid cells (Methods). This allows us to extract the temperature sensitivity to individual land cover transitions at the grid cell scale (i.e., local effects) from scenario simulations in which multiple PFTs are changed simultaneously. Conceptually similar approaches have been used to extract land cover change signals from simulations with multiple forcings^{41,72}.

At the European scale, the NfN and NfS scenarios show important contributions to cooler summer temperatures from transitions of needleleaf tree and shrub PFTs to rainfed crop and grass PFTs (Fig. 8a,b). Warming contributions are linked to transitions of rainfed crop to grass PFTs and bare soil, and broadleaf tree to grass PFTs. The NaC scenario shows similar effects, but the temperature response is dominated by a much stronger warming contribution from transitions of crop to grass PFTs (Fig. 8c). Differences between scenarios and subregions result primarily from the scale of individual transitions. For example, in the NaC scenario in the South transitions of irrigated crop to grass PFTs are associated with substantial warming, whereas in the other scenarios and subregions irrigated crops are negligible. The estimated temperature sensitivity (i.e., the temperature response per percent transition) is broadly consistent across scenarios. In all scenarios broadleaf tree and irrigated crop PFTs are associated with the strongest cooling, followed by rainfed crop and grass PFTs, and finally needleleaf tree and shrub PFTs which only show cooling effects in transitions from bare soil (Supplementary Fig. 10).

Regressions performed at the subregional scale can result in varying sensitivity estimates, particularly for transitions between tree and non-tree PFTs. For instance, transitions of broadleaf tree to rainfed crop PFTs are linked with cooling in the North and warming in the South in all scenarios, and with diverging and often uncertain effects in the West and East (Supplementary Figs. 11-14). This might be due to differences in incoming shortwave radiation and water availability between North and South and within the West and East subregions. Such differences in background climate influence the relative importance of changes in albedo vs. evapotranspiration efficiency for temperature impacts, which also explains latitudinal differences in the response to deforestation (i.e., cooling in high latitudes and warming in the tropics)⁷³. Although we focus on near-surface air temperature, we note that temperature sensitivities and contributions of individual land cover transitions are similar in relative terms for surface (skin) temperature but associated with lower uncertainty (Supplementary Fig. 15).

Fig. 5. Temperature sensitivity to individual land cover transitions and contribution to the 2 m air temperature response in summer (2036-2050 mean) over Europe. Sensitivities are given by the unstandardised regression coefficients and contributions by the sensitivities multiplied by the percentage change over the region. Results are shown for **a**, the NfN scenario, **b**, the NfS scenario, and **c**, the NaC scenario. For each scenario, land cover transitions are shown that occur on >0.2% of the area and in >20% of grid cells. Land cover transitions are named by the dominant direction per scenario (e.g., shrub to crop rainfed in NfN, crop rainfed to shrub in NaC) and sorted by strength of effect. Bars show the mean across 200 linear regression fits and error bars extend from the 2.5th to the 97.5th percentile of the bootstrap sample. Results for subregions are shown in Supplementary Figs. 11-14 and results for surface skin temperature in Supplementary Fig. 15.

Methods

Analysis of drivers

To investigate how land cover changes affect the simulated temperature, differences in surface (skin or radiometric) temperature can be attributed to components of the surface energy balance^{54,107,124}. The surface energy balance equation states that energy inputs from net radiation (left hand side) are balanced by energy outputs (right hand side):

$$\underbrace{(1 - \alpha)SW_{down}}_{net\ SW} + \underbrace{LW_{down}}_{net\ LW} - \underbrace{\varepsilon\sigma T_s^4}_{turbulent\ fluxes} = \underbrace{LH + SH}_{turbulent\ fluxes} + G \quad (1)$$

where α is surface albedo, SW shortwave radiation (down for downwelling), LW longwave radiation, ε surface emissivity, σ the Stefan-Boltzmann constant, T_s surface temperature, LH latent heat flux, SH sensible heat flux, and G ground heat flux. By taking the partial derivative of Eq. 1 and assuming constant surface emissivity, the surface temperature response can be expressed as:

$$\Delta T_s = \frac{1}{4\varepsilon\sigma T_s^3} \left(\underbrace{-\Delta LH - \Delta SH - \Delta G - SW_{down}\Delta\alpha}_{surface\ contribution} + \underbrace{(1 - \alpha)\Delta SW_{down} + \Delta LW}_{atmospheric\ feedbacks} \right) \quad (2)$$

where $\frac{1}{4\varepsilon\sigma T_S^3}$ ($K Wm^{-2}$) is the surface temperature sensitivity. Following Luysaert et al.¹⁰⁷, we summarize the cooling effects of in latent heat flux, sensible heat flux, ground heat flux, and albedo as direct contribution from changes at the land surface, and the warming effects of downwelling shortwave and longwave radiation as indirect contribution caused by changes in the overlying atmosphere. Possible mechanisms behind atmospheric feedbacks include changes in atmospheric humidity, cloud cover, temperature, and emissivity. The inputs to Eq. 2 are obtained from model output, except for ε which is set to 1 and G which is calculated as the residual of the surface energy balance per grid cell (by definition, the energy balance is closed in COSMO-CLM²).

Analysis of the contribution of individual land cover transitions

To assess the contribution of PFT changes to the simulated temperature response, we aggregate the PFTs into seven categories: tree needleleaf (including evergreen temperate, evergreen boreal, and deciduous boreal), tree broadleaf (including evergreen temperate, deciduous temperate, and deciduous boreal), shrub (including broadleaf evergreen temperate, broadleaf deciduous temperate, and broadleaf deciduous boreal), grass (i.e., herbs excluding annual crops, including C3 arctic, C3, and C4), crop rainfed, crop irrigated, and bare ground. PFT changes are enabled by opposing changes in another PFT, and the temperature response depends on both PFTs involved in a land cover transition (e.g., grass replacing tree PFTs). However, in fractional PFT maps and in the COSMO-CLM2 model, PFTs do not occupy specific parts of grid cells, so it is not clear which PFT replaced another PFT. To resolve this, we compute net transitions per grid cell iteratively using a greedy approach that matches the most increased and decreased land cover categories in each iteration until all PFT changes are captured. Net transitions are named after the dominant direction per scenario across Europe (e.g., if crop to grass transitions are larger than grass to crop, the net transition is named crop to grass). This results in 20 different net transition types, because transitions between rainfed and irrigated crops occur in none of the scenarios.

To estimate the sensitivity of the local (i.e., grid cell scale) temperature response to each of the $m = 20$ possible land cover transitions, we fit a multiple linear regression across $i = 1, 2, \dots, n$ grid cells in the EU+ region or within subregions:

$$\Delta T_i = \beta_0 + \beta_1 x_{i1} + \beta_2 x_{i2} + \dots + \beta_m x_{im} + \varepsilon_i \quad (3)$$

where ΔT is the temperature response (i.e., the difference between a scenario and the SSP1 reference), x_p is the fraction of land cover transition p , and β_p is the corresponding temperature sensitivity. Conceptually, this approach attributes the local temperature response to land cover transitions in the same grid cell, whereas non-local effects originating from land cover changes in other grid cells contribute to the grid cell-specific error ε or the intercept β_0 if the effects are spatially uniform.

Linear regression models with many predictors often suffer from multicollinearity (i.e., two or more predictors are correlated), resulting in erroneously high coefficient estimates. To prevent this, we employ ridge regression, a variant of multiple linear regression that includes a regularisation term in the calculation of the residual sum of squares to shrink coefficients in proportion to their initial size and effectively penalise high coefficient estimates. To determine an appropriate regularisation strength per model fit, we test logarithmically spaced values between 0.1 and ~ 2000 and perform cross-validation to choose the best value. Mean coefficients and their confidence intervals are

estimated based on 200 iterations of the pipeline for model fitting and evaluation (i.e., train-test split, feature standardisation, and ridge regression with cross-validation for hyperparameter tuning). The 200 models are trained and evaluated using spatial block cross-validation to increase the independence of data used for training (75% of blocks) and testing (25% of blocks). For spatially structured data, blocked sampling yields higher but more realistic error estimates¹²⁵. In our application, blocking increases the confidence interval (calculated as 2.5th to 97.5th percentiles of the bootstrap sample) while the means are robust and insensitive to the chosen block size.

We note that the inclusion of spatial predictors like latitude and longitude or background temperature and precipitation improves model performance scores (e.g., mean absolute error, R^2), but hardly affects the contribution of individual land cover transitions. Since we focus on explaining contributions and not on predictions, and a clear interpretation of such independent effects is missing, we do not include additional predictors in the final model formulation.

Discussion

Turbulent fluxes and atmospheric feedbacks drive the surface temperature response in summer

Our results suggest that the temperature response in summer is strongly influenced by changes in turbulent fluxes at the surface and atmospheric feedbacks. In the NfN and NfS scenarios warming from reduced latent and/or sensible heat flux is outweighed by cooling from atmospheric feedbacks, resulting in cooler surface temperatures. In contrast, the NaC scenario shows warmer surface temperature because the net warming from reduced turbulent fluxes is reinforced by atmospheric feedbacks. The strong influence of atmospheric feedbacks could have important implications for regional land use management, because atmospheric feedbacks can affect nearby or remote areas (i.e., non-local effects)^{65,66}.

Surface temperature decomposition is a robust physics-based approach, but we note some caveats that should be considered in the interpretation of the results. First, the considered surface energy balance components are not independent due to process coupling (e.g., energy availability for latent and sensible heat flux is influenced by downwelling shortwave radiation and albedo) and the underlying biogeophysical mechanisms cannot be separated. Alternative methods focus on disentangling the effects of changes in surface albedo, evaporative fraction, and roughness length, but assume no atmospheric feedbacks^{76,77}, independent effects⁷⁸, or linear effects³⁶. Second, surface temperature and near-surface air temperatures are tightly coupled and their responses correlated (>0.8 in more than 90% of grid cells, with lower correlations over rough terrain, see Supplementary Fig. 9). However, air temperature responses are typically weaker (see Fig. 7 and refs^{55,79-81}) and influenced more strongly by atmospheric processes. This might explain the different sign of surface and near-surface air temperature response in the East subregion in NfN and NfS.

Two land cover transition types contribute most strongly to cooler and warmer summer temperatures

Regressions over Europe reveal that the small cooling in NfN and NfS is mostly explained by transitions of needleleaf tree to crop PFTs, due to high temperature sensitivity and high spatial coverage particularly in the West. This cooling effect could be due to higher albedo and evapotranspiration of crop PFTs during the growing season in the model, but there is contradictory evidence from models and observations on the summer temperature difference between needleleaf

forests and croplands^{55,63,64,74,82}. Nevertheless, the estimated cooling effect is consistent in our regression models across scenarios, most subregions, and for both temperature variables (i.e., near-surface air and surface skin temperature).

The strong warming in NaC is mostly explained by widespread transitions of crop to grass PFTs, which dominates the temperature response in all subregions. Based on spatial patterns and simple correlations, we speculate that the warming is due to lower evaporative cooling of grass PFTs in the model, which in turn could result from lower leaf area index and vegetation height, and from the lack of irrigation in the South. The simulated effects agree with observational evidence of differences between cropland and grassland, showing that non-radiative mechanisms dominate and that temperatures are warmer over grasslands than rainfed and irrigated croplands in summer (and other seasons except winter) over Europe⁶³ and the US⁵⁵. However, results from another regional climate model indicated spatial variability and uncertainty about the sign of the annual mean temperature response to transitions between cropland and grassland⁴¹. Since model benchmarking and evaluation has mostly focused on forest cover change^{34,36,83} (and more recently also on cropland expansion and irrigation^{68,84,85}), more research is needed to corroborate the signal in ESMs and to represent the variety of cropland and grassland types in Europe.

Results and discussion:

1) The main findings appear to be theoretical inferences based on existing eco-climatological and physical mechanisms (e.g., line 187: "We expect warmer temperatures from albedo decrease, EF decrease, and roughness decrease"), rather than causal relationships that cannot be established by ridge regression itself (e.g., lines 190 -191: EF is the dominant predictor for cooler temperatures in all scenarios and subregions; line 193: albedo is also mostly linked to cooling as expected; line 207: the decline in irrigated area contributes strongly to warmer temperatures). The strong signal of EF is apparent, but the effect of albedo is far from clear from the Fig. 5a, Fig. S7a and Fig. S9a (inconsistent signs and sizes of albedo effect across regions and scenarios). The authors explain that some discrepancies exist because albedo differences are very small in summer. But the differences are even smaller under NFN and NFS scenarios than under NAC, then why can you derive that albedo is more important in Alpine subregion in NFN and NFS? Although the seemingly important role of albedo in Alpine regions is expected, which is consistent with the dominant role of albedo in high-latitude regions (similarly high snow cover to Alpine regions), the unstable performance of the regression analysis across regions and scenarios currently prevents a robust understanding of the drivers behind temperature changes and warrants further refinement of the analysis (see above).

Considering these concerns, we revised the analysis of drivers as described under the previous comment.

2) The study analyzed land use change effects on annual maximum temperatures and precipitation. The Discussion could briefly consider heat stress and droughts, particularly in Southern Europe, where such impacts could be severe. This would enhance the study's relevance to climate adaptation and risk management. Readers will wonder the current small effects on temperature and precipitation may not incur significant ecological and social-economic damages compared to the reference scenario. Moreover, the effect of irrigation on air temperature is only the tip of iceberg of

heat impact (irrigation may increase heat stress due to increased humidity which is currently not included in the analysis; see Mishra, V. et al, Lobell et al).

Thank you for the suggestion. We now evaluate warming and drying effects in the hottest/dryest and most heating/drying areas of Europe, and emphasise the importance of further warming and drying in these areas as well as in Southern Europe.

We added the following to the results: *“Significant warming can be observed in the East (0.25 ± 0.08 °C), West (0.24 ± 0.06 °C), and South (0.18 ± 0.05 °C) subregions, which show strong PFT differences (Fig. 5). The warming reaches 0.68 ± 0.05 °C in the most affected areas. The hottest areas under SSP1 would experience 0.23 ± 0.08 °C temperature rise, but the most heating areas under SSP1 would not get significantly warmer. The warming effect is particularly strong during the hottest times of the years (TXx $+0.33 \pm 0.14$ °C) and reaches $+1.86 \pm 0.29$ °C in the most affected areas. In contrast, the NfN and NfS scenarios show weaker and spatially more heterogeneous temperature responses (Fig. 4a-b) owing to weaker and more heterogeneous PFT differences. Temperatures do not differ significantly from those under SSP1 at the European scale. A small yet robust cooling of summer temperatures can be observed in the South and in the most heating areas under SSP1. Nevertheless, locally in the most affected grid cells, hot temperatures can increase by roughly 1 °C (TXx $+0.98 \pm 0.33$ °C in NfN and $+1.06 \pm 0.50$ °C in NfS).”*

And in the discussion: *“Overall, our results show that the NaC scenario would have the strongest detrimental effects on climate. Higher annual maximum temperatures, higher summer temperatures, and lower summer precipitation under NaC than SSP1 indicate trade-offs with local climate impacts and the resulting need for adaptation. The additional warming and drying compared with recent climate could further threaten local biodiversity and human well-being, particularly in hot spot areas for heat stress and droughts in Southern Europe but also in the West and East. A detailed assessment of losses in species climatic range size (see refs.^{9,86}) or health impacts (see refs.^{87,88}) was, however, outside the scope of this study.”*

We did not calculate a heat stress metric in this study. The main driver of warmer temperatures is conversion of irrigated cropland to grassland in the NaC scenario, so if anything, our results overestimate the severity of heat impacts because humidity declines. However, as acknowledged in the discussion, this is outside the scope of our study.

3) The authors state that precipitation changes at the subregional scale are driven by non-local effects of atmospheric moisture transport (lines 154–158). Although relevant studies were cited in support of this claim, this mechanism is not directly analyzed in the study. Therefore, it should be framed as a hypothesis rather than a confirmed conclusion. It is recommended that the Discussion section should consider alternative factors, including: (1) Scale effect – vegetation-precipitation or ET-climate feedbacks may be weak at small local scales but stronger at regional scales; (2) Time-lag effects - the impact of PFT changes on precipitation may have seasonal delays.

We agree with this view and rephrased for clarification as follows: *“The precipitation response is spatially heterogeneous (Fig. 4c) and not statistically significant at the grid cell level. Robust effects at the subregional scale might be explained by non-local effects due to atmospheric moisture transport and precipitation recycling at larger spatial scales^{68,69}.”*

The reason for precipitation recycling at larger spatial scales could be scale effects, but this is not analysed in our study so we did not hypothesise further. The statement refers to differences between grid cell and subregional scales, so we focus on spatial aspects (and not temporal shifts such as those due to time-lagged effects).

4) Your manuscript mentioned “biodiversity” and its tradeoff with climate goals many times, but your current methods couldn’t quantify the effect on biodiversity under different scenarios. Merely saying “additional warming and drying could threaten biodiversity” (e.g., lines 238-240) is insufficient without quantitative analysis of biodiversity metrics and climate-biodiversity relationships (and the projected changes in temperature and precipitation are very small). There are relevant works that have quantified the tradeoffs between biodiversity conservation and climate mitigation services under scenarios of nature-based solutions, which are not cited yet (Smith et al; Fastré et al.).

Thank you for these suggestions. We included the recently published study by Smith et al. as an example of a study that quantified climate feedback effects on biodiversity, which were triggered by mitigation. We also clarified that this step was outside the scope of our study: *“The additional warming and drying compared with recent climate could further threaten local biodiversity and human well-being, particularly in hot spot areas for heat stress and droughts in Southern Europe. A detailed assessment of losses in species climatic range size (see refs.^{9,86}) or health impacts (see refs.^{87,88}) was, however, outside the scope of this study.”*

Fastré et al. quantified trade-offs in using land for biodiversity conservation vs. water-related ecosystem services (e.g., water stress), but they do not quantify how changes in water stress feedback on biodiversity. In that sense, their approach is similar to Dou et al. who focus on either biodiversity conservation or climate mitigation, but not their interactions.

5) The Discussion currently focuses on statistical correlations without deeply analyzing underlying physical mechanisms. Therefore, it is recommended that the authors consider causal inference or physically based attribution methods and expand discussion on the physical mechanisms driving temperature changes (there is a broad literature to compare). Note that the statement in line 212 “reduced turbulence (negative correlation with roughness)” is wrong.

Considering these concerns, we revised the analysis of drivers as described under an earlier comment and specifically added a physics-based attribution method.

The statement in line 212 was meant differently, i.e., the negative correlation between LAI and roughness explains reduced turbulence and thus warmer temperatures, everything else being equal. However, since LAI is not a predictor anymore in the revised ridge regression approach, we removed this part from the manuscript.

6) In the Discussion, the authors propose additional indicators to assess the value of different NFF scenarios (lines 274–278), which aligns closely with existing ecosystem service valuation frameworks. The authors could cite relevant literature on established ecosystem valuation systems, such as The Economics of Ecosystems and Biodiversity (TEEB), and Natural Capital Accounting (NCA), and discuss

how the NFF framework can leverage these approaches to enhance its practical applicability and policy implications.

Thank you for the suggestion. We expanded the discussion on the scenarios, their implications and limitations in response to reviewers 1 and 3. In this context, we included references to ecosystem service quantification in landscape planning: *“To support landscape planning in Europe, spatial optimisation techniques can be used to develop scenarios that maximise target variables (e.g., biodiversity and carbon sequestration¹⁰³) under given constraints. Such methods can be applied to co-optimize the delivery of various ecosystem services (see, e.g., refs.104,105). Particularly, decision-making towards just and sustainable futures could be improved by integrating diverse nature values (i.e., market-based instrumental values balanced with intrinsic, non-market instrumental, and relational values) and valuation methods (i.e., ways to quantify relevant indicators) is still challenging¹⁰⁶.”* Ref 106 (Pascual et al. 2023) synthesises findings from the IPBES report on the assessment of the diverse values and valuation of nature, including non-instrumental values and non-monetary valuation methods.

Minor comments:

Line 103: no need to cite unpublished work.

The relevant dataset has been published in the meanwhile and is now cited.

Lines 119-120: this is surprising. Nature conservation scenarios should normally promote tree diversity instead of crop expansion.

We agree that this sounds counterintuitive, so we expanded the first results section to explain the underlying reasons (see **point B** for improvements regarding the description of scenario effects on land systems and PFTs). Specifically, there are two reasons why crop PFT cover increases in NfN and NfS compared with the SSP1 reference:

1) Cropland declines in all scenarios relative to recent conditions (year 2015), but the declines are smaller under NfN and NfS than under the SSP1 reference. Compared with SSP1, NfN and NfS show stronger extensification than intensification, so that crop PFT cover increases. We clarified this in the text: *“NfN and NfS foresee only mild intensification of mid-intensity croplands, so that crop PFT cover increases because of the greater co-occurring extensification (yet cropland declines compared with recent land cover).”*

2) PFT cover changes in the scenario can be caused by multiple drivers, so the results are not always intuitive (e.g., intensification to compensate for increases in natural ecosystems or expansion to compensate for extensification). Cropland expansion is possible in the NfN scenario in ways compatible with species conservation, particularly if species can be protected in other (more valuable) areas.

We also added further clarification in the discussion: *“The scenario outcomes reflect different processes across Europe, which can be opposing due to contextual differences or compensatory effects. Particularly at the local scale, changes can be difficult to interpret or seem counterintuitive. The results are path-dependent and complex, and the link to the initial scenario assumptions is not always evident (e.g., NfN projects cropland expansion to compensate for extensification, NfS projects forest decline enabled by intensification). Such emergent outcomes allow a realistic representation of*

how land systems may change across Europe and deviate from overall non-spatial policy targets and visions. Considering the implemented land system allocation rules, it is unlikely that the scenarios outcomes go against their premises. However, the scenarios are driven by policy means and not their outcomes, so they are not necessarily efficient at achieving policy goals (e.g., NfS promotes forests in areas with high carbon sequestration potential but does not maximise carbon sequestration). An in-depth evaluation is outside the scope of this study and the reader is referred to Dou et al.³³ for details on the scenarios and their limitations."

Line 185: "emergent" is a popular word, but it should refer to properties not directly apparent or arising from complex interactions between simpler components. Are these variables mentioned really emergent characteristics? At least, albedo is an intrinsic physical property, but not emergent.

This definition of "emergent" indeed applies to the presented variables. We calculated them as follows: albedo = upwelling/downwelling solar radiation, evaporative fraction = latent heat/net radiation, and roughness length = function of PFT or snow fraction and their respective canopy height, leaf area index, stem area index, and aerodynamic parameters. None of the three variables is prescribed in the model, but they are calculated dynamically depending on environmental conditions, PFT fractions and PFT properties in each grid cell. While intrinsic albedo (bihemispherical reflectance, used e.g. in scattering equations) is an intrinsic physical property, apparent albedo (calculated here and relevant for the surface energy balance) depends on radiation scattering in the atmosphere. In CLM (and COSMO-CLM2), apparent albedo depends on leaf and stem area, PFT-specific optical properties (VIS and NIR reflectance and transmittance), leaf orientation, solar zenith angle, and incoming direct beam and diffuse radiation (depending on atmospheric scattering in COSMO).

Line 202: It is unclear whether this is your hypothesis or the analysis outcome. If you already knew it would not perform well, why still including all these inter-correlated variables?

Considering these concerns, we revised the analysis of drivers as described under an earlier comment and limited the ridge regression to non-correlated predictors. Generally, less good performance of regression models in terms of, e.g., mean absolute error or R^2 does not necessarily mean that the model has no value in explaining contributions. We agree it would not be very useful for predictions, but this is not our application. In the revised application, we use predictors that we believe work well in a linear model (e.g., CLM5 calculates the grid cell level temperature as a linear combination of PFT-specific temperatures weighted by their coverage).

Line 212: surface turbulence should be positively correlated with roughness.

This statement was meant differently, i.e., the negative correlation between LAI and roughness explains reduced turbulence and thus warmer temperatures, everything else being equal. However, since LAI is not a predictor anymore in the revised ridge regression approach, we removed this part from the manuscript.

Line 228: NAC highly values agricultural landscape. Then, why there is large conversion of cropland to grassland in this scenario?

Several reviewer comments indicate that there were unclarities in our description of the scenario design and how the scenarios affect land systems, habitat types, and eventually PFTs. We therefore

improved the introduction and added a new figure (Fig. 1), see **point A**. Furthermore, we improved the description of scenario effects in the first results section and added Fig. 3a to show changes in land systems, habitat types, and PFTs, see **point B**.

Agricultural landscapes include croplands, grasslands (including pastures), and mosaic systems. For the NaC scenario, we now explain in the introduction that *“cultural value [is represented] by agricultural heritage landscapes (i.e., rural areas characterised by traditions, low management intensity, landscape heterogeneity, and high value and meaning)⁵⁴.”* In the land systems maps by Dou et al., the NaC scenario shows cropland decline, grassland increase, and mosaic increase. We now explain in the first results section that *“the NaC scenario focusing on cultural landscapes foresees strong intensification and diversification (i.e., conversion to mosaic systems) of mid-intensity croplands and thus an overall cropland decline (Fig. 2a). Mosaic systems consist of various habitats including grassland, shrubland, forest, and man-made types (Supplementary Table 1), which in turn consist of various PFTs (Supplementary Table 2). As a result, we find reduced crop PFT cover compared with the reference and more strongly increased grass PFT cover than in the other scenarios (Fig. 2c).”*

Lines 238-239: it is unconvincing that such small additional warming and drying compared with recent climate could further threaten local biodiversity and human well-being over large parts of Europe. Only in 1% of most affected grid cells, there could be some additional climate impacts when warming exceeds 0.5 degC (lines 220-221)

Spatial aggregation (here regional averages) and temporal aggregation (here multi-year seasonal averages across the diurnal cycle) dampen the signal size, so that the reported values seem small. We believe that the community is aware of this effect and that reporting average values is useful to avoid exaggerating the results (e.g., several degrees difference can be found by taking the highest seasonal values in individual years and grid cells, but this would concern only a small area and period). Nevertheless, we agree that it is useful to highlight where strong effects occur in each scenario. Therefore, we revised the colourbars of the map plots (e.g., Fig. 4) to show a small range around zero in white, thereby masking small cooling/warming and highlighting stronger differences (same for drying/wetting). The choice of the white range is arguably arbitrary (yet very common), but this solution avoids the need to define a temperature threshold above which warming starts to matter (e.g. 0.5 °C). We believe there is no evidence for a single threshold that would apply everywhere across Europe for all kinds of impacts (e.g., human health, plant and animal species, etc.). Moreover, the warming simulated in our scenarios would be on top of the warming under SSP1 relative to pre-industrial conditions.

We modified the discussion as follows: *“Overall, our results show that the NaC scenario would have the strongest detrimental effects on climate. Higher annual maximum temperatures, higher summer temperatures, and lower summer precipitation under NaC than SSP1 indicate trade-offs with local climate impacts and the resulting need for adaptation. The European continent has warmed by more than 2 °C since the pre-industrial period and is expected to further warm in the coming decades⁸⁶. Even small increases in temperature can increase the intensity and frequency of extreme weather and affect ecosystems and human health^{5,87,88}. The additional warming and drying under NaC could thus further threaten local biodiversity and human well-being, particularly in hot spot areas for heat stress and droughts in Southern Europe but also in the West and East. A detailed assessment of losses in*

species climatic range size (see refs.^{9,89}) or health impacts (see refs.^{87,90}) was, however, outside the scope of this study.”

Line 247: In the discussion, the readers would expect a bit more elaboration about specific cultural/value drivers behind the land use changes under different NFF scenarios, although they have been described in methods. Some unexpected land use transition mentioned above should also be clarified.

Agreed. This should be resolved by the improvements made in response to previous comments, particularly related to see **points A and B**.

Lines 256-258: and more limitations in models, for example, tree age and phenological development after land use transitions and plant adaptation to heat and drought stresses are not represented yet in most ESMs including CLM5.

Thank you for the suggestion. We added the following at the end of the paragraph: *“We acknowledge general simplifications, such as the lack of interannual vegetation development (i.e., we model a new land use state and not a transition), forest age, or interactive plant phenology (i.e., due to use of CLM5 with prescribed leaf area index and canopy height).”*

Line 294: the scale of land cover changes in these NFF scenarios are actually rather small, compared to historical land use changes across many parts of the world since a century ago caused by agricultural expansion and urbanization.

We agree that it would be helpful to contextualise what is small or large. We modified the beginning of the discussion as follows (removing the word “large” before scale): *“Here, we model effects on the distribution of PFTs and find that implementing climate and biodiversity policies could lead to PFT cover changes on 10.0% of European land (49 Mha) in a NfN scenario focusing on biodiversity conservation and land sparing, 10.5% (51 Mha) in a NfS scenario promoting climate change mitigation and land sharing, and 21.0% (103 Mha) in a NaC scenario prioritising cultural landscapes and human-nature interactions. By performing ESM simulations, we demonstrate that land PFT cover changes on **such a scale** can have profound effects on land surface characteristics and thus on local to regional climate.” ... “Effects at the European scale are weaker and comparable with those of recent historical land cover changes in 1992-2015, affecting 70 Mha⁴¹. This can be explained by the limited area converted and compensatory effects between opposing transitions.”*

In the final paragraph of the discussion (the mentioned line 294), we generalised (removing the word “such” before a large scale): *“our results highlight the importance of quantifying potential climate effects of scenarios that transform the land surface on a large scale”.*

In the results, we clarified as follows: *“The climate forcing due to land cover change is thus spatially extensive and modest in intensity over the EU+ region, but strong in greatly modified grid cells and subregions.”*

Lines 295-296: These studies (Luyssaert et al. 2018 and Naudts et al. 2016) and others (Zhang et al. 2024) about unexpected effects of afforestation and deforestation also suggest that the estimated/expected climate effects shown in this study may be subject to uncertainties without fully consider the biogeochemical and biophysical effects of land surface change (see major comments on other variables and attribution methods to consider).

We addressed this concern under two previous comments, with related clarifications in the methods section (i.e., explaining the uncertainty trade-off between SP and BGC modes of CLM5) and in the discussion (i.e., acknowledging the lack of interactive plant phenology).

Lines 361-362: The satellite phenology mode will disable two-way vegetation-climate feedback effects and biogeochemical cycles, which are important for the analysis of land use impact on climate. And please clarify how LAI and canopy height are prescribed for future scenarios after the observational period?

Please see previous responses and clarifications made.

Lines 373-374: Climate forcing from one model "MPI-ESM1-2-HR" and not bias-corrected would have substantial uncertainty.

Please see previous response under main concerns.

References:

1. Jeffrey R. Smith et al. Variable impacts of land-based climate mitigation on habitat area for vertebrate diversity. *Science* 387,420-425(2025). DOI:10.1126/science.adm9485
2. Fastré, C., Possingham, H.P., Strubbe, D. et al. Identifying trade-offs between biodiversity conservation and ecosystem services delivery for land-use decisions. *Sci Rep* 10, 7971 (2020). <https://doi.org/10.1038/s41598-020-64668-z>
3. Thompson, J. R. et al. The consequences of four land - use scenarios for forest ecosystems and the services they provide. *Ecosphere* 7, e01469 (2016).
4. Naudts, K. et al. Europes forest management did not mitigate climate warming. *Science* 351, 597–600 (2016).
5. Zhu, L. et al. Comparable biophysical and biogeochemical feedbacks on warming from tropical moist forest degradation. *Nat. Geosci.* 1–6 (2023) doi:10.1038/s41561-023-01137-y.
6. Bala, G. et al. Combined climate and carbon-cycle effects of large-scale deforestation. *PNAS* 104, 6550–6555 (2007).
7. Betts, R. A., Cox, P. M., Lee, S. E. & Woodward, F. I. Contrasting physiological and structural vegetation feedbacks in climate change simulations. *Nature* 387, 796–799 (1997).
8. Bonan, G. *Ecological Climatology: Concepts and Applications*. (Cambridge University Press, Cambridge, 2015). doi:10.1017/CBO9781107339200.
9. Zeng et al. Climate mitigation from vegetation biophysical feedbacks during the past three decades. *Nature Climate Change* 7, 432–436 (2017).
10. He, M. et al. Amplified warming from physiological responses to carbon dioxide reduces the potential of vegetation for climate change mitigation. *Commun Earth Environ* 3, 1–10 (2022).
11. Chen, L. & Dirmeyer, P. A. Distinct Impacts of Land Use and Land Management on Summer Temperatures. *Frontiers in Earth Science* 8, (2020).
12. Mishra, V. et al. Moist heat stress extremes in India enhanced by irrigation. *Nat. Geosci.* 13, 722–728 (2020).
14. Lobell, D. B., Bonfils, C. J., Kueppers, L. M. & Snyder, M. A. Irrigation cooling effect on temperature and heat index extremes. *Geophysical Research Letters* 35, (2008).

15. Zhang, Y. et al. Asymmetric impacts of forest gain and loss on tropical land surface temperature. Nat. Geosci. 1–7 (2024) doi:10.1038/s41561-024-01423-3.

Reviewer #5 (Remarks to the Author):

Thank you for your contribution.

Reviewer #2 (Remarks to the Author):

Thank you for revising your manuscript and thoroughly addressing my comments. The clarity of the conceptual sections, particularly regarding trade-offs and ecosystem services, has improved significantly. I also greatly appreciate the addition of the new Figure 1.

Thank you for the positive feedback.

However, there remain some ambiguities in the scenarios and assumptions depicted in Figure 1. It is unclear where some of these assumptions originate from, such as the increase of 5 billion trees, urban planning, and others. Could you elaborate on how you developed the rules for the modeling? While Figure 1 is indeed useful, the implementation details of these policies are still somewhat obscure. For instance, what do terms like "Prioritization of areas with high habitat" or "Increased tree density in urban areas" entail? How did you arrive at the figure of 5 billion new trees, and what method governs their distribution? All assumptions underlying the scenarios should be thoroughly justified and explained, especially concerning the species selection for increased tree cover, which significantly impacts temperature outcomes. Moreover, the details and justification of the "Urban planning" component, along with its assumptions, are missing completely. I could not find any explanation, justification, or link to the Nature Future scenarios within the main text.

Thank you for your patience with the scenario description. We reiterate that we did not develop new scenarios in our study. The scenarios were adopted from Dou et al., who operationalised European policies and the Nature Future Framework perspectives in a land use model to generate spatially explicit land system scenarios in the form of maps. The cited paper Dou et al. 2023 is dedicated entirely to the generation of these scenarios and provides all scenario assumptions and technical details. In our study, we used the land system maps as an input to model the spatial distribution of PFTs, which could then be implemented in the Earth System Model to simulate the climate response.

Since we adopted the scenarios from Dou et al., it is outside the scope of our study to re-explain all scenario assumptions and modelling rules adopted in prior work (see Dou et al. 2023, section 2.2.). Nevertheless, we agree that the scenarios should be understandable at high level so the readers can interpret the results. The new introductory text and Fig. 1 serve this purpose, and we made some additional clarifications (see relevant text below, with modifications highlighted).

Introduction: *The Nature Futures Framework was recently used as a lens for developing plural land system scenarios for Europe in 2050³³: a reference that complies with the projected population, commodity demands, and climate under a sustainable low-emissions scenario (SSP1-2.6), and three scenarios that additionally implement targets of the Kunming-Montreal Global Biodiversity Framework and the European Union's Green Deal. **Dou et al.³³ implemented these climate and biodiversity policies while focusing on either intrinsic, instrumental, or relational values according to the Nature Futures Framework (Error! Reference source not found.). (...) Detailed scenario assumptions, their origins, and parameterisation are described in Dou et al.³³.***

Added in the caption of Fig. 1: *Climate and biodiversity policies were derived from the Kunming-Montreal Global Biodiversity Framework and the European Union's Green Deal (see Dou et al. ³³ for details).*

Methods (and corresponding additions in Fig. 1): *The three other scenarios additionally implement targets of the Kunming-Montreal Global Biodiversity Framework and the European Union's Green Deal related to nature conservation (30% of the land area protected by 2030), restoration (1% annual increase in natural areas 2030-2050), increased tree cover (3 billion new trees by 2030, followed by an additional 2 billion new trees by 2050), reduced fertiliser application in agriculture (-20% by 2030, followed by an additional -10% by 2050), and urban planning (improved access to green spaces) (Error! Reference source not found.).*

For convenience and to answer the specific questions raised, we copy some information from Dou et al. below (please see their paper and supplementary information for further relevant details).

Dou et al. 2023: "The identified targets for the EU Green Deal are therefore primarily laid out in the EU Biodiversity Strategy for 2030, and related Forest Strategy and Farm to Fork Strategy (European Commission, 2020; Working group of Convention on Biological Diversity, 2021). We reviewed and compiled respective policy targets and translated them to parameterizations of the land use simulations (Table 2)."

Dou et al. 2023: "We summarized compatible policies from the EU Green Deal and post-2020 GBF within three main policy domains: nature protection and restoration, sustainable agriculture, and increased tree cover. If a specific target has not been mentioned in (one of) the two policy statements, we used targets from the other policy, or assumed a value in line with the overall description in the policy documents. The action points regarding nature protection and restoration, derived from both the GBF and EU Green Deal, refer to expanding current protected areas under the Natura 2000 network from 18 to 30 % of European land by 2030 and to implement a "no-net-loss" policy for all natural ecosystems, meaning that nature outside protected areas may be converted to a different use, but only if compensated by increases in natural area elsewhere in the region. After 2030, we assumed an annual 1 % increase in natural areas until 2050 as a result of ecosystem restoration or rewilding. Natural areas were defined by land system classes referring to low-intensity forest or grassland systems (thereby also including land comprised by the low-intensity agricultural mosaics class). Hereafter, any reference to simulated "natural" land systems or areas will therefore be referring exclusively to these three land systems. The expansion of natural areas was calculated based on the 2015 extent of low-intensity forests. For sustainable agriculture, we focused on the policy that aims at reducing the excessive use of nitrogen. According to the EU Green Deal, the total use of nitrogen in agricultural production should be reduced by 20 % by 2030, compared to 2015. We extended this reduction target to 30 % by 2050. Europe additionally plans to plant 3 billion new trees by 2030 according to the Green Deal. This target was implemented in the scenarios and extended by an additional 2 billion trees to be planted between 2030 and 2050. In addition to these targets, we implemented specific rules addressing the process of urbanization. These rules vary according to the NFF scenario to reflect different perspectives on nature appreciation (e.g., greater urban density to allow for land sparing for nature, vs. lower urban density to increase access to green and blue spaces within cities – see Section 2.2.3)."

Dou et al. 2023: "From the Nature for Nature perspective, priority is given to preserving the intrinsic value of nature and species conservation. To identify where additional protected areas should be implemented within this scenario, we therefore prioritized regions based on the distribution of all vertebrate species known to occur within Europe (Maiorano et al., 2013; O'Connor et al., 2021). (...)

The prioritization was performed using Zonation software (Lehtomäki and Moilanen, 2013). (...) The tree planting target instead involved (1) increasing the density of trees in urban areas and low-intensity agricultural systems, representing an increase in urban parks and (linear) landscape elements in low-intensity agricultural systems, and (2) favoring the expansion of forest and shrub systems.”

Dou et al. 2023: “All specific targets were implemented in CLUMondo by either: (1) introducing new “target goods and services” to be delivered by different land systems (notably for trees, nature areas, and nitrogen use), or (2) by changing land system preferences or conversion rules.”

The “the species selection for increased tree cover” was done in our study, since the land systems of Dou et al. only provide changes in land use and intensity (e.g., low/mid/high intensity forest or mosaic systems) and not forest or tree types. To determine the most appropriate tree type for the environmental conditions of each grid cell in the future, we used (1) spatial predictions of future habitat types across Europe and (2) the observed species composition for each habitat type in plots of the European Vegetation Archive. This approach allowed us to consider that forests have different tree types and densities (and that some area is occupied by other PFTs such as shrubs, grasses, or bare soil) depending on environmental conditions and habitat type. This approach is explained for all ecosystem/land use types in the introduction, which we expanded to improve clarity:

*In this study, we aim to incorporate the land system scenarios of Dou et al.³³ in a regional ESM⁵⁷ to simulate the biogeophysical climate effects of implementing European policies for improved nature management with a focus on biodiversity conservation, climate change mitigation, or cultural services. Implementing these scenarios in an ESM is challenging because ESMs usually represent land use through the spatial distribution of plant functional types (PFTs), which control important ecosystem properties (e.g., vegetation structure, rooting depth, leaf reflectance, photosynthetic capacity) and processes (e.g., reflection of sunlight, evapotranspiration, and photosynthesis). Land use changes such as afforestation, cropland expansion, or land abandonment can be expressed as changes in PFTs. **However, the same land system (e.g., forest/shrub and grassland mosaic) can consist of different PFTs depending on environmental conditions (e.g., cold climates and acidic soils favour needleleaf evergreen trees, whereas temperate climates with marked seasonality and fertile soils support deciduous broadleaf trees). To obtain the PFT composition while accounting for future habitat suitability and plausible species composition, for each land system scenario we harness high-resolution predictions of EUNIS (European Nature Information System)⁶⁰ habitat types at level 3⁶¹ and disaggregate them into PFTs according to the species composition in >800k plots of the European Vegetation Archive (EVA)⁶², the most extensive repository of vegetation records for Europe (Error! Reference source not found. and Methods).***

Our ESM implementation focuses on plant functional types. Changes in ice and urban areas are not considered, as noted in the Methods.

Each assumption translated into the single Nature Futures concept scenarios must be well-explained and justified. E.g, why is "Increased demand for low-intensity grassland" categorized under Nature as Culture value?

This is a scenario assumption of Dou et al. 2023, who interpreted the policy target of nature restoration as expansion of low-intensity forest and grassland systems, i.e., systems with minimal or no human management. This assumption is identical in all three scenarios, as stated in Fig. 1.

Dou et al. 2023: "The target relating to no-net-loss and restoration of natural land systems referred to the preservation and expansion of low-intensity forest and grassland system classes."

I recommend providing a table of scenario assumption rules, including specific numbers, justifications, and references. This table could also clarify how each rule or assumption aligns with the intrinsic, instrumental, or relational values within the Nature for Future framework.

As explained above, the cited paper Dou et al., 2023 is dedicated entirely to the generation of the scenarios. It contains detailed descriptions of the scenario assumptions, their origins, and parameterisation, including specific numbers and references. We believe it was useful to add Fig. 1 so that readers can understand what the scenarios entail, but it is outside the scope of our study to (re-)explain what others have done to generate them. For consistency, we removed citations of input datasets that were required for the work of Dou et al. but not ours (i.e., previous refs. 52-53 which provide the maps of carbon sequestration potential and cultural value used for spatial prioritisation).

Finally, in the abstract, while you now mention "land use management," you continue to refer to "nature management." Please ensure that terms are defined and used consistently throughout the manuscript.

Thank you for pointing this out. In the abstract and main text, we use both "land use management" and "nature management" depending on the context: land use management to focus on land use activities (e.g., crop production) and nature management to focus on the balance between human needs and the health of natural systems (e.g., protection, restoration, and limited land use). We added the following definition: *To explore specific policies for improved nature management (i.e., the use of natural resources balancing human needs and the health of natural systems), drivers of change in biodiversity and ecosystem services must be considered in the scenario development*^{46,49,51}.

Reviewer #3 (Remarks to the Author):

Rereview of Nature Communications NCOMMS-24-79005A

The changes made by the authors to clarify the scenarios used have improved the clarity of the manuscript. I find now Figure 1 a fairly helpful summary of the different scenarios.

Thank you for the positive feedback.

They have also done a good job trying to make clearer how land cover changes from current conditions to SSP1 and thence to their three scenarios. The new Figure 3 does a good job with that. I would suggest the authors edit the figure legend for part 3A though to make it clearer, since the "baseline of the hat graph" may not be clear to readers. It might be possible, for instance, to draw the baseline in another color like red, and then have the figure caption say something like "Differences are shown for the scenarios relative to the reference (SSP1, represented by the red baseline) and recent conditions (represented by the zero line). For instance, forest cover increases by 4% [or whatever] in SSP1 relative to current conditions but is reduced below the SSP1 scenario by all three of our scenarios, and in the case of the NaC scenario forest cover would actually be less than

under current conditions.” (I believe that is the correct interpretation of Figure 3A, but I am not sure!). The authors shouldn’t feel bound by my suggestion, it is just one of I am sure multiple ways to make the caption clearer.

Thank you for the suggestion. We changed the baseline colour and modified the caption as follows: *“a, Land systems are translated into habitat types and plant functional types (PFTs). Differences are shown for the scenarios relative to the reference (SSP1, represented by the red baseline) and recent conditions (represented by the zero line). For instance, forest cover is lower in all scenarios than under SSP1 but higher under SSP1 than it is today. Only the NaC scenario would reduce forest cover relative to recent conditions.”*

Overall, they have made a good faith effort to respond to most of my specific comments, although I will confess I have not had time to carefully review all of them.

It would be helpful if the authors found time to explain, in the Methods or Results or Discussion, more why (if I am understanding Figure 3 correctly) that all three Nature Future Framework scenarios decreased forest cover, relative to SSP1. In particular, I don’t understand (and didn’t go to Dou et al. to check their methods) why under the NfS scenario, there wouldn’t be MORE forest cover than under SSP1. As someone who works for an organization that promotes (where appropriate) reforestation in part for its climate mitigation benefits, it is surprising for me that the NfS scenario has less tree cover than the SSP1 scenario. After all, allowing forest to naturally regenerate after agricultural abandonment (which I assume is part of what is happening under SSP1) could a way to achieve significant carbon sequestration, as it was in the eastern US after agricultural abandonment and subsequent reforestation (and an increase in some rare biodiversity).

This is an interesting question. The pattern is not a new result of our study but originates from the scenarios of Dou et al. (see Fig. 3a). The study of Dou et al. is dedicated entirely to scenario modelling and provides detailed descriptions of the scenario assumptions and parameterisation, analyses of the outcomes, and cross-scenario comparisons. It is outside the scope of our study to (re-)assess their results. Nevertheless, we mention several contributing factors throughout the paper (see relevant text copied further down) and we highlight some aspects more strongly now (modifications are highlighted in bold). Besides the clarification of land system differences (i.e., scenario vs. reference), we now distinguish more clearly between differences in 2025 (i.e., scenario vs. reference) and land use changes over time since 2015 (i.e., scenario 2050 vs. scenario 2015).

In short, the SSP1 scenario assumes a strong decline in agricultural areas, which enables a pervasive expansion of forests (and other natural land). Compared to SSP1, the NfN, NfS, and NaC scenarios must fulfil additional requirements such as restoration of both natural forest and grassland systems and reduced fertiliser use in agriculture. This results in overall more grassland and cropland than under SSP1, leaving less space to forest expansion. This result should not be interpreted as a deviation from SPP1 (i.e., forest loss), but as an alternative future with less forest expansion due to other demands. Indeed, the CLUMondo land system change model used by Dou et al. starts every scenario in 2015 and runs to 2050 considering different demands. In the NfS scenario of Dou et al., carbon sequestration can be achieved in 3 ways: (1) intensification of forestry systems, (2) forest area expansion relative to 2015 (see Fig. 3a), and (3) strategic (re-)allocation of forests to grid cells with high carbon sequestration potential (i.e., spatial prioritisation).

Methods: *The SSP1 storyline for the land use sector includes strong environmental regulation (e.g., reduced deforestation and emissions from land use), increased agricultural productivity, and little growth in food demand including low-meat diets^{43,44}. Based on these assumptions, land use projections for Europe under SSP1 by 2050 indicate a **strong** decline of pastures and annual crops, which enables the **pervasive** expansion of forests and (semi-)natural vegetation^{33,43}.*

Results: *To quantify the effects of **improved** nature management **by 2050**, we compare the NfN, NfS, and NaC scenarios with the SSP1 reference, which complies with future demands and climate but does not include specific European climate and biodiversity policies. To contextualise the land system differences and their effects on PFT cover **in 2050**, we also analyse **land use changes over time since 2015 under the different scenarios**. Changes in land systems, habitat types, and PFTs are shown in **Error! Reference source not found.a**. Under SSP1, mid- and high-intensity forest systems **strongly expand after 2015**, and we find a corresponding increase in tree PFT cover, particularly broadleaf deciduous trees (**Error! Reference source not found.a**).*

*At the European level, all three scenarios project a substantial expansion of low-intensity systems to meet the targets for protection, restoration, tree cover, and sustainable agriculture (see Fig. 1 and Dou et al.³³ for details). This conversion and extensification is enabled by intensification of mid-intensity forests and croplands to satisfy demands. **Consequently, the scenarios hardly expand forests but increase grasslands after 2015. As a result, they have less tree PFT cover and more grass PFT cover than the reference and tend to counterbalance the changes under SSP1 in 2050 compared with recent PFT cover (Error! Reference source not found.c)**. (...) the NfS scenario promoting climate change mitigation shows the strongest increase in high-intensity systems, particularly forests.*

Discussion: *Particularly at the local scale, **differences between scenarios** can be difficult to interpret or seem counterintuitive. The results are path-dependent and complex, and the link to the initial scenario assumptions is not always evident (e.g., **NfN requires more cropland than SSP1 to compensate for extensification, NfS can afford less forest expansion than SSP1 but carbon accumulation could further be enabled by intensification and strategic placement of forests in productive areas**).*

We hope this is informative for someone working to promote reforestation. Yet, we highlight that the NfS scenario should not be understood as a “reforestation scenario” compared with SSP1 because forest carbon sequestration is promoted compared with 2015 and it is only one goal among several (see Fig. 1). Please also note that, generally, the scenario approach of Dou et al. does not optimise for certain outcomes, see Discussion: *However, the scenarios are driven by policy means and not their outcomes, so they are not necessarily efficient at achieving policy goals (e.g., NfS promotes forests in areas with high carbon sequestration potential but does not maximise carbon sequestration).*

Reviewer #4 (Remarks to the Author):

Thanks for authors' thorough revisions and responses, which have considerably improved the manuscript. The methodological clarifications, restructuring of the driver analysis, and strengthened discussion represent major progress compared to the original submission. Most of my initial concerns have been adequately addressed. Nevertheless, a few important issues remain that require

further clarification or minor additions to ensure transparency and robustness of the study's conclusions. The detailed comments are as follows:

Thank you for the positive feedback.

1. The use of the satellite phenology (SP) mode in CLM5 is still a key limitation and the justification for carbon sequestration results can still be improved. I am not asking for redoing BGC simulations at this stage but choosing SP mode definitely limited the robustness of your findings regarding carbon and climate responses of nature management scenarios. Computational cost is not a strong motivation nowadays for this choice. It is partly true that this simplification reduces uncertainties related to the seasonality (monthly phenology) of vegetation structure, but you lost the big picture of carbon cycle responses to land use changes. Any carbon cycle variable (including GPP) generated in the SP mode and its interannual dynamics would be hardly comparable with observations (you can try a comparison with FLUXNET or one of MODIS-based GPP products for example) because the nitrogen cycle, soil biogeochemistry and carbon cycle-climate feedbacks are all turned off in SP and only monthly climatology of LAI, SAI and canopy height are prescribed, without interannual variations. Given this key limitation, it's good that GPP figures are moved to the supplement, but the following concerns should be addressed. In the revised methods, it remains unclear how future vegetation structure for 2034-2050 is specifically prescribed, i.e., whether monthly LAI and canopy height are repeated from historical satellite climatology? This is very likely the case, as SSP scenarios prescribe land cover change in the future according to LUH2 but hardly know the future vegetation structure and phenology. Thus, the potential implications of this static assumption and neglect of interannual and decadal variations in vegetation structure should be discussed. Even under the low-emission SSP1-2.6 scenario, such a static prescription could overestimate afforestation benefits (afforested grid cells or land-units would abruptly attain the climatology LAI and canopy height without undergoing a lengthy growing stage from seedling to mature as in the full BGC mode) and underestimate deforestation and degradation impacts by ignoring perturbations to soil carbon pools that are only interactively simulated in the BGC mode. So on. The actual differences between SP and BGC modes are huge – SP means only biogeophysical modules such as water and energy fluxes (including stomatal response) are active but the vegetation structure is not responsive to climate (no interannual vegetation dynamics), whereas the prognostic BGC mode involves full above- and below-ground biogeochemical cycles and their interaction with climate (including feedback loops between vegetation dynamics, carbon cycle and the climate system). Simply putting "The only difference lies in whether LAI is prescribed or simulated, and the prescribed LAI might be less uncertain than the predicted LAI" is not convincing and misleading. Please also remove the CLM aspects of biogeochemical processes and biological processes such as "Biogeophysical and biogeochemical processes are simulated independently for each PFT, but the atmosphere is assumed horizontally homogeneous within each grid cell" in Methods because they are only for the BGC mode. Also note that nitrogen cycle and its effects on carbon cycle are fully turned off in the SP mode of CLM5, so please remove 'nitrogen' from the new sentence: "stomatal conductance and photosynthesis, which both respond to environmental conditions including temperature, radiation, humidity, CO₂ concentration, day length, and availability of water and nitrogen". I recommend (a) explicitly stating in the Methods whether LAI and canopy height are based on a repeated historical satellite climatology, and (b) expanding the Discussion to address the potential biases in carbon sequestration

findings from this static assumption and the limitations of not enabling BGC and not considering carbon cycle-climate feedback associated with land use change.

We agree that the SP mode has important limitations. However, we believe these limitations are not critical for our main findings because, e.g., we focus on comparisons between scenarios, we evaluate different future land use states and not transitions over time, we analyse multi-year seasonal means and not interannual variation, the scenarios are moderate and thus unlikely to trigger substantial carbon cycle-climate feedbacks. The BGC mode also has important limitations and uncertainties (e.g., absence or very simplified representation of human land use/management practices such as grazing, harvesting, etc.) and there is no guarantee (and no evidence) that the prognostic vegetation structure would turn out more realistic at the spatial and temporal scales of our study (although benefits have been demonstrated for individually parameterised sites and interannual variation). Nevertheless, we share the concern that such nuances might not be evident to all readers, so we implemented the suggestions to improve clarity and eradicate potentially misleading statements.

Methods: In the model description of CLM5, we removed “biogeophysical and biogeochemical” so the sentence now reads: *“Processes are simulated independently for each PFT, but the atmosphere is assumed horizontally homogeneous within each grid cell.”*

Methods: We added the following clarification: *“CLM5 is configured in satellite phenology mode, i.e., for each grid cell and PFT, monthly leaf area index, stem area index, and canopy height are prescribed **based on a repeated** monthly climatology obtained from satellite data.”*

Methods: We removed nitrogen so the sentence now reads: *“... which both respond to environmental conditions including temperature, radiation, humidity, CO₂ concentration, day length, and **water availability.**”*

Methods: We removed the statement about computational cost. We moved the advantages/disadvantages of the SP mode to the discussion of limitations (see below).

Discussion: We added the following clarification: *“Reduced vegetation carbon uptake ($-11 \pm 0.4\%$) suggests lower potential for ecosystem-based mitigation, but dedicated simulations **with interactive biogeochemistry** would be needed to quantify the net response of the land carbon cycle (i.e., additionally considering respiration, disturbances, and soil carbon dynamics) **and to represent the underlying biogeochemical processes.**”*

Discussion: We expanded the part on model-related limitations: *“We acknowledge simplifications in our model setup, such as the lack of interannual vegetation development as we model a new land use state and not a transition over time. Furthermore, prescribing leaf area index and canopy height based on observations reduces uncertainties related to the reliable prediction of vegetation structure but omits certain aspects of vegetation-climate coupling. Specifically, our results do not capture the effects of interannual variability and climate change on vegetation growth and phenology and their consequences for biogeophysical processes and carbon uptake.”*

2. The authors’ rationale for using MPI-ESM1-2-HR is now convincing, and the comparison with other CMIP6 models in the supplementary notes and additional sensitivity analysis of regional climate response to the background climate are very helpful. However, please also acknowledge in the Discussion or the relevant Methods section that the study relies on a single uncorrected GCM, that

inter-model spread remains an important source of uncertainty, and that boundary conditions were not bias-corrected. Then you can add your argument that “we explore possible future scenarios (so predictability is not of major importance) and we investigate differences between scenarios under identical background climate (so all scenarios face the same forcing biases).”

Thank you for this suggestion. In the discussion, we further expanded the part on model-related limitations: *“Further uncertainty results from the reliance on boundary conditions from one global climate model that performs well over Europe, MPI-ESM1-2-HR. A single model cannot represent the range of plausible outcomes under the SSP1-2.6 scenario, so we cannot assess the sensitivity of the climate response to the large-scale background climate simulated by different global climate models (see Supplementary Notes for a comparison of candidate CMIP6 models). This aspect is still less explored than the substantial uncertainty resulting from the representation of land use and land cover change processes in different regional and global climate models^{36,40} (but see refs.^{42,97}).”*

We decided not to include aspects on predictability and bias-correction because throughout the study, we analyse differences between scenarios and not absolute values of, e.g., temperature.

3. The new results “Drivers of temperature response” based on two separate attribution analyses are very good. But in the text, “The warming effect is partly compensated by increased sensible heat flux, ground heat flux,” may need some revision or clarification. Currently, Fig. 7 shows the individual effects of changes in these fluxes on surface/skin temperature. But such description of increased sensible heat together with decreased latent heat is a typical phenomenon associated with surface air warming and the increased ground heat flux is also the driver of soil temperature increase. Thus, it may be better to differentiate the responses of surface temperature and 2-m air temperature here.

Thank you for pointing this out. Ideally, we would quantify the drivers of the 2m air temperature response, but surface energy balance decomposition applies to the surface skin temperature response (which often is a good proxy for the 2m air temperature response, as shown in Fig. 7). All text on the contribution of individual fluxes in this section applies to surface skin temperature. We clarified this in the text by adding “surface” in the beginning of the paragraph: *“The NaC scenario shows an important **surface** warming contribution from reduced latent heat flux”*.

4. The addition of results for warming and drying in hottest/dryest and most heating/drying areas of Europe increases the policy relevance of the study. The definition of “most heating areas” is not very clear at the moment, especially in this sentence “but the most heating areas under SSP1 would not get significantly warmer” which is confusing. Although the authors say that detailed heat impact analysis is outside the scope of this study, the following sentence is not accurate: “The additional warming and drying compared with recent climate could further threaten local biodiversity and human well-being, particularly in hot spot areas for heat stress and droughts in Southern Europe but also in the West and East”, because drying and reduced humidity due to conversion of irrigated cropland to grassland can mitigate (rather than further exacerbate) heat stress caused by dry-bulb temperature alone. I suggest that the authors explicitly clarify in the Discussion that the study focuses on temperature and precipitation only, and that interactive humidity effects on heat stress and human well-being were not assessed. This will help readers correctly interpret the limitations of the study regarding health impacts of different scenarios.

Thank you for pointing this out. The definition is provided in the previous paragraph: the 1% most extreme areas in the SSP1 reference (e.g., hottest areas), and the 1% most changing areas in SSP1 compared with recent climate (e.g., most heating areas). We improved the sentence as follows: *“The hottest areas under SSP1 would experience 0.23 ± 0.08 °C **additional** temperature rise **under NaC**, but the most heating areas would not get significantly warmer **under NaC than SSP1.**”*

We agree and removed the term “heat stress”. The part now reads: *“The additional warming and drying under NaC could thus further threaten local biodiversity and human well-being, particularly in hot spot areas for **heat and drought** in Southern Europe but also in the West and East. A detailed assessment of losses in species climatic **range size**^{9,89} **or health impacts**^{87,90} was, however, outside the scope of this study.”*

Reviewer #5 (Remarks to the Author):

Thank you for your contribution.

Reviewer #2 (Remarks to the Author):

Thank you for the detailed clarification of my concerns. The manuscript has improved. No further comments from my side.

Thank you for the positive feedback.

Reviewer #3 (Remarks to the Author):

This document is a review of a revision (NCOMMS-24-79005B), in response to comments from myself and other reviewers.

Overall, the paper has improved upon revision, I do not have any other major comments or critiques.

Thank you for the positive feedback.

Small comments:

Note that the letter to reviewers has a formatting error at a few points where it says "Reference source not found". I assume this is just a formatting error from citation software after copying this text from the manuscript document to the letter for reviewers, but the authors might want to check that all the citations are correct in the main manuscript text.

The citations are correct in the main manuscript text. This was indeed due to copying into the response letter. We removed the citations and errors from the response letter and provide an updated version.

I appreciate the additional info on the Dou et al. scenarios added to the Introduction it has made the paper clearer. Figure 1 is very helpful. I also appreciate the text in the letter to reviewers, describing the Dou et al. methods more. In a sense, the authors have done what they can to make the Dou et al. methods clear. I disagree with some of the assumptions baked into the Dou et al. scenarios regarding reforestation, but it is not that those assumptions are wrong per se (there are multiple future scenarios possible), and they are descriptions of one plausible future.

Thank you for the positive evaluation. Some of the land use changes reflected in Dou et al. originate not from assumptions in the CLUMondo model but from the output of the Integrated Assessment Model GLOBIOM for the SPP1 storyline (which is used as an input to CLUMondo). We clarified this in the Methods section. We agree that the resulting scenarios reflect plausible futures (not most likely or optimal), which we further clarified in the Discussion section.

I appreciate the changes to Figure 3 and its caption, which have improved its readability.

Thank you.

Reviewer #4 (Remarks to the Author):

Thanks for the authors' careful responses. I don't have further comments.

Thank you for the positive feedback.

Reviewer #5 (Remarks to the Author):

Thank you for your contribution.